# Global-scale prevalence of low nutrient use efficiency across major crops

Ji Liu [1,2,3,4], Hai Wang[2], Josep Penuelas [3,4], Juan Mou[2], Manuel Delgado-Baquerizo [5], Jordi Sardans [3,4], Fernando Coello [3,4], Zhi Quan [6,7], Tianyi Qiu[8], Yanyan Li[9], Yahui Guo[2], Ziying Hu[8], Yanrui Ying[8], Jingyi Lv[8], Yufeng Zhang[8], Wenfeng Tan [10], Guiyao Zhou[5], Lu-Jun Li[11] & Linchuan Fang [8] ✉

Enhancing nitrogen (N) and phosphorus (P) use efficiency (NUE and PUE) is essential for advancing sustainable agriculture and reducing dependency on non-renewable fertilizers. However, the long-term dynamics of NUE and PUE across major crops remain poorly understood at a global scale. Here, we compile a comprehensive global database encompassing 3360 observations across 205 countries to analyze trends in NUE and PUE for major crops from 1961 to 2018. Today, PUE and NUE are still suboptimal, particularly in developing regions, emphasizing the need for context-specific strategies to improve nutrient use efficiency. Global mapping shows that NUE and PUE are highly context-dependent, with variations observed by crop type and region. For instance, rice achieves optimal NUE and PUE in tropical zones, while wheat performs best in temperate climates. Notably, maize continues to exhibit significant nutrient inefficiencies, especially in China and the United States, with considerable N and P surpluses. Taken together, this global analysis provides spatially explicit insights to guide region-specific efforts toward improving nutrient use efficiency, supporting sustainable agricultural practices and reducing global fertilizer dependence.

Nitrogen (N) and phosphorus (P) are critical nutrients that limit food production globally, as they are essential for plant growth, protein synthesis, energy transfer, and genetic replication[1,2]. Enhancing the efficiency of inorganic N and P use (NUE and PUE) is vital for promoting agricultural sustainability, enabling high crop yields while safeguarding

ecosystems from eutrophication and soil acidification[2-4]. Thus, nutrient use efficiency is integral to achieving food security and protecting the environment, serving as a valuable metric for shaping global environmental policies. Unfortunately, the widespread adoption of fertilizers, driven by the Haber-Bosch ammonia synthesis process and the

[1]State Key Laboratory of Loess Sciences & National Observation and Research Station of Earth Critical Zone on the Loess Plateau, Institute of Earth Environment, Chinese Academy of Sciences, Xi'an, China. [2]Hubei Province Key Laboratory for Geographical Process Analysis and Simulation, Central China Normal University, Wuhan, China. [3]CSIC, Global Ecology Unit CREAF-CSIC-UAB, Bellaterra, Catalonia, Spain. [4]CREAF, 08193 Cerdanyola del Vallès, Bellaterra, Catalonia, Spain. [5]Laboratorio de Biodiversidad y Funcionamiento Ecosistémico. Instituto de Recursos Naturales y Agrobiología de Sevilla (IRNAS). Consejo Superior de Investigaciones Científicas (CSIC). Av. Reina Mercedes 10, Sevilla, Spain. [6]Institute of Applied Ecology, Chinese Academy of Sciences, Shenyang, China. [7]Weifang Institute of Modern Agriculture and Ecological Environment, Weifang, China. [8]Key Laboratory of Green Utilization of Critical Non-metallic Mineral Resources, Ministry of Education, Wuhan University of Technology, Wuhan, China. [9]College of Ecology and Environment, Xinjiang University, Urumqi, Xinjiang, China. [10]College of Resources and Environment, Huazhong Agricultural University, Wuhan, China. [11]State Key Laboratory of Black Soils Conservation and Utilization, Northeast Institute of Geography and Agroecology, Chinese Academy of Sciences, Harbin, PR China. ✉e-mail: flinc629@hotmail.com

expansion of industrial phosphate mining, has resulted in a decoupling of nutrient availability from nutrient use efficiency. From 1961 to 2018, the application of inorganic N and P fertilizers surged from 13.0 Tg N yr$^{-1}$ and 5.04 Tg P yr$^{-1}$ to 109 Tg N yr$^{-1}$ and 19.4 Tg P yr$^{-1}$ on cropland, respectively (Supplementary Fig. 1). This escalating trend has surpassed sustainable thresholds (52–113 Tg N yr$^{-1}$ and 6–12 Tg P yr$^{-1}$)[5], raising concerns particularly regarding the non-renewability of phosphate rock and ecological pollution[6–8]. Consequently, improving NUE and PUE is essential for realizing the goals of sustainable agriculture, balancing high crop yields with environmental protection[9,10].

Despite the importance of NUE and PUE, significant knowledge gaps remain regarding their global patterns and drivers. First, current N and P balance (N(P)UE$_{bala}$) approach approaches[11], which incorporate national-scale fertilizer inputs—including atmospheric deposition, biological N fixation, and both inorganic and organic fertilizers—and nutrient harvests (including grain and residue), are the predominant methods for assessing global NUE and PUE[2,3] (See Supplementary Table 1 for detailed definitions and calculation formulas). However, comprehensive investigations are lacking that forecast the fate of NUE and PUE across diverse climates and crop species. This gap hinders the identification of hotspots and mechanisms influencing NUE and PUE at finer spatial scales[12,13]. Ecological theory suggests that while P is the primary limiting factor in tropical regions, N limits productivity in higher latitudes[14,15], underscoring the need to understand how NUE and PUE interact with biogeographical patterns of nutrient availability under various fertilization regimes. Thus, understanding how NUE and PUE interact with global biogeographical patterns in nutrient availability under different fertilization regimes is critical to forecasting the future of sustainable agriculture. Second, the temporal dynamics of NUE and PUE across various crop species are not well understood[1,2]. Although the rise in fertilization during the 1960s had a significant impact on both NUE and PUE, there is a notable lack of comprehensive global analyses examining these changes across major crop species in recent decades. Gaining insights into these dynamics is essential for elucidating the varying capacities of developed and developing countries to effectively regulate NUE and PUE. Finally, existing assessments of NUE and PUE at large spatiotemporal scales primarily rely on the N(P)UE$_{bala}$ method, which does not directly reflect the actual utilization efficiency of inorganic fertilizers. In contrast, the N and P difference (N(P)UE$_{diff}$) method provides a clearer indication of the effects of inorganic fertilizers by directly comparing crop yields and soil nutrient changes between fertilized and unfertilized controls, thereby enhancing assessment accuracy[16]. Findings from China's grain ecosystems illustrate this, where NUE$_{diff}$ and NUE$_{15N}$ ($^{15}$N tracer) ranged from 0.27 to 0.37, in contrast to an approximate NUE$_{bala}$ value of 0.68[11]. This suggests that NUE$_{15N}$ ≈ NUE$_{diff}$ < NUE$_{bala}$, emphasizing the need for a global assessment based on NUE$_{diff}$ to identify historical trends, spatial patterns, and primary drivers of inorganic NUE and PUE.

Here, we aim to address these knowledge gaps by compiling a comprehensive global database on nutrient use efficiency (NUE and PUE), comprising 3360 observations across 205 countries and regions (Fig. 1). We analyze trends and patterns of NUE and PUE using the N(P)UE$_{diff}$ framework for major crops, utilizing dynamic national-scale data that includes crop yields, cropland areas, fertilizer N and P input intensities, nutrient uptake by crops, and residue-grain ratios. Our focus on key global crops—specifically rice, wheat, maize, and soybean—that collectively account for over half of global crop production and 49% of cropland area[17,18]. Additionally, we develop machine learning models utilizing point-scale NUE and PUE data ($n = 2354$ for NUE, $n = 1006$ for PUE) alongside climate, soil properties, and agricultural management information to delineate the current global spatial distribution patterns of NUE and PUE for these crops and identify their underlying drivers. The primary objectives of our study are: 1) to quantify trends in overall and major crop NUE and PUE from 1961 to 2018; 2) to identify current spatial distribution patterns and drivers of

NUE and PUE; and 3) to pinpoint regions with high (>50%) and low (<50%) NUE and PUE to direct targeted improvements (Supplementary Fig. 2).

## Results and Discussions
### Historical trends in NUE and PUE
Since 1961, global applications of N and P fertilizers have shown a consistent upward trend, though the growth rates for N inputs have decelerated since 1985 ± 1 (95% confidence interval) and for P inputs since 1979 ± 2 (Fig. 2a, b). Notably, between 1961 and 1975, NUE and PUE declined progressively in response to increasing fertilizer application rates (Fig. 2c). However, beginning around 1972 ± 1 and 1975 ± 3, a modest upward trend in both NUE and PUE emerged, indicating enhancements in fertilizer utilization efficiency. This improvement can likely be attributed to advancements in technology and shifts in agricultural policy, particularly the reduction in fertilizer input intensities aimed at improving nutrient efficiency (Supplementary Fig. 1). This pivotal breakpoint aligns with previous assessments based on the N(P)UE$_{bala}$, which identified a similar trend around 1980[2,19]. The observed upward trajectory in NUE and PUE coincides with the end of the first Green Revolution, a period characterized by significant increases in agricultural productivity from the 1960s to the 1980s[20].

The first Green Revolution spurred a substantial global increase in food production, primarily through the adoption of high-yielding crop varieties and intensified fertilizer use[20]. While this transformation enhanced short-term productivity, it coincided with a period of decline in estimated NUE and PUE. Several factors may have contributed to this pattern. For example, high-yielding varieties often required substantial nutrient inputs, which encouraged the routine application of large amounts of N and P fertilizers[9]. Prior studies suggest that this approach, while effective in boosting yields, placed limited emphasis on improving nutrient uptake efficiency. Moreover, continuous heavy use of inorganic fertilizers—particularly nitrogen—has been associated with soil acidification, which can negatively affect nutrient absorption[7,12]. During this period, breeding programs primarily targeted yield and disease resistance, rather than traits related to nutrient efficiency[20]. However, we note that our analysis does not explicitly model these agronomic or soil-related mechanisms; thus, the observed decline in NUE and PUE during the early Green Revolution should be interpreted as a temporal correlation rather than a direct causal relationship. In summary, while the Green Revolution successfully addressed food security challenges, it may also have contributed to inefficiencies in nutrient use that persisted into subsequent decades.

Developed countries consistently exhibit higher NUE and PUE compared to developing nations ($P < 0.05$). However, developing countries have been more efficient in NUE since 1977 ± 1, gradually closing the efficiency gap (Fig. 2d). These disparities in NUE and PUE can be attributed to a combination of environmental, agricultural, and socioeconomic factors. Developed nations, often benefiting from stricter environmental policies and government interventions, have increasingly shifted toward more sustainable nutrient management practices. For example, policy incentives, such as subsidies for sustainable farming practices and regulation of nutrient input intensities, have played a significant role in promoting efficient nutrient use[21]. In contrast, developing countries tend to prioritize yield maximization over efficiency, driven by high population pressures, limited arable land, and less robust policy frameworks. Socioeconomic factors, such as farmer behavior, economic incentives, and limited access to technology, contribute to less efficient nutrient use in these regions. Since the 1970s, countries, such as the United States and members of the European Union, have transitioned from a focus on yield-driven increases in nutrient inputs to a more sustainable development approach that emphasizes optimized nutrient management[2,3]. This shift has been partly motivated by concerns regarding water quality

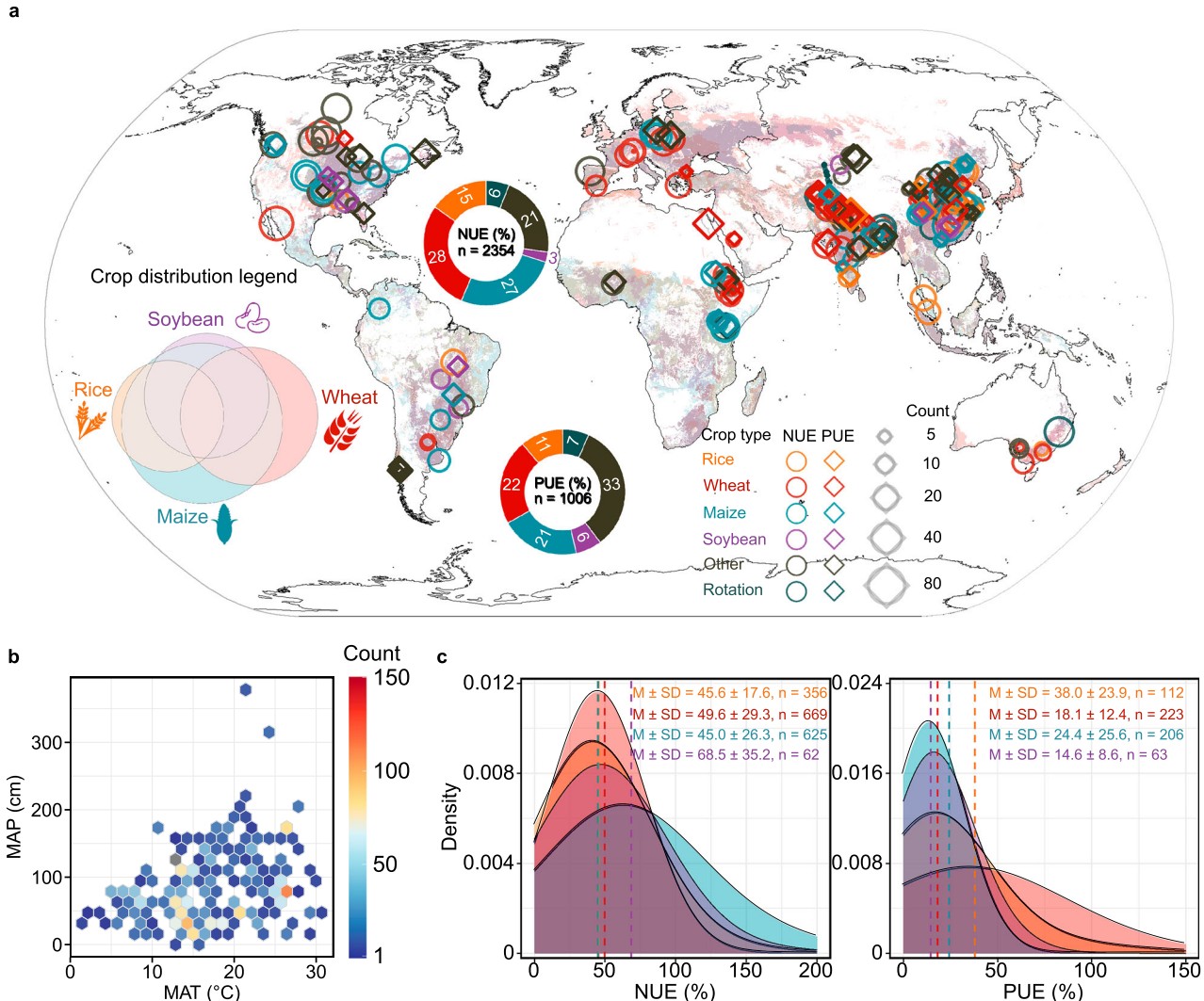

**Fig. 1 | A global assessment of field-based nitrogen (NUE) and phosphorus use efficiency (PUE) across four major crops (rice, wheat, maize, soybean). a** The circular legend indicates the global distribution of the four major crops, where the overlap represents intercropping and rotational cropping. Crop plantation base map from SPAM 2010 V2r0[70]. **b** The distribution of temperature and precipitation at the sample points. MAT, mean annual temperature; MAP, mean annual precipitation. Count represents the frequency of the collected samples under the same location and within similar climatic conditions. **c** The four colors of the crops are referenced to the global distribution of four major crops, distributed from top to bottom to represent rice, wheat, maize, and soybean. The letter M, SD, n refer to the mean value, standard deviation and count, respectively.

degradation, resulting in stringent regulations on nutrient application rates to minimize N and P losses[21,22]. For instance, in the United States, N fertilizer input intensity stabilized at 110 kg N ha$^{-1}$ after 1988, while P input intensity peaked at 22.3 kg P ha$^{-1}$ in 1972, followed by a decline and subsequent stabilization. Similarly, the European Union experienced a plateau in N fertilizer input intensity after 1988, with P input intensity peaking at 40.4 kg P ha$^{-1}$ in 1978, subsequently witnessing a gradual reduction in fertilizer applications. Conversely, developing countries, including China, India, Brazil, and Indonesia, continue to face significant challenges from rapid population growth, rising food demands, and limited economic incentives for sustainable farming practices. These pressures likely contribute to continued reliance on intensive fertilizer use in these regions, potentially delaying alignment with global trends toward reduced input intensities.

Looking ahead, since the first Green Revolution, the total amount and intensity of N and P fertilizer inputs globally have continued to increase at a rate of 1.21 Tg N yr$^{-1}$ and 0.14 Tg P yr$^{-1}$, 0.46 kg N ha$^{-1}$ yr$^{-1}$ and 0.03 kg P ha$^{-1}$ yr$^{-1}$, albeit at a slower pace (Fig. 2 and Supplementary Fig. 1). However, NUE and PUE have shown some improvement in recent decades, which may be associated with a

broader shift in agriculture toward sustainability as a guiding principle. Advancing a second Green Revolution could involve strategies, such as developing high-efficiency crop varieties, deploying precision agriculture, promoting soil health through rotation and intercropping, and leveraging biological nutrient sources, including P-solubilizing and N-fixing microorganisms. While not directly assessed in this study, these practices have been proposed in prior work as potential avenues to enhance nutrient use efficiency. Ensuring the comprehensive implementation of these strategies, especially in developing nations, requires providing economic incentives to farmers. Financial support, subsidies for adopting sustainable practices, and access to affordable technologies can incentivize farmers to embrace these innovations, facilitating their widespread adoption and ensuring long-term success[20,22–26]. Further, inorganic fertilizer input intensities have gradually stabilized or even declined in the world's major agricultural countries (Supplementary Fig. 3), especially in China, the world's largest user of inorganic fertilizers. N and P input intensities in China peaked in 2012 and 2010, suggesting that China has taken an important step forward in sustainable agricultural practices in recent years.

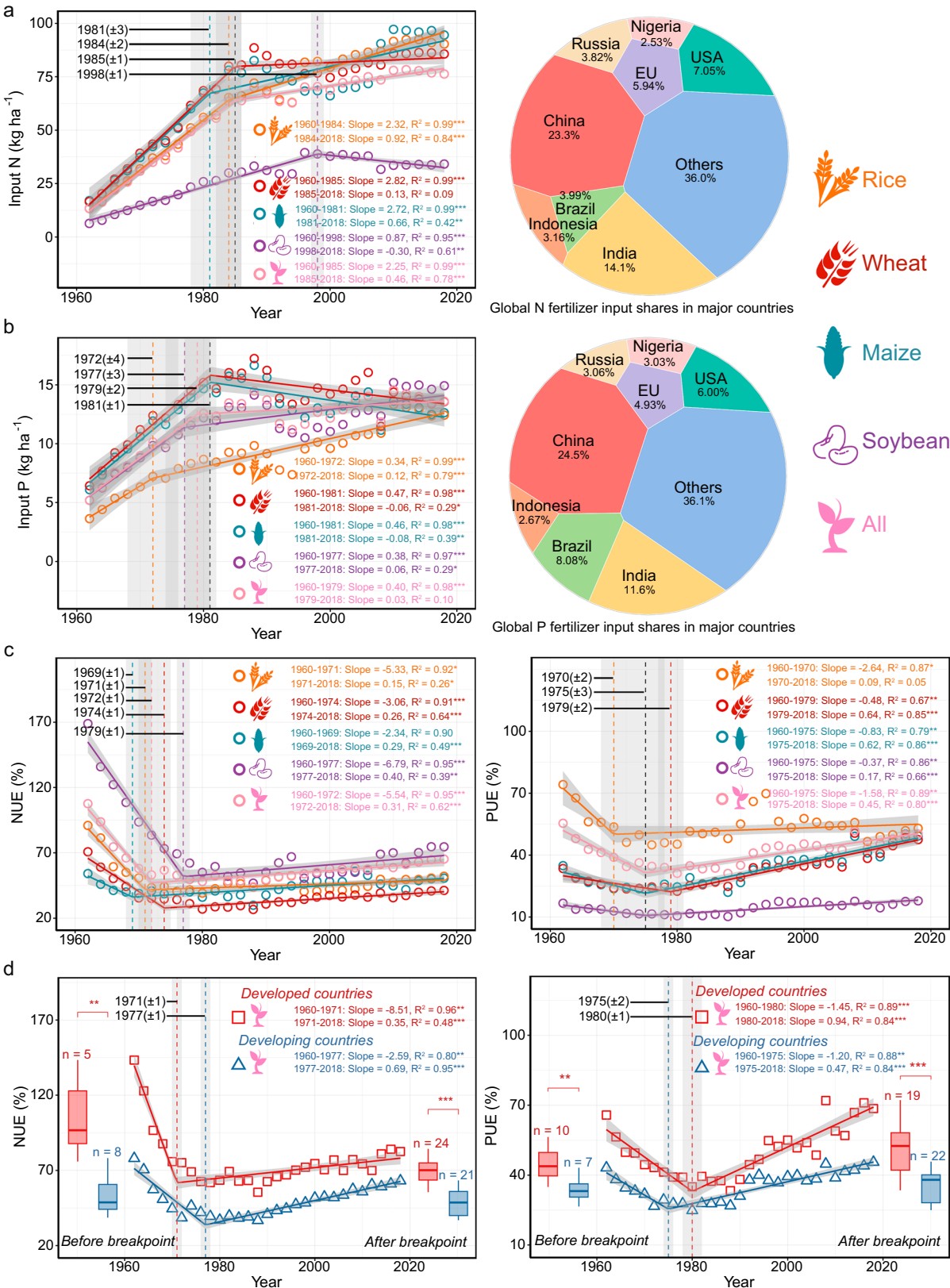

## Global hotspot of NUE and PUE

Under the heterogeneous influences of global climate, soil properties, and agricultural management, NUE and PUE in major crops exhibit substantial spatial variability and generally low overall efficiencies (Fig. 3). Only wheat and soybean achieve NUE values exceeding 50%, while PUE remains below 50% across all major crops (Fig. 4),

highlighting a widespread risk of inorganic fertilizer over-application on a global scale.

Our comparative analysis using the $N(P)UE_{diff}$ and $N(P)UE_{bala}$ methods, based on data from 2008 to 2012, reveals markedly lower PUE values with the $N(P)UE_{diff}$ approach 26[2,3]. For instance, PUE for rice, wheat, maize, and soybean is 44%, 23%, 34%, and 19%, respectively,

**Fig. 2 | Temporal patterns in nitrogen (N) and phosphorus (P) fertilizer inputs and nitrogen (NUE) and phosphorus use efficiency (PUE) worldwide. a–d** Trends in crop fertilizer input and utilization efficiencies over time are fitted using a segmented fit, with solid lines indicating segmented fitting lines and corresponding shaded areas representing 95% confidence interval of the fitting lines. The significance of the non-zero coefficients in the segmented model fitting is evaluated through a two-tailed t-test. If $p > 0.05$, it indicates that there is no obvious trend of value over time. The data marked in the upper left corner are the values of vertical dashed lines and shaded areas, indicating breakpoints (turning points) and standard error, respectively. The color of dashed lines is consistent with that of the crop legend, and black indicates that the breakpoints of multiple crops are consistent. The major countries shown in the pie chart are the eight countries with the largest crop cultivation area. The significance test of the difference in all crops in developed and developing countries is represented by a box plot based on the Wilcoxon method, using a two-tailed test. For the boxplot, the straight line in the center represents the median, or second quartile (Q2), the top edge of the box represents the third quartile (Q3), and the bottom edge of the box represents the first quartile (Q1). n represents the number of data points in each box. *, $p < 0.05$; **, $p < 0.01$; ***, $p < 0.001$.

with the $N(P)UE_{diff}$ method, in contrast to 63, 80, 74, and 76% with the $N(P)UE_{bala}$ method. Notably, NUE remains relatively consistent between the two methods. These findings suggest that current crop P uptake relies primarily on soil P mobilization rather than inorganic fertilizer inputs. This trend, likely driven by historical, intensive P fertilizer application, has gradually pushed soil P reserves toward saturation[2,27]. While this process has helped meet crop P demands in the short term, it has also created long-term sustainability challenges. P availability in soils is influenced not only by inorganic fertilizer inputs but also by soil organic matter, which plays a crucial role in enhancing P availability by releasing organic acids that solubilize bound P[12,28]. However, many soils worldwide, particularly in regions with low organic matter content, fail to fully benefit from this mechanism, limiting the PUE. To further improve PUE, adopting integrated nutrient management strategies is critical. These strategies include optimizing P fertilizer placement, improving irrigation practices, and managing crop rotations to minimize continuous cultivation of high-P-demanding crops[12,29–31]. In many developing countries, the lack of access to soil testing facilities or failure to implement recommendations based on soil tests exacerbates inefficient P use, leading to imbalanced fertilization and suboptimal P availability[32,33]. In addition, minimizing fallow periods and managing soil organic matter content through practices, such as cover cropping or organic amendments, could enhance P availability and utilization[34]. Thus, while regulating the release of retained soil P could reduce P fertilizer inputs over time, it is crucial to adopt a holistic approach that addresses the complex factors influencing P availability and uptake.

Recognizing the distinct NUE and PUE patterns across different climates offers a pathway for optimizing crop distribution based on regional characteristics. Such strategic crop deployment could reduce fertilizer dependency in high-efficiency areas, allowing for resource reallocation to regions with lower nutrient efficiency (Fig. 4 and Supplementary Fig. 4).

Rice exhibits superior NUE (Mean ± SD = 47 ± 6.7%) and PUE (49 ± 15%) in tropical regions, outperforming the global average in countries, such as Indonesia, Myanmar, and Nigeria (Supplementary Fig. 4a). This advantage can be attributed to multiple factors: 1) tropical countries, often in the developing world, tend to have lower N and P inputs (76.8 kg N ha⁻¹ and 12.6 kg P ha⁻¹) relative to the global average (95.3 kg N ha⁻¹ and 16.3 kg P ha⁻¹), thus avoiding nutrient surplus and achieving higher NUE and PUE[2,3,16]; 2) as a C3 plant, rice maintains high photosynthetic efficiency and rapid growth under the high temperatures and ample sunlight of tropical conditions, optimizing N and P utilization for protein synthesis and energy metabolism[16]; 3) the prevalent practice of long-term flooding in tropical rice cultivation creates a water layer that reduces N volatilization and enhances P solubility, thereby improving nutrient utilization efficiencies[35,36]. Additionally, this water layer suppresses weed growth, reducing nutrient competition; 4) the warm, wet tropical climate supports the development of deeper, denser root systems in rice, enhancing nutrient absorption capabilities[37]; 5) the long growing season facilitates efficient nutrient uptake during critical growth stages, potentially allowing for multiple cropping cycles each year[16].

Wheat in temperate zones, including China, demonstrates higher NUE (53 ± 11%) and PUE (28 ± 8.1%) than the global average (Supplementary Fig. 4b). This improvement is driven by: 1) optimal average daily temperatures of 16 °C–18 °C for wheat growth, resulting in higher biomass and increased fertilizer demand[27,38]; 2) wheat cultivation relies heavily on rainfall for irrigation, with temperate regions benefiting from significant rainfall levels that support wheat growth and increase biomass, thus enhancing fertilizer demand and utilization efficiency. Rainfall also facilitates P mobility to root zones, increasing its availability[27,38].

Maize achieves the highest NUE (51 ± 14%) in tropical zones and PUE (42 ± 16%) in arid zones, with countries like Brazil surpassing global averages (Supplementary Fig. 4c). The mechanisms underpinning maize's elevated NUE in the tropics are multifaceted: 1) as a C4 plant, maize thrives under the intense light and high temperatures typical of tropical climates, facilitating rapid growth and development. This rapid growth escalates N demand for protein synthesis and other vital functions, thereby optimizing N utilization; 2) tropical regions generally offer longer growing seasons, affording maize ample time to absorb and utilize N, which in turn enhances NUE. Conversely, the mechanisms contributing to the highest PUE in arid zones include: 1) drought conditions can induce alterations in plant root architecture, promoting expansion into deeper soil strata to increase P uptake; 2) under such stress, plants may augment their capacity for inter-root acidification, releasing organic acids and other substances that improve P availability and efficacy in the soil[39].

Soybeans attain the highest NUE (51 ± 22%) in temperate regions and PUE (24 ± 2.3%) in arid regions, mirroring maize's spatial efficiencies, with the Indian and Ukraine standing out among major agricultural countries for exceeding global NUE and PUE averages (Supplementary Fig. 4d). The superior NUE of soybeans in temperate regions can be attributed to several factors: 1) The moderate temperatures, adequate precipitation, and extended growing season in temperate climates foster optimal growth conditions, enhancing both root development and N fixation[18]; 2); coupled with nutrient-rich soils and favorable microbial activity, support higher biological N fixation by *Rhizobium* species, reducing the need for synthetic fertilizers[40]. In arid regions, soybeans attain the highest PUE due to: 1) the minimal organic N deposition characteristic of these areas mitigates microbial competition for soil P during organic N decomposition, enhancing P availability for soybeans and improving PUE[27,41]; 2) drought conditions prompt soybean plants to develop deeper root systems, improving access to water and nutrients, including P, in subsoil layers; 3) under drought stress, soybeans may enhance rhizosphere acidification or upregulate the expression of P transport proteins in their root systems, thereby boosting PUE[40].

## Optimizing global crop distribution and management for better NUE and PUE

The current global scenario reveals that rice and maize predominantly exhibit NUE and PUE below 50%, whereas wheat and soybean NUE, not PUE, largely surpass this threshold, indicating substantial room for improvement, particularly in PUE (Fig. 4 and

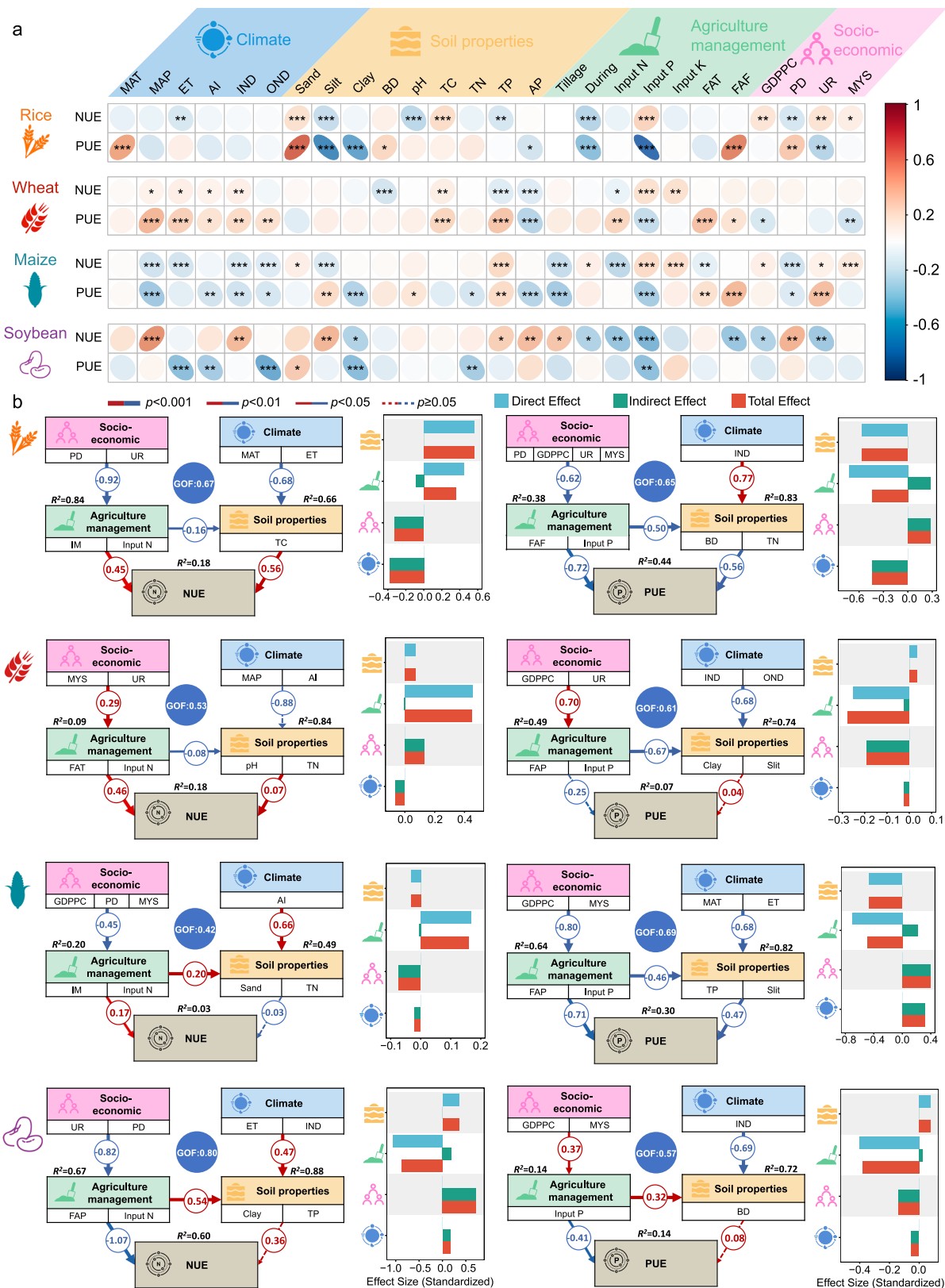

Supplementary Fig. 5). Although instances exist where rice and maize achieve over 50% NUE and PUE in parts of South Africa and Latin America, these figures primarily result from lower-than-average fertilizer inputs. For example, in Nigeria, rice's N and P input intensities stand at just 8.38 kg N ha⁻¹ and 0.85 kg P ha⁻¹, significantly below global averages of 95.3 kg N ha⁻¹ and 16.3 kg P ha⁻¹. Similarly, for maize,

input intensities in Brazil (68.5 kg N ha⁻¹ and 17.5 kg P ha⁻¹) and Nigeria (9.18 kg N ha⁻¹ and 0.93 kg P ha⁻¹) are significantly lower than the global averages of 112 kg N ha⁻¹ and 19.7 kg P ha⁻¹. With South Africa and Latin America also facing growing populations and escalating food demands, there's a looming risk that NUE and PUE might decline as fertilizer inputs increase.

**Fig. 3 | Global-scale correlations and structural equation model showing the relative importance of climate, soil properties, agriculture management, and socio-economic on nitrogen (NUE) and phosphorus use efficiency (PUE) of four major crops (rice, wheat, maize, soybean). a** Spearman's rank-order correlation between the NUE (PUE) of four major crops and environmental variables. All tests were two-sided. To control the false discovery rate in multiple comparisons, the resulting *p*-values were adjusted using the Benjamini-Hochberg method. Both the correlation coefficients and their corresponding adjusted exact p-values are annotated in the figure3a. *: *p* < 0.05; **: *p* < 0.01; ***: *p* < 0.001; ****: *p* < 0.0001. **b** Structural equation model between the NUE (PUE) of four major crops and environmental variables. Data are presented as path coefficients derived from bootstrap resampling (*n* = 100 iterations). The exact p-values are reported above the paths. The overall predictive performance of the

model is evaluated using the Goodness-of-Fit (GOF) statistic, with higher values indicating better prediction. The amount of variance explained by the model (*R²*) is also shown for each response variable. All abbreviations are as follows: MAT, mean annual temperature; MAP, mean annual precipitation; ET, evapotranspiration; AI, aridity index; IND, inorganic nitrogen deposition; OND, organic nitrogen deposition; Sand, sand content; Silt, silt content; Clay, clay content; BD, bulk density; TC, total carbon; TN, total nitrogen; TP, total phosphorus; AP, available phosphorus; Tillage, no-till or not; During, experimental period; Input N, nitrogen fertilizer input; Input P, phosphate fertilizer input; Input K, potash fertilizer input; FAT, fertilizer application types; FAF, fertilizer application frequency; FAP, fertilizer application placement; IM, irrigation method; GDPPC, gross domestic product per capita; PD, population density; UR, urbanization rate; MYS, mean years of schooling.

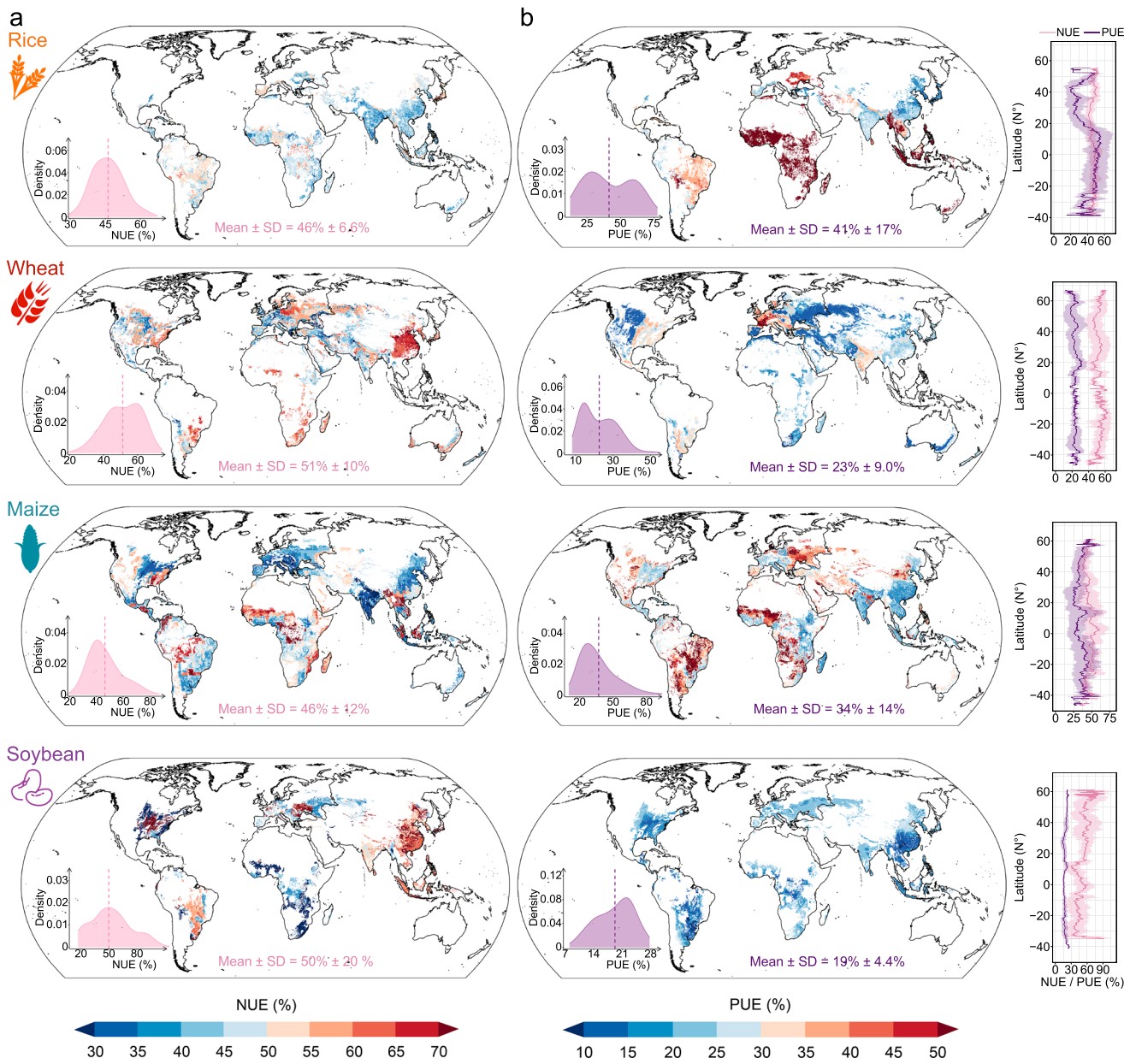

**Fig. 4 | Global spatial patterns of nitrogen (NUE) and phosphorus use efficiency (PUE) of four major crops (rice, wheat, maize, soybean) that were predicted based on random forest models. a,b** Mean refers to the mean value and SD refers to the standard deviation. The solid lines and shaded areas in the latitudinal pattern map represent the latitudinal mean and the latitudinal standard deviation, respectively.

Reevaluating global crop distribution patterns emerges as a viable strategy for enhancing NUE and PUE. Encouraging rice cultivation in the tropics and wheat cultivation in temperate zones, where efficiencies are notably higher than in other regions of the world (Fig. 4 and Supplementary Fig. 4), could significantly boost production efficiencies. However, as the 2008–2010 food crisis highlighted, countries might still want to produce their own food for reasons of food sovereignty, cultural preferences, and national security, even when efficiencies are lower compared to global averages. For example, in Ethiopia, wheat production is prioritized despite lower efficiencies, as local food production is crucial for national food security and cultural practices[42]. This can pose a challenge for adopting large-scale, region-specific crop distribution strategies that might require shifts in national agricultural priorities.

Practical challenges arise when implementing region-specific crop distribution. For instance, in countries like India, overdependence on irrigation for rice cultivation in water-stressed regions, such as Punjab and Haryana, has led to severe groundwater depletion[43]. Additionally, tropical regions, despite receiving higher rainfall, often lack the necessary irrigation infrastructure for optimal rice cultivation[44]. Soil conditions also pose significant challenges, with high P fixation in acidic tropical soils reducing PUE, while salinity and nutrient depletion limit the expansion of wheat cultivation in temperate zones[28,45]. Moreover, historical and economic factors, such as the profitability of rice cultivation under government procurement schemes, often make farmers reluctant to change cropping patterns. Expanding rice cultivation in tropical regions could exacerbate water stress, and expanding wheat in nutrient-depleted areas might increase reliance on chemical fertilizers. Imbalanced fertilizer use and limited access to advanced soil testing facilities further hinder improvements in NUE and PUE[9]. Despite these hurdles, optimizing crop distribution could still lead to significant environmental and food security benefits, provided these challenges are addressed through comprehensive, region-specific strategies.

Furthermore, yields per unit area of major food-producing countries in these regions are higher than in other regions of the world, promising a win-win situation in terms of both environmental mitigation and food security (Supplementary Fig. 5b). For example, rice yields averaged 7028 kg ha$^{-1}$ in China, 5203 kg ha$^{-1}$ in Indonesia, and 5818 kg ha$^{-1}$ in Vietnam, all of which exceed the global average of 4673 kg ha$^{-1}$. Similarly, wheat yields averaged 5242 kg ha$^{-1}$ in the European Union and 5416 kg ha$^{-1}$ in China, surpassing the global average of 3423 kg ha$^{-1}$. While these yield figures suggest a win-win situation in terms of both environmental mitigation and food security, the actual implementation in specific regions requires overcoming challenges related to irrigation infrastructure, soil health, and economic incentives for farmers. Therefore, a more nuanced, context-specific approach is essential for translating these general yield trends into actionable recommendations.

The inefficiencies in NUE and PUE in global crop production have resulted in significant nutrient surpluses, which have had profound impacts on the ecological environment (Fig. 5). Maize, in particular, registers the highest N (73 kg N ha$^{-1}$) and P (15 kg P ha$^{-1}$) surplus intensities among major crops. China and the United States, which are the major N and P surpluses in the maize ecosystem, contribute 56.7% of the N surplus load and 61.9% of the P surplus load due to higher (99.4–115 kg N ha$^{-1}$ and 16.8–33.7 kg P ha$^{-1}$) than the global average surplus intensities and large planted areas (4.21 × 10$^5$ km$^2$ for China and 3.29 × 10$^5$ km$^2$ for the United States). Furthermore, our structural equation model reveals that agricultural management practices are the most significant factor influencing NUE and PUE in maize, with tillage methods having the most pronounced impact (Fig. 3). Specifically, the adoption of no-till practices can concurrently improve NUE and PUE in maize, offering a practical approach to reduce the N and P surpluses in maize ecosystems.

## Looking forward

Our study investigates the dynamics of NUE and PUE for major crops globally from 1961 to 2018. The analysis reveals a significant temporal trend: while global fertilizer applications surged since the 1960s, NUE and PUE initially declined until approximately 1975. Since then, although some improvement in these efficiencies has been observed, they remain suboptimal for major crops, particularly in developing regions. The disparities between developed and developing countries highlight the need for targeted strategies that address specific regional challenges in nutrient management. Moreover, our findings identify potential hotspots for improvement, especially in regions where crop distribution aligns with optimal environmental conditions for nutrient uptake. We advocate for a reevaluation of global agricultural strategies that prioritize not only yield maximization but also the optimization of nutrient use, thereby ensuring food security while minimizing adverse ecological consequences. As we approach the second Green Revolution, it is essential to strike a balance between agricultural productivity and environmental stewardship. Collaborative efforts among policymakers, researchers, and farmers will be crucial to advancing practices that enhance NUE and PUE, thereby fostering a more sustainable and resilient global food system.

While this study provides important insights, several limitations warrant consideration. First, the reliance on national-level data may obscure regional differences in historical trends, particularly in countries with diverse climates and agricultural practices. While national datasets offer a broad overview, they may not adequately capture local variations that are essential for developing region-specific strategies. For example, in China, certain regions are transitioning from small-scale farming to more intensive agricultural practices—a shift that national-level data may overlook[46]. Establishing a global collaborative framework to systematically collect and analyze regional agricultural policies would enable more fine-grained assessments in future research. Second, this study primarily focuses on the analysis of crop NUE and PUE in the historical and present contexts. However, we also observe that climate change can influence NUE and PUE. For instance, under the 'fossil-fueled development-high forcing' scenario, NUE in rice is expected to decrease, while PUE is projected to increase (Supplementary Fig. 6). This may be attributed to the elevated temperatures predicted under this scenario, which significantly impair root nitrogen uptake capacity, thereby reducing NUE[47]. Additionally, in high-emission scenarios, increased carbon dioxide concentrations can stimulate the exudation of organic compounds and organic acids from roots. These exudates promote the mobilization of soil phosphorus and stimulate microbial activity, thereby improving the plant's phosphorus uptake efficiency and contributing to the observed increase in PUE[48]. These findings, based on external scenario modeling, suggest that NUE and PUE of crops may continue to fluctuate under future climate change conditions. By incorporating future changes in soil properties and socioeconomic factors, we can adopt optimal agricultural management practices more effectively to improve crop NUE and PUE. Future research should integrate high-resolution, region-specific data and adopt cross-disciplinary approaches to refine NUE and PUE estimations. Experimental studies tailored to diverse cropping systems and specific regional contexts could offer actionable insights into optimizing nutrient efficiency, supporting scalable improvements in agricultural sustainability.

## Methods

### Global NUE and PUE at sampling points

**Database compile.** To quantitatively evaluate global trends and drivers of the NUE and PUE of major crops (rice, wheat, maize, and soybean), we undertook a comprehensive data collection initiative (Supplementary Fig. 7) by a systematic literature review of peer-reviewed articles obtained from the Web of Science (https://www.web of science.com/), Google Scholar (https://scholar.google.com/), and

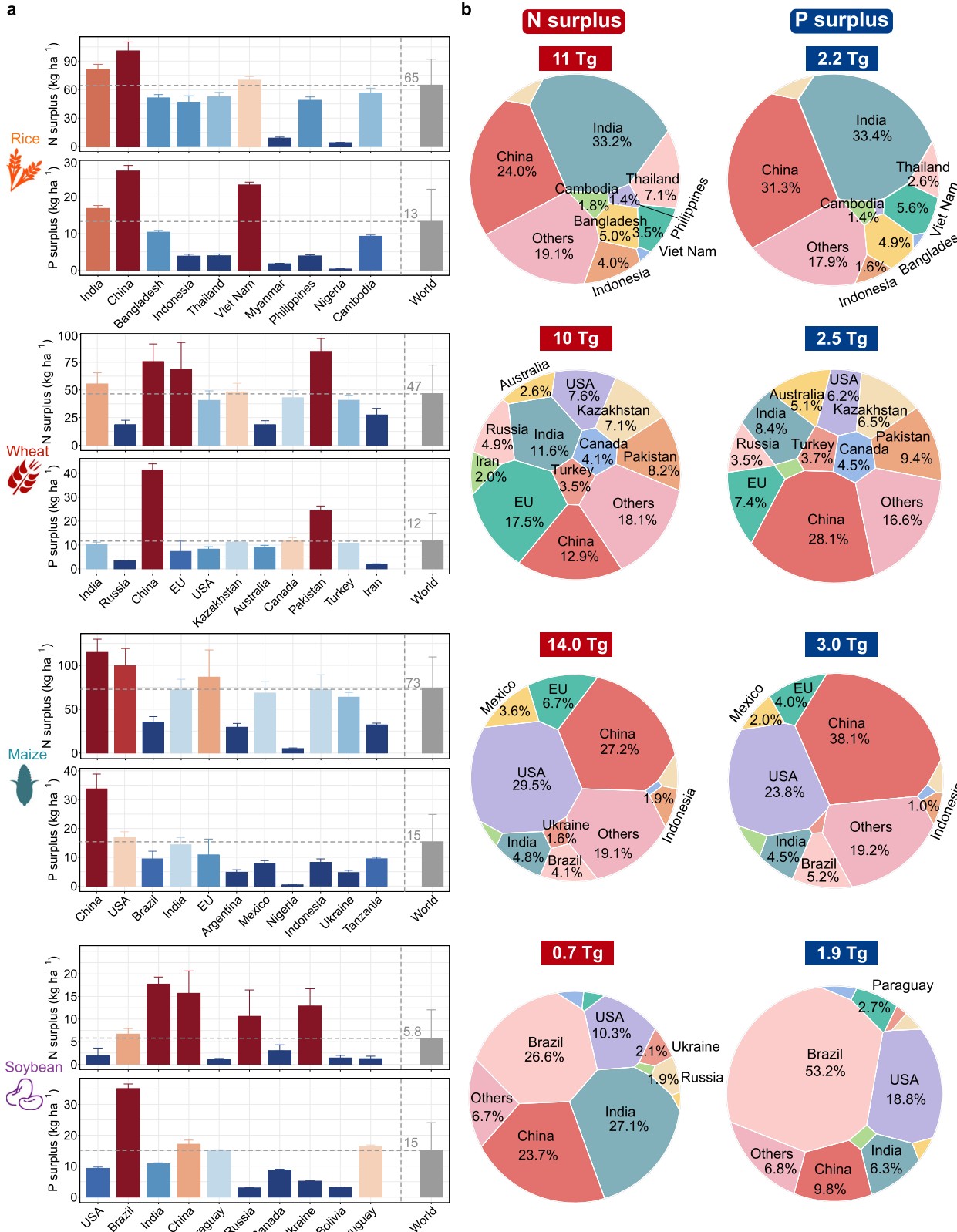

**Fig. 5 | Global nitrogen (N) and phosphorus (P) fertilizer surplus across four major crops (rice, wheat, maize and soybean). a,b** The intensity and load of N and P surplus in the ten countries with the largest global crop cultivation area across four major crops. Ranking among countries is based on the area planted to specific crops. The data are presented as mean ± sd (error bars). Each bar summarizes the statistical distribution characteristics across different countries and regions based on a global predictive raster dataset (*n* = 748,604).

China National Knowledge Infrastructure (https://oversea.cnki.net/index/), up to April 2023 (Supplementary Fig. 7). Articles included in the statistical analysis met the following screening criteria: (1) provided explicit records of NUE and PUE or crop uptake of these nutrients. Both NUE and PUE were calculated based on $N(P)UE_{diff}$; (2) spanned at least one complete crop cycle; (3) specify geographical locations; (4) control (no fertilizer) and experimental treatments (fertilizer applied) are executed in the field conditions; and (5) the information on crop species, cropping system, and fertilizer inputs are reported. Since the actual fertilizer use efficiency will not be less than 0, we exclude these outliers. Based on these criteria, our final dataset comprised 2919 paired independent observations from 173 global studies, including 2354 N data and 1006 P data, with sampling years spanning ranging from 1982 to 2021 (Fig. 1).

**Compilation of NUE and PUE data.** Data extracted from the selected studies included: (1) experimental site details, such as latitude, longitude, elevation and the country; (2) crop types; (3) climate conditions, detailed through mean annual precipitation (MAP), mean annual air temperature (MAT), mean annual evapotranspiration (ET), and the aridity index (AI); (4) initial soil physical properties including texture (proportions of sand, silt, and clay) and bulk density (BD), along with soil chemical properties like total carbon (C), nitrogen (N), phosphorus (P), available phosphorus (AP) contents and pH level; (5) the nature of tillage practices, categorized as either conservation or conventional based on descriptions within the articles; (6) the period of the experiment, indicating the start year and duration; (7) inputs of N, P, and potassium (K) fertilizers; (8) fertilizers application types (FAT) including organic fertilizers, inorganic fertilizers, and the combined application of organic and inorganic fertilizers; (9)fertilizers application placement (FAP) including surface application, deep application, mixed application, and foliar spraying; (10) fertilizer application frequency (FAF); (11) irrigation methods (IM) including permanent flooding, intermittent irrigation, drip irrigation, and no irrigation; (12) the period of the experiment, indicating the start year and duration; (13) the number of experimental replicates conducted; and (14) the NUE and PUE values alongside the N and P uptake by crops. Preference was given to using nutrient uptake data to calculate the efficiencies when both nutrient uptake data and NUE and PUE values were provided. Data were extracted directly from tables and textual descriptions provided in the selected studies. For data presented in graphical form, we employed GetData Graph Digitize software (version 2.25.0.32) to ensure accuracy in data retrieval.

For calculating N and P uptake, both grain and residue yields were considered, acknowledging residue as a potential resource. The uptake efficiency based on $N(P)UE_{diff}$ was calculated using the formula:

$$NUE(PUE) = \frac{U_{N(P)} - U_{N_0(P_0)}}{Input\ N(P)} \times 100\% \tag{1}$$

where $U_{N(P)}$ (kg ha$^{-1}$) represents the N and P uptake by mature crops from both soil and fertilizers, $U_{N_0(P_0)}$ (kg ha$^{-1}$) denotes the uptake under no fertilizer conditions and $Input\ N(P)$ is the amount of fertilizer applied (including the organic fertilizer and inorganic fertilizer, kg ha$^{-1}$).

If the article lacks direct information on both $U_{N(P)}$ and NUE(PUE), but presents data on aboveground biomass and N(P) elemental content, we employ the following formula to compute the $U_{N(P)}$:

$$U_{N(P)} = G \times N(P)_G + S \times N(P)_S \tag{2}$$

$$U_{N_0(P_0)} = G_0 \times N(P)_{G_0} + S_0 \times N(P)_{S_0} \tag{3}$$

where $G$ (kg ha$^{-1}$) represents the mass of grains under fertilization treatment, $G_O$ (kg ha$^{-1}$) represents the mass of grains without fertilization treatment, $S$ (kg ha$^{-1}$) represents the mass of residue under fertilization treatment, and $S_O$ (kg ha$^{-1}$) represents the mass of residue without fertilization treatment. $N(P)_G$, $N(P)_S$, $N(P)_{G_0}$ and $N(P)_{S_0}$ (% of dry weight) represent the nitrogen and phosphorus content of grains and residue under fertilized and unfertilized treatments. All these calculations were done at a yearly time scale.

**Imputation of missing values in environment variable.** In cases where MAT, MAP, potential evapotranspiration (PET) and ET data were not explicitly provided, we utilized historical climate data spanning 2000–2020 from the CRU TS[49] and GLEAM v3[50]. This approach leverages the relative stability of climatic conditions within specific regions to infer missing data, utilizing the raster R package for spatial analysis.

To calculate the AI, a critical factor in assessing environmental conditions impacting fertilizer use efficiency, we implemented the following formula[51]:

$$AI = \frac{MAT}{PET} \tag{4}$$

For locations where soil chemical and physical property data were missing, we compared HSWD[52], SoilGrids[53], and the Global Soil Dataset for Earth System Modeling[54]. The Global Soil Dataset for Earth System Modeling contains all the soil environmental covariates we need, so we used this dataset for interpolation and prediction to maintain consistency of soil properties. Furthermore, to evaluate the influence of N deposition on fertilizer use efficiency, we integrated organic N and inorganic N deposition data[55]. This additional step enabled a more nuanced understanding of the environmental factors contributing to nutrient utilization by crops, particularly the role of atmospheric N deposition at our sampling locations.

In addition, we also incorporated socio-economic factors into the analysis of the impacts on the NUE and PUE of major crops. Socio-economic factors including gross domestic product per capita (GDPPC), population density (PD), urbanization rate (UR) and mean years of schooling (MYS). Due to the lack of reports on relevant population and economic aspects in the literature, we obtained information at the national level from the World Bank database and Human Development Data.

### Historical global NUE and PUE calculation

**Historical global fertilizer and production data collection.** To elucidate the temporal trends of NUE and PUE among major crops globally since 1961, we undertook a comprehensive data compilation across three distinct categories.

(1) Harvested area and production quantities: data sourced from Food and Agriculture Organization of the United Nations (https://www.fao.org/home/en) (FAOSTAT). We aggregated data on the harvested area, production quantity, and N and P fertilizer inputs for each crop across 205 countries (Supplementary Fig. 1). These datasets were used for the analysis of temporal trends in crop fertilizer inputs, detailed in the subsequent sections on **Calculations of NUE and PUE.**

(2) Fertilizer input intensities: data from International Fertilizer Association (https://www.fertilizer.org/) (IFA) reports were critical in assessing the intensities of N and P fertilizer applications across different countries, crops, and years[19,56]. This allowed for the refinement of FAOSTAT data by filling in gaps and ensuring a comprehensive dataset for our analyses.

(3) Crop nutrient content and residue-grain ratio: we incorporated global mean data on crop N and P content[1,2], along with residue-grain ratios[57], to convert FAOSTAT-reported grain yields into estimates of whole plant biomass (Supplementary Table 2).

**Predicted nutrient uptake by crops from soil.** To develop predictive models of nutrient uptake by crops from soil across different crop types (Supplementary Fig. 8), we first calculated the proportion of nitrogen (*nitrogen from soil*, %) and phosphorus (*phosphorus from soil, %*) uptake from soil, based on data (including Input N(P), crop N(P) uptake and NUE(PUE)) originated from a systematic review of peer-reviewed publications (Supplementary Fig. 7). The calculated formula as follows:

$$Nitrogen(Phosphorus)from\,soil = \frac{crop\,N(P)\,uptake - NUE(PUE) \times Input\,N(P)}{crop\,N(P)\,uptake} \times 100\%$$

(5)

Then, we utilized above-calculated nitrogen(phosphorus) from soil, climate (MAT, MAP), soil properties (BD, sand, silt, clay, TC, TN, TP, AP, pH) and agriculture management (Input N, Input P) factors to train random forest model across different crops (Supplementary Fig. 8). These environmental variables were sourced from a systematic review of peer-reviewed studies.

The dataset was partitioned into 80% for training and 20% for testing, sampled randomly. Parameter tuning was conducted via grid search to identify the combination that minimized root mean square error (RMSE). Each final model was built using a 10-fold cross-validation approach to ensure robustness and predictive accuracy.

Finally, the trained RF models (Supplementary Fig. 8) were employed to predict annual nitrogen and phosphorus uptake from soil for the four major crops. Climate data (MAT, MAP) were obtained from CRU TS[49], while agriculture management data (Input N, Input P) factors were sourced from FAOSTAT (Supplementary Table 3). Both datasets exhibit interannual variability. Since soil properties are relatively stable over time and no globally comprehensive datasets capture their interannual variations, temporal changes in soil properties were not incorporated into the model (Supplementary Table 3).

**Calculations of NUE and PUE.** The analytical framework for the calculation of NUE and PUE across major crops, including their cumulative assessment, involved a detailed multi-step process based on data from FAOSTAT and IFA. This process comprised four principal steps:

(1) Selection of baseline years:

We selected IFA reports from 1990, 1999, and 2018 as baseline years to anchor our analysis. To address data gaps within these reports, we employed a data supplementation strategy, prioritizing (i) data extraction from adjacent years; (ii) regional data correlation; and (iii) supplementation using global averages.

(2) Adjustment of Fertilizer Input Intensities:

To derive fertilizer input intensity data for different years, countries, and crops globally, we integrated FAOSTAT and IFA data to adjust fertilizer input intensities for baseline year as follows[2,3]:

$$Input\,N_{co,cr,yr} = Input\,N\_IFA_{co,cr} \times \frac{TN\_FAO_{input,yr}}{\sum_{co}\sum_{cr}\left(Input\,N\_IFA_{co,cr} \times A\_FAO_{co,cr,yr}\right)}$$

(6)

$$Input\,P_{co,cr,yr} = Input\,P\_IFA_{co,cr} \times \frac{TP\_FAO_{input,yr}}{\sum_{co}\sum_{cr}\left(Input\,P\_IFA_{co,cr} \times A\_FAO_{co,cr,yr}\right)}$$

(7)

where *Input N_IFA_{co,cr}* is the cropland N fertilizer input intensities (kg ha$^{-1}$) across different countries (co), crops (cr) based on IFA reports for the baseline years 1990, 1999, 2018; *TN_FAO_{input,yr}* is the total N fertilizer input (t) across different years (yr) from FAOSTAT; *A_FAO_{co,cr,yr}* is the harvested area (*ha*) across different countries, crops, years from FAOSTAT. *Input N_{co,cr,yr}* is the corrected N fertilizer input intensities (kg ha$^{-1}$) across different countries, crops, years. The formula (7) serves

as a correction for the P fertilizer input intensities, with each parameter referenced from the formula (6).

(3) Temporal trend estimation

Using the corrected fertilizer input intensities from 1961 to 2018 (Fig. 2a and Fig. 2b), we derived temporal trends for fertilizer use. The calculations were based on the baseline years 1990, 1999, and 2018, as data availability in IFA is limited for other years, restricting effective regional assessments of fertilizer intensity. Specifically, the period divisions were: 1961–1994, 1995–2006, and 2007–2018, corresponding to the respective baseline years.

(4) Calculation of NUE and PUE (1961–2018):

We calculated NUE (%) and PUE (%) over this period (Fig. 2c) based on the potential relationship ($P < 0.05$) between NUE(PUE) and N(P) fertilizer input intensities (Supplementary Fig. 9). These were formulated as:

$$F_{N(P)} = \sum_{co}(Input\,N(P)_{co,cr,yr} \times A_{FAO\,co,cr,yr})$$

(8)

$$U_{N(P)} = \frac{Q \times N(P)_G + Q \times R \times N(P)_S}{100}$$

(9)

$$U_{N_0(P_0)} = \frac{U_{N(P)} \times Nitrogen(Phosphorus)from\,soil}{100}$$

(10)

$$NUE(PUE) = \frac{U_{N(P)} - U_{N_0(P_0)}}{F_{N(P)}} \times 100\%$$

(11)

where the parameters are production quantity (Q, t), the nitrogen ($F_N$, t) and phosphorus ($F_P$, t) inorganic fertilizer inputs for each crop (Supplementary Fig. 1), residue-grain ratio (R), nitrogen ($N_G$, %; $N_S$, %) and phosphorus ($P_G$, %; $P_S$, %) content in grain and residue (Supplementary Table 2), nitrogen from soil (%), phosphorus from soil (%), respectively; $U_{N(P)}$ (t) are the N and P uptake by mature crops from the soil and fertilizers; $U_{N_0(P_0)}$ (t) are the N and P uptake by mature crops from only soil under the condition of no fertilization.

**The temporal trend of NUE and PUE.** For representing the temporal trend of fertilizer input intensities (Fig. 2a and Fig. 2b), NUE (Fig. 2c and Fig. 2d), PUE (Fig. 2c and Fig. 2d) for global major and overall crops from 1961 to 2018, we used the "segmented" software package[58] in R to calculate breakpoints in the regression model. The principle of the calculation involves dividing the dataset into two segments by introducing the median of the interval of the dependent variable as the initial breakpoint, allowing each segment to have a different slope. By iteratively minimizing the residual sum of squares, the position of the breakpoint is refined until convergence is achieved[59,60]. We then conducted linear regressions for the periods before and after the breakpoint. Then, we examined whether there was a significant difference between two adjacent groups before and after the breakpoint across different crops using the Wilcoxon method[61] (Fig. 2d and Supplementary Fig. 3 and Supplementary Fig. 10). Notably, we selected the eight regions with the largest crop cultivation areas as representatives of developed countries (the United States and the European Union) and developing countries (China, India, Brazil, Indonesia, Russia, and Nigeria) (Fig. 2d and Supplementary Fig. 3). This approach aims to minimize uncertainties in fertilizer input intensity and grain yield data associated with smaller nations.

The "ggplot2" package was used to visualize the temporal trends of raw data (Supplementary Fig. 1), as well as the fertilizer input intensities for major countries with larger harvested areas of overall crops over time (Supplementary Fig. 3a). The gray dashed lines indicated the breakpoint which derived from segmented fitting (Fig. 2). To represent global fertilizer input shares in major countries with the larger harvested area, we used the "voronoiTreemap" package in R to

design pie figures base on the 2018 (Fig. 2a and Fig. 2b). European Union (EU) participation is based on January 2024 including 27 countries, namely Austria, Belgium, Bulgaria, Cyprus, Czech Republic, Croatia, Denmark, Estonia, Finland, France, Germany, Greece, Hungary, Ireland, Italy, Latvia, Lithuania, Luxembourg, Malta, the Netherlands, Poland, Portugal, Romania, Slovakia, Slovenia, Spain, Sweden.

## Mapping the spatial variation of global NUE and PUE

**Spearman correlation analysis.** Spearman pearson correlation analyses was used to test for relationships between NUE (or PUE) and the factors including climate (MAT, MAP, ET, AI, IND, OND), soil properties (Sand, Silt, Clay, BD, pH, TC, TN, TP, AP), agriculture management (Tillage, During, Input N, Input P, Input K, FAT, FAF), society and economy (GDPPC, PD, UR, MYS) by using the "corrplot" R package[62] (Fig. 3a).

**Partial least squares path modeling.** To investigate the path relationships between environmental, agricultural management, and socio-economic factors and their effects on NUE (or PUE), we employed Partial Least Squares Path Modeling (PLS-PM)[63] to construct a path model aimed at identifying the key driving factors (Fig. 3b). In the modeling process, climate, soil properties, agricultural management practices, and socio-economic factors were treated as primary latent variables. The sub-variables contained within each category were considered secondary manifest variables. To mitigate collinearity issues, secondary variables with loadings below 0.7 were excluded from the model. The goodness-of-fit (GOF) index was used as a criterion to evaluate model performance, with values closer to 1 indicating a better fit. By maximizing the GOF, we selected representative secondary variables and constructed the final path model[64].

**Random forest modeling.** We selected variables that showed significant correlations with NUE (or PUE) and the amount of N (or P) fertilizer inputs as factors. Random forest models were primarily employed to assess variable importance and influence using the "randomForest" R package[65]. Unlike traditional regression models, random forests are unaffected by correlations among predictors, which do not compromise model accuracy[66].

We randomly sampled 80% of the dataset for model training, with the remaining 20% allocated for testing. Each final model was developed through 10-fold cross-validation. Parameter tuning was performed via grid search to identify the combination of nodesize (minimum number of samples required in a leaf node; when the number of samples in a node is less than or equal to nodesize, the node no longer splits and is considered a leaf), ntree (total number of decision trees in the forest; each trained independently, with the final prediction obtained by averaging their outputs), and mtry (number of features randomly selected at each split when constructing each tree) that minimized the root mean squared error (RMSE), thus ensuring model robustness.

Furthermore, contour plots were used to illustrate the influence of different parameter combinations on model performance (Supplementary Fig. 11). Model accuracy was evaluated using the coefficient of determination ($R^2$) and the *P*-value between predictions and observations (Supplementary Fig. 12).

**Global NUE and PUE predictions.** Trained RF models were used to predict global NUE and PUE for the four major crops. We established a global 5-arcmin resolution grid within the planting areas of the four major crops. A series of globally gridded datasets (Supplementary Table 3) were used for model calculations. The fertilizer input data for each crop were sourced from IFA (2018). We set the prediction timeframe for planting to be one year. This choice was based on two main reasons. Firstly, most of the originally collected NUE and PUE were calculated annually, ensuring more accurate predictions. Secondly, a

one-year planting cycle better reflected the efficiency of current fertilizer inputs, enhancing the relevance of the model to agricultural practices.

The random forest model performed multiple predictions for each grid cell according to the number of trees. We derived the final predicted value for each grid cell by averaging these multiple predictions, with the standard deviation of these predictions indicating the uncertainty associated with the model (Supplementary Fig. 13). Then, the global model predictions of NUE and PUE were spatially corrected through Kriging interpolation based on NUE and PUE residuals (observed minus predicted values) using the 'gstat' package in R. This approach better captures the geographical variability of nutrient use efficiency while providing more robust uncertainty estimates for spatial predictions[67,68], with the RMSE between predicted and observed values of NUE and PUE decreasing by 0.09 and 0.13 after applying Kriging interpolation for correction, respectively. Additionally, we visualized the Kriging-adjusted NUE and PUE prediction of the four major crops on a world map and illustrated the differences between latitudes using line graphs. The overall distribution range and probability density of the data are displayed using the "hrbrthemes" package in the bottom left corner of the map (Fig. 4).

We also investigated the impact of environment and agricultural management in different climatic zones and major agricultural countries. We utilized the Köppen climate classification (Supplementary Fig. 14) and data from the world's major grain-producing countries. The harvested area ranks among the top ten countries worldwide. Additionally, we included the EU in the comparison of major crop-planting countries. We conducted pairwise comparisons of fertilizer use efficiency among different climate zones or countries and regions using the Dunn's Kruskal-Wallis method implemented in the "multcompView" package[69]. Significant differences ($P < 0.05$) were denoted by different letters (Supplementary Fig. 4).

**Assessment of nutrient use efficiency.** To assess the relative size of fertilizer utilization and surpluses, we divided fertilizer use efficiency into four categories based on a threshold of 50%; low NUE and low PUE (NUE < 50%, PUE < 50%); high NUE and low PUE (NUE > 50%, PUE < 50%); high NUE and high PUE (NUE > 50%, PUE > 50%) and low NUE and high PUE (NUE < 50%, PUE > 50%). Each independent prediction in the grid cell was assigned to one of these four categories, and the pie chart represented the proportion of global crop areas where NUE and PUE fall within each of these four categories (Supplementary Fig. 5a). The area proportions of the four categories can be calculated using the following formula:

$$Area\ proportion = \frac{\sum_{i=c_1}^{c_n} Area_i}{\sum_{i=1}^{m} Area_i} \tag{12}$$

where c represents the fertilizer use efficiency category, $c_1$-$c_n$ represents all grid cells belonging to that category, m represents the total number of grid cells globally for that crop, and the planting area in each grid cell comes from SPAM 2010 V2r0[70]. At the same time, we calculated the average NUE and PUE for the main cultivating countries of the four crops. We used NUE as the x-axis and PUE as the y-axis, dividing both axes into four quadrants with 50% as the boundary. Based on the position of each country's NUE and PUE within these quadrants, we depicted the regional differences in global NUE and PUE (Supplementary Fig. 5b).

Exploring the nitrogen-phosphorus asynchrony worldwide, we visualized the predicted nitrogen-phosphorus utilization efficiency ratio (NUE:PUE) on a global map (Supplementary Fig. 15). The "multcompView" package was utilized for testing the differences in NUE:PUE among different countries or regions[69].

**Assessment of fertilizer surplus.** While evaluating fertilizer use efficiency, we also calculated fertilizer surplus for the main cultivating countries globally. The formulas for calculating the N(P) surplus loads, intensities, and standard deviation for each country are as follows:

$$\text{N(P) surplus loads} = \sum_{i=e_1}^{e_n} Area_i \times input\,N(P)_i \times (1 - NUE_i(PUE_i)) \tag{13}$$

$$\text{N(P) surplus intensities} = \frac{\sum_{i=e_1}^{e_n} Area_i \times input\,N(P)_i \times (1 - NUE_i(PUE_i))}{\sum_{i=e_1}^{e_n} Area_i} \tag{14}$$

$$sd_{\text{N(P) surplus intensities}} = \sqrt{\frac{\sum_{i=e_1}^{e_n} (input\,N(P)_i \times (1 - NUE_i(PUE_i)) - \text{N(P) surplus intensities})^2}{n-1}} \tag{15}$$

where $e$ represents a country or region, $e_1$-$e_n$ represents all grid cells belonging to that country or region. We compared the average nitrogen fertilizer surplus intensities for each country with the global average level, presenting the total surplus as a percentage (Fig. 5). Additionally, we compared the fertilizer surplus per unit yield to better assess the dual effects of food security and fertilizer pollution in different regions (Supplementary Fig. 16).

**Changes in NUE and PUE under future climate scenarios.** To assess the changes in NUE and PUE of rice, wheat, maize, and soybeans under future climate conditions, we utilized the widely used climate model simulation data from the Coupled Model Intercomparison Project Phase 6 (CMIP6). Specifically, the data were sourced from six climate models: ACCESS-CM2 (Australian Community Climate and Earth System Simulator Coupled Model Version 2), BCC-CSM2-MR (Beijing Climate Center Climate System Model Medium Resolution), CanESM5 (Canadian Earth System Model Version 5), CESM2-WACCM (Community Earth System Model Version 2 with Whole Atmosphere Community Climate Model), FIO-ESM-2-0 (First Institute of Oceanography Earth System Model Version 2.0), and MRI-ESM-2-0 (Meteorological Research Institute Earth System Model Version 2.0)[71]. We integrated these six datasets from different institutions with matching spatial and temporal resolutions to reduce the uncertainties in future climate projections caused by different model algorithms.

To investigate the changes in crop NUE and PUE under varying warming scenarios, three Shared Socioeconomic Pathways (SSP) scenarios were established: SSP126 (sustainable development pathway: low-carbon, equitable society with rapid climate mitigation), SSP245 (intermediate pathway: moderate development under continued current policies with partial climate action), SSP585 (fossil fuel-driven pathway: high emissions, energy-intensive growth with minimal sustainability efforts)[72]. This framework enables a systematic analysis of how nutrient use efficiency in agricultural systems is influenced by high, medium, and low carbon emission, as well as different levels of climate warming (radiative forcing targets of 2.6, 4.5, and 8.5 W/m² by 2100)[72] (Supplementary Fig. 6). The above statistical analyses were performed using R 4.2.2 (R Development Core Team, 2020).

**Reporting summary**
Further information on research design is available in the Nature Portfolio Reporting Summary linked to this article.

## Data availability
All data used in this study is available at Figshare [https://doi.org/10.6084/m9.figshare.25998922][73].

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

## Acknowledgements

Ji Liu is funded by the National Natural Science Foundation of China (42207107; 42577108), the Horizon Europe Framework Program (101205485), Shaanxi Provincial Youth Science and Technology Star Project (2025ZC-KJXX-130). Wenfeng Tan is funded by the National Key Research and Development Program of China (2021YFD1901205). Zhi Quan is funded by the Youth Innovation Promotion Association CAS (2021195). Josep Penuelas is funded by the Spanish Government grants (PID2020115770RB-I, PID2022-140808NB-I00), and TED2021-132627 B–I00 funded by MCIN, AEI/10.13039/ 501100011033 European Union Next Generation EU/PRTR.

## Author contributions

J.L. conceived the project, designed the methodology, performed investigations, prepared visualizations, and wrote the original draft. H.W. and J.M. contributed to methodology, investigation, and visualizations. Z.H., Y.Y., J.L., and Y.Z. performed investigations. L.F., W.T., and J.P. supervised the research. J.L., H.W., J.M., J.P., M.D.-B., J.S., F.C., Z.Q., T.Q., Y.L., Y.G., Z.H., Y.Y., J.L., Y.Z., W.T., G.Z., L.L., and L.F. reviewed and edited the manuscript. All authors discussed the results and approved the final version of the manuscript.

## Competing interests

The authors declare no competing interests.
