## [Peer Review file · Nature Communications]

Global-scale prevalence of low nutrient use efficiency across major crops

Corresponding Author: Professor Linchuan Fang

Version 0:

Reviewer comments:

Reviewer #1

(Remarks to the Author)

The paper tackles the critical global challenge of low nitrogen and phosphorus use efficiency (NUE and PUE), emphasizing their profound implications for sustainable agriculture and environmental protection. By compiling an extensive global database of 3,360 observations spanning 205 countries, the authors provide a robust foundation for their findings. They also utilized advanced statistical tools, like random forest, to analyze and predict patterns of NUE and PUE, adding depth to their analysis. The differentiation of these efficiencies across crops, regions, and climates is meticulously examined, alongside an insightful exploration of temporal trends, such as the decline in NUE and PUE during the early Green Revolution and their gradual recovery since the mid-1970s. The inclusion of biogeographical and environmental factors further enriches the contextual analysis. Notably, the study offers practical, actionable strategies—such as optimizing crop distribution and adopting precision agriculture techniques—that hold significant potential for enhancing nutrient use efficiency on a global scale. Nevertheless, I have some concerns that must be addressed. Therefore, I recommend a major revision.

My concern #1: I understand that the authors acknowledge the limitations of national-level data and mention that the lack of region-specific, high-resolution data might lead to generalized conclusions that obscure local variations (L360-361). Therefore, I suggest discussing how future iterations of the study could benefit from incorporating regional datasets or experimental validations. Additionally, could you include at least a few regional examples or case studies to make the findings more relatable and actionable at the local scale?

My concern #2: The study focuses heavily on environmental and agricultural factors, but the role of socioeconomic influences (e.g., farmer behavior, government policies) in determining NUE and PUE appears underexplored. The authors have briefly mentioned the Green Revolution and its impact on government policies (L165-204 and elsewhere). I suggest expanding on these aspects to enrich the analysis and enhance the relevance of the recommendations. For example, consider integrating data on farmer practices, economic incentives, and regulatory frameworks to provide a more holistic view of the factors influencing nutrient use efficiency.

My concern #3: L210-213: These suggestions present valuable insights for driving a second Green Revolution. However, without providing economic incentives to farmers, implementing these recommendations comprehensively—particularly in developing nations—may prove challenging.

My concern #4: L232-234: “This trend... preserves toward saturation.” I do not understand this line. Are you referring to ‘P reserves’ or ‘preserves’? Please clarify.

My concern #5: Many conclusions are drawn from data spanning 1961–2018, with limited emphasis on recent trends or future projections. I recommend addressing this gap through scenario modeling or projections beyond 2020. Additionally, I suggest emphasizing how the findings can be translated into actionable recommendations for policymakers, particularly in developing nations.

My concern #6: The authors have discussed NUE extensively. Nonetheless, I find the discussions of PUE to be less thorough. A more balanced discussion is needed, especially considering the critical importance of phosphorus as a non-renewable resource. For example, in L234-245, the authors propose some good solutions; however, they overlook a critical factor: the role of soil organic matter. Organic matter plays a vital role in enhancing phosphorus availability by releasing

organic acids that solubilize bound phosphorus. Unfortunately, many soils worldwide are characterized by low organic carbon content, which diminishes this beneficial effect. Improving PUE also relies on adopting integrated nutrient management strategies, optimizing P fertilizer placement techniques, and enhancing irrigation practices. Continuous cultivation of high phosphorus-demanding crops, coupled with minimal fallow periods, significantly depletes the available phosphorus pool. Additionally, many farmers in developing countries lack access to soil testing facilities or fail to follow recommendations based on soil tests, resulting in inefficient phosphorus use. These critical challenges warrant thorough discussion.

My concern #7: L312-314: The authors propose encouraging rice cultivation in tropical regions and wheat cultivation in temperate zones; however, this recommendation faces several practical challenges. For example, in India, implementing region-specific crop distribution to enhance NUE and PUE faces several obstacles. Overdependence on irrigation for rice cultivation in water-stressed regions like Punjab and Haryana has led to severe groundwater depletion, while tropical areas, despite higher rainfall, often lack the necessary irrigation infrastructure. Soil conditions also pose hurdles, with high phosphorus fixation in acidic tropical soils reducing PUE and salinity or nutrient depletion limiting wheat expansion in temperate zones. Additionally, historical and economic factors, such as the profitability of rice cultivation under government procurement schemes, make farmers reluctant to change cropping patterns. Environmental concerns also arise, as cultivating rice in tropical regions can exacerbate water stress, while expanding wheat in nutrient-depleted areas may increase reliance on chemical inputs. Moreover, imbalanced fertilizer use and limited access to advanced soil testing facilities further hinder NUE and PUE improvements. Addressing these challenges requires robust policy support, infrastructure development, and farmer education to promote sustainable and efficient cropping practices. Thus, I believe the authors should expand the discussion to incorporate multiple perspectives.

My concern #8: L554-569: The authors used RF to create global NUE and PUE maps. I recommend using digital soil mapping (DSM) (McBratney et al., 2003) for this. DSM-based Regression Kriging (RK) instead of relying solely on the RF model for predicting global NUE and PUE could offer several advantages. A simple RF model lacks a spatial component to account for geographic variation. Regression Kriging, on the other hand, combines a regression model (e.g., RF, Cubist, or any other model) with spatial interpolation (kriging), allowing it to incorporate both environmental covariates and spatial autocorrelation explicitly. This makes RK particularly suited for capturing the spatial heterogeneity of NUE and PUE across different regions and climates. Since this is a global study and accurately delineating spatial patterns is essential, RK would enhance prediction accuracy by using both the regression relationships between soil properties, climate, and NUE/PUE, and the spatial dependence of these variables. Additionally, RK provides more robust uncertainty estimates for spatial predictions, which is crucial for guiding targeted interventions. Thus, I recommend integrating RK into DSM to address the spatial complexities inherent in global nutrient use efficiency mapping.

McBratney, A. B., Santos, M. M., & Minasny, B. (2003). On digital soil mapping. *Geoderma*, 117(1-2), 3-52.

Reviewer #2

(Remarks to the Author)

The manuscript titled "Global-scale prevalence of low nutrient use efficiency across major crops" addresses a significant issue in sustainable agriculture by examining nutrient use efficiency (NUE and PUE) on a global scale. However, several critical weaknesses and areas for improvement warrant serious consideration before publication.

General comments:

-The study lacks novelty, as it does not present new methodologies or significant findings compared to existing literature. While it compiles a comprehensive dataset, the insights drawn are largely reiterative of previously established concepts regarding nutrient inefficiency. Specifically, the research does not advance the current understanding of Nutrient Use Efficiency (NUE) and Phosphorus Use Efficiency (PUE) in major crops, relying heavily on existing literature and established trends without offering substantial new data or innovative perspectives that would justify publication in a high-impact journal such as *Nature Communications*. The study primarily reiterates well-established knowledge about global nutrient use efficiency trends, such as the low NUE and PUE in developing regions and the disparities between developed and developing countries. These findings are not novel and have been extensively documented in prior literature, including works by Zhang et al. (2015) and Zou et al. (2022) cited in this paper.

-While the authors mention gaps in understanding NUE and PUE dynamics, they fail to articulate a compelling rationale for why their specific research approach and dataset are critical or advantageous over existing studies. The importance of their research question seems overstated without clear analytical frameworks or hypotheses that guide the investigation.

-The study presents several significant weaknesses that undermine its rigor and contribution to the field of NUE and PUE. Firstly, the methodology section lacks clarity and rigor in key areas. Although the authors compiled a comprehensive global database, the criteria for data inclusion are vague, and the filtering processes and justification for excluding outliers are not specified. This lack of transparency raises concerns about the validity and representativeness of the dataset, particularly as the selection of data points appears biased towards those with higher availability, potentially skewing results and leading to misleading conclusions about global nutrient use efficiency trends. The statistical methods used for analysis, such as random forests, are inadequately defined, with insufficient explanation regarding parameter settings, variable selection, and validation procedures. Furthermore, the reliance on national-level data obscures critical regional variations, especially in agriculturally diverse countries like China and India, while the historical fertilizer input data used is inconsistent, particularly before 1990, raising concerns about the accuracy of long-term trends. Additionally, the imputation of missing environmental and soil data introduces uncertainty, and the absence of experimental validation weakens the robustness of the findings. The study also inadequately considers socioeconomic and policy factors, focusing primarily on environmental influences without addressing the impact of subsidies, farmer education, and governance structures on nutrient use efficiency. Broad

generalizations made about NUE and PUE trends overlook crop-specific and regional nuances, while the global mapping of NUE and PUE, though visually appealing, fails to provide actionable recommendations for improving nutrient management. The use of the N(P)UEdiff method does not yield significant new insights, as results align with existing studies, and the distinctions made do not translate into groundbreaking conclusions. Furthermore, the manuscript lacks practical recommendations for improving nutrient management, and the writing suffers from verbosity and repetition, making key findings difficult to discern. The implications of the results for policy and practice are not well articulated, and the exploration of regional variability is insufficient. The review of existing literature is not exhaustive, and the discussion of environmental impacts is lacking depth. Methodological rigor in estimations is also a concern, as assumptions regarding fertilizer application rates may not accurately reflect field conditions. Lastly, conclusions drawn from the results are often broad and generalized, lacking substantial backing from the presented evidence. To enhance the manuscript, the authors should consider incorporating innovative methodologies, conducting a detailed regional analysis, integrating socioeconomic factors, providing a comprehensive discussion on future agricultural practices, improving clarity, and updating references.

Specific comments:

- (i) In this study, it is crucial to consider the impact of different fertilizer application methods (such as foliar spraying, soil application, and drip irrigation) on nutrient use efficiency (NUE and PUE). For instance, methods such as foliar spraying of urea can significantly enhance nutrient uptake compared to traditional soil applications, especially in conditions where soil nutrient availability is limited. Addressing the effectiveness of various application techniques would provide a more comprehensive understanding of the factors influencing NUE and PUE. Incorporating a review of existing literature on the comparative efficiencies of these methods and offering recommendations for optimizing application based on specific crop types and regional conditions would greatly enhance the depth and practical relevance of your findings. This addition could lead to more targeted nutrient management strategies, ultimately supporting sustainable agricultural practices.
- (ii) Based on the assessment of the paper, it appears that the authors may have overlooked several critical considerations that could enhance the study's rigor and relevance. First, they should ensure a comprehensive review of existing literature on NUE and PUE, as omitting key studies may lead to gaps in understanding and weaken the foundation of their analysis. Additionally, the authors may not have included a diverse range of studies from various geographical regions and agricultural contexts, which could restrict the applicability of their findings to broader agricultural practices. There also seems to be a lack of critical assessment regarding the quality and methodologies of the studies included in their screening, potentially resulting in reliance on studies with varying levels of rigor and leading to biased conclusions. Furthermore, insufficient examination of different fertilizer application methods, such as foliar spraying versus soil application, is a notable gap, as this aspect can significantly impact nutrient efficiency outcomes. Lastly, the authors may not have integrated socioeconomic factors that influence nutrient use efficiency, which could limit the relevance of their findings to real-world agricultural practices. To enhance the depth and relevance of the study, the authors should conduct a more thorough literature review, incorporate a broader range of sources, critically evaluate methodologies, explore the impact of various application methods, and integrate socioeconomic factors affecting nutrient use efficiency. Addressing these areas will significantly strengthen their study and provide valuable insights into improving nutrient use efficiency across major crops.
- (iii) It is beneficial to suggest that the authors consider additional metrics alongside NUE and PUE for a more comprehensive analysis of fertilizer use trends. Incorporating metrics like Fertilizer Application Rate (FAR) and Nutrient Balance (NB) provides a broader context for NUE and PUE, allowing for a more complete understanding of nutrient dynamics. Additionally, metrics such as Year-over-Year Change and Moving Averages can help identify trends over time, offering insights into how fertilizer practices are evolving. A Sustainability Index (SI) that combines various efficiency metrics can further assess the environmental impact and sustainability of fertilizer use, which is crucial in discussions about sustainable agriculture. Furthermore, different regions may have unique challenges and practices; additional metrics can help tailor recommendations and insights to specific contexts. Finally, a comprehensive approach can provide policymakers with more actionable data, leading to better-informed decisions regarding fertilizer regulations and practices. While NUE and PUE are essential metrics, suggesting additional analyses will enhance the manuscript's contribution to the field and provide a more thorough understanding of nutrient management in agriculture.
- (iv) It is not clear which factors have the most effect on NUE and PUE. Several methods can be employed to identify and analyze these influential factors. Statistical analyses, such as multiple regression and correlation analyses, can quantify the relationships between NUE/PUE and various independent factors, helping to identify significant influences. Sensitivity analyses can determine how changes in specific factors, like fertilizer application rates or irrigation practices, etc., impact NUE and PUE, prioritizing factors for further investigation. Additionally, finding field experiments that manipulate different factors—such as varying fertilizer types, application methods, or crop rotations—can provide causal insights into their effects on nutrient efficiency. Longitudinal studies tracking changes in NUE and PUE over time in relation to agricultural practices or environmental conditions can reveal trends and correlations. A meta-analysis of existing studies can aggregate data on NUE and PUE across different contexts, highlighting common influential factors. Machine learning techniques can also be utilized to analyze large datasets and identify key predictors of NUE and PUE. Finally, engaging with agronomists or experts in nutrient management can yield qualitative insights based on their experience and knowledge. By employing these methods, the authors can gain a clearer understanding of which factors significantly affect NUE and PUE, which is crucial for developing effective strategies to enhance nutrient use efficiency in agricultural practices.
- (v) While the use of random forest models is appropriate, the study does not sufficiently address the limitations of this approach. For example, the models may overfit the data, and the variable importance analysis lacks depth. The study does not provide a thorough sensitivity analysis to assess the robustness of the model predictions.
- (vi) The segmented regression approach used to identify breakpoints in NUE and PUE trends is not adequately justified. The choice of breakpoints (e.g., 1975 for NUE and PUE improvements) appears arbitrary and is not supported by a robust theoretical or empirical framework.
- (vii) The study briefly mentions the ecological consequences of nutrient surpluses but does not delve deeply into the environmental impacts of low NUE and PUE. For example, the implications for eutrophication, greenhouse gas emissions, and soil health are not thoroughly explored.

- (viii) The discussion of nutrient surpluses in maize production in China and the United States is superficial. The study does not provide actionable recommendations for mitigating these surpluses or addressing their environmental consequences.
- (ix) The study frequently conflates correlation with causation. For example, the observed improvements in NUE and PUE after 1975 are attributed to technological advancements and policy changes without providing direct evidence to support these claims.
- (x) The study makes broad generalizations about NUE and PUE trends without adequately accounting for crop-specific and region-specific nuances. For instance, the assertion that rice achieves optimal NUE and PUE in tropical zones overlooks the variability within tropical regions due to differences in soil types, farming practices, and climatic conditions.
- (xi) The study focuses almost exclusively on environmental and agricultural factors, neglecting the critical role of socioeconomic and policy drivers in shaping nutrient use efficiency. For example, the impact of subsidies, farmer education, and access to technology on NUE and PUE is not addressed.
- (xii) The analysis does not consider the role of institutional frameworks, market dynamics, or governance structures in influencing nutrient management practices. These factors are essential for understanding the disparities between developed and developing countries.
- (xiii) The global maps of NUE and PUE, while visually striking, oversimplify complex spatial patterns. The resolution of the analysis is insufficient to inform targeted interventions at the local or regional level.
- (xiv) While the study identifies regions with low NUE and PUE, it fails to provide concrete, actionable recommendations for improving nutrient management. The suggestion to reevaluate global crop distribution patterns is vague and lacks specificity.
- (xv) The study relies heavily on meta-analysis and modeling without experimental validation. The absence of field experiments or case studies to corroborate the findings weakens the robustness of the conclusions.
- (xvi) The imputation of missing environmental and soil data using global datasets (e.g., CRU TS, GLEAM, SoilGrids) introduces significant uncertainty. These datasets may not accurately reflect local conditions, leading to potential biases in the analysis.
- (xvii) The study does not engage with the practical challenges of implementing precision agriculture, developing high-efficiency crop varieties, or promoting sustainable farming practices in developing regions.
- (xviii) The use of aggregated national-level data obscures critical regional and local variations in nutrient use efficiency. This approach is particularly problematic in large, agriculturally diverse countries like China, India, and the United States, where NUE and PUE can vary dramatically across regions.
- (xix) The study relies on historical fertilizer input data, which is acknowledged to be inconsistent, especially before 1990. This raises concerns about the accuracy of long-term trends in NUE and PUE, particularly in the early decades (1961–1990).
- (xx) The authors rely on estimates for NUE and PUE that may not accurately reflect field conditions, particularly when extrapolating data across different crops and regions. Further validation of assumptions relating to fertilizer application rates and nutrient uptake from soil is necessary to fortify their claims.
- (xxi) The authors compare NUE and PUE trends across various regions but fail to contextualize these results adequately within the broader agricultural and environmental policy frameworks. For instance, differences in crop management practices or regional agricultural policies should be considered when interpreting nutrient use efficiency trends.
- (xxii) Some results lack statistical significance, yet are presented as conclusive findings. The authors must be cautious in making statements about overall trends without robust statistical evidence.
- (xxiii) The practical implications of the findings are not well articulated. While the authors highlight the need for improved nutrient management, there is little discussion on how these findings should influence agricultural policies or practices. Specific recommendations for targeted interventions in regions with low NUE and PUE need to be detailed, especially given the paper's substantial focus on global patterns.
- (xxiv) Although the abstract mentions reducing ecological pollution, the discussion does not adequately address the environmental implications of low NUE and PUE. More significant attention should be paid to the potential impact on soil health, water quality, and climate change, linking these issues more explicitly to the findings of the research.
- (xxv) The manuscript highlights overall global patterns but fails to adequately analyze regional disparities in NUE and PUE. For example, factors that contribute to inefficiencies in high-input agricultural systems versus low-input regions are not sufficiently discussed. The authors need to explore these differences in greater depth to provide a comprehensive perspective.
- (xxvi) The review of existing literature is not exhaustive, failing to incorporate relevant studies that may provide alternative perspectives or contradict the authors' claims. This lack of engagement diminishes the scholarly rigor of the manuscript.
- (xxvii) The manuscript is overly verbose and repetitive, particularly in the Results and Discussion sections. The key findings are buried in lengthy descriptions, making it difficult for readers to discern the main contributions of the study.
- (xxviii) The abstract and introduction do not effectively highlight the novelty or significance of the study. The objectives are stated in broad terms, and the rationale for the research is not compelling.
- (xxix) The conclusions drawn from the results are often broad and generalized. Many assertions about the future implications of their findings lack the substantial backing of robust evidence presented in the results section. The authors should ensure that their conclusions are directly supported by the results and that they make cautious, evidence-based recommendations.
- (xxx) The figures, while visually appealing, are not always clearly explained. For example, the global maps of NUE and PUE (Figure 4) lack sufficient detail on the underlying data and assumptions.

Reviewer #3

(Remarks to the Author)

I have read with great interest the manuscript 'Global-scale prevalence of low nutrient use efficiency across major crops' and I would like to congratulate the authors for the quality of their study. The paper is rather strong on the spatial analysis of

variations in nutrient use efficiency, across major food crops. The authors have built and compiled an impressive dataset on field-level observations across the globe. The introduction rightly points to the need of field-based estimates of fertiliser use efficiency, to complement existing nation-wide evaluation using nutrient balance approaches. This is very critical, as many global-level studies currently lack this 'bottom-up' approach where ground observations are used for upscaling and understanding of global patterns.

The training of the statistical model with the field observations, and its application for spatial prediction using existing geodata, is adequately detailed. This leads to convincing and critical results on the contrasting impact of climate on nutrient use efficiencies for different crops, and the identification of hotspots where inefficiencies are high. The paper also provides a nice perspective on the optimisation of crop distribution across the globe in order to increase nutrient use efficiency. For all these reasons, I believe this manuscript deserves publication.

I have one major comment though, that I think needs to be addressed. The temporal analysis of variations in nutrient use efficiency seems biased. Below I explain why:

The relationships established in Figure S6 between nutrient use efficiency, soil nutrient supply, and nutrient input intensity, though significant, have a small explanatory power (in most cases explain less than a quarter of the observed variation, $r^2 < 0.25$). These relationships therefore neglect the impact of climate, soil, and management (beyond input intensity, e.g. timing, placement etc), on nutrient use efficiency and soil nutrient supply. They tend to overemphasize the impact of nutrient input intensity on the observed efficiency. Yet, 60% efficiency can be achieved at almost any level of N input (Figure S6, though colours of the dots are not easy to distinguish, this may need to be improved), and efficiency can be very variable at a given nutrient input level.

As far as I understand, these simple relationships are then used to compute historical values of nutrient use efficiencies (L502-514). The authors then analyse historical changes in these efficiencies with the lens of possible changes in crop management (beyond input intensity, e.g. varietal choice, good agronomic practices) in different regions of the world, that would have helped increase or decrease nutrient use efficiency (L150-218). But because intensity of nutrient input and efficiency are so much correlated in the computation, isn't that an artificial construct where greater use of nutrients automatically leads to lower efficiency (conversely, lower use leading to greater efficiency), neglecting how changes in management (and possibly climate) might have altered these connections? Very possibly, lower input use can lead to lower or stable efficiency, if crop management does not improve (this may be happening in some regions of the world). The use of the random forest model (L554-557), that better accounts for the influence of climate, soil and management on efficiency could be an alternative to the equations of Figure S6. But then, historical data on crop management is hard to find.

I suggest the authors provide more details on the computations and its logic, in order to convince the reader that this bias does not exist, or if it does, discuss the current limitations around the data and the computations, and stick to describing the historical changes in input intensity, without concluding too strongly into possible changes in efficiency (both in the abstract and the result sections).

Other minor comments:

Figure 1: Subplot b is not explained (axis, acronyms, title, colors). The caption should indicate that the figure shows the location of the reviewed studies.

L103: 'current N(P)UEbala approaches' the acronym has not been defined and this looks awkward at this stage.

L570: To my understanding 'management' beyond input use intensity is only the variable 'tillage/no tillage'. Did the authors have a look at the number of applications, the type and placement of fertiliser? Surely these variables exist in the reviewed study. Of course, the geodata for these variables do not exist, but the relationships could be built with the training dataset. Do these variables help explain some of the observed variability? The new model could then be used to explore what-if scenarios around improvement in management at different locations.

Figure 3: It is hard to understand what the authors mean with 'During, planting years'. To my understanding, the planting year is confounded with the climatic data. Why would the authors want to have it as a predicting variable? How is this predictor then used for the spatial extrapolation? This deserves clarification.

L297: Organising the food system at global level is a noble objective. Yet, countries might still want to produce their own food (e.g. wheat in Ethiopia) despite lower efficiencies than elsewhere in the world, because food sovereignty (and cultural preferences) is so critical for them, and a better strategy than heavy reliance on imports (see e.g. this paper: <https://iopscience.iop.org/article/10.1088/1748-9326/11/3/035007>). The authors might want to reflect on this issue.

L392: The authors have built an impressive dataset that, if made publicly available, would be of great value for the global research community interested in nutrient management.

L390: This should be Figure S15 I think – but the figure is not so useful, unless it shows the overlap between location of reviewed studies and the biome classification.

Version 1:

Reviewer comments:

Reviewer #1

(Remarks to the Author)

I believe the authors have adequately addressed all my concerns, and I am satisfied with the revised manuscript.

Reviewer #2

(Remarks to the Author)

The authors have made substantial efforts to address the reviewers' concerns, particularly in enhancing methodological transparency, expanding discussions on socioeconomic factors, and improving model validation. However, several critical issues remain unresolved. In terms of methodology and data reliability, the authors incorporated regional examples such as

China's transition and applied Kriging interpolation to reduce spatial uncertainty, achieving a reduction in RMSE for NUE (0.09) and PUE (0.13). They also corrected historical fertilizer data using IFA/FAOSTAT baselines. Nevertheless, the reliance on national-level data and inconsistencies prior to 1990 persist, with the lack of high-resolution regional datasets or early-decade validation introducing potential biases that were not fully addressed. Regarding statistical rigor, segmented regression with breakpoints (e.g., 1975 ± 3) and Wilcoxon tests were added, along with uncertainty intervals for breakpoint years. Despite these improvements, the selection of breakpoints remains inadequately justified, and the absence of autocorrelation analysis or validation against external policy and technological milestones continues to undermine the temporal validity of the claims. In the realm of model validation, the authors implemented 10-fold cross-validation, grid search for hyperparameter tuning, and Kriging interpolation, supported by contour plots and R^2 values. Yet, overfitting risks are not explicitly addressed—such as through learning curves—and the rationale behind parameter choices like nodesize and mtry lacks depth, with spatial uncertainty still evident in Supplementary Fig... On contextual and socioeconomic analysis, GDP and population density were integrated into models, and structural equation modeling identified tillage as a key driver for maize productivity. Still, institutional frameworks such as subsidies and governance were deemed beyond the study's scope, leaving important gaps in understanding regional disparities. Concerning literature review and novelty, the discussion was expanded to include PUE mechanisms and recent references like those on integrated nutrient management. Nonetheless, there is limited engagement with cutting-edge developments such as precision agriculture or CRISPR-edited crops, and the study's originality remains questionable due to its reliance on established methods like NUEbala instead of NUEdiff. Finally, in addressing climate change and policy implications, CMIP6 scenarios (SSP126/245/585) were introduced, projecting fluctuations in NUE and PUE. However, the analysis remains superficial without mechanistic explanations—for instance, why PUE increases under high-emission scenarios—and while policy recommendations have become more specific, such as promoting no-till practices, they lack assessments of scalability or cost-benefit analyses. In conclusion, while the authors have diligently addressed many concerns, critical gaps in methodological transparency, data validation, and policy relevance remain. The study's contributions, though valuable, do not meet the high standards of novelty and rigor expected for a Nature-level publication. The revised manuscript would benefit from deeper mechanistic insights, robust validation of historical trends, and actionable, region-specific policy frameworks. Until these issues are resolved, the work remains better suited for a specialized agricultural journal rather than a broad-scope, high-impact publication like Nature.

Reviewer #3

(Remarks to the Author)

I thank the authors for the implemented changes

Yet, I am not 100% convinced by the modifications done by the authors to accommodate the major comment I made.

The comment was that the relationships between nutrient use efficiency, soil nutrient supply, and nutrient input intensity neglect the impact of climate, soil, and management (beyond input intensity, e.g. timing, placement etc), and that they therefore tend to overemphasize the impact of nutrient input intensity on the observed historical changes in nutrient use efficiency. Intensity of nutrient input and efficiency are so much correlated in the computation, that there is a risk of an artificial construct where greater use of nutrients automatically leads to lower efficiency (conversely, lower use leading to greater efficiency), neglecting how historical changes in management (and possibly climate) might have altered these connections.

The authors updated the prediction of soil nutrient supply with a machine learning model that now accounts for climate variable (mean annual rainfall and temperature), and soil characteristics. But do these predictors vary historically from a year to another? For soil I doubt because there is no such historical dataset of soil characteristics – and I cannot find the information in the manuscript. For climate what was the source of the data? Management data is still not included in the computation of the efficiency, as far as I understand (eq 8,9,10,11).

I reiterate my comment: 'I suggest the authors provide more details on the computations and its logic, in order to convince the reader that this bias does not exist, or if it does, discuss the current limitations around the data and the computations, and stick to describing the historical changes in input intensity, without concluding too strongly into possible changes in efficiency (both in the abstract and the result sections).'

On one of my minor comment: I don't see the locations of the reviewed studies on the map of the climate zones.

Version 2:

Reviewer comments:

Reviewer #2

(Remarks to the Author)

I believe the authors have adequately addressed all my concerns, and I am satisfied with the revised manuscript.

Reviewer #3

(Remarks to the Author)

I would like to thank the authors for the improvement in the description in the methods used to compute historical change in nutrient use efficiency. It is now clear that no management data beyond input use intensity was used in the computation, nor possible changes in soil nutrient over time. Therefore, the analysis cannot robustly conclude on the direction of historical change in nutrient use efficiency, and this should be reflected both in the abstract and the result section.

In their response, the authors indicate that 'we have softened our language in both the Abstract and Results sections to avoid overstating conclusions regarding changes in efficiency over time'. However, I don't see any changes in that direction in the manuscript file with track change. The manuscript cannot be published as such. Also there are several instances in the result section where the computed change in nutrient use efficiency is analysed with the lens of changes in management (beyond mere intensity, e.g. 'precision fertilization, use of efficient varieties) and/or soil characteristics (e.g. soil acidification). One cannot explain the output of a model with processes that are not incorporated into that model. What the authors show here is a decline in efficiency when input intensity increases. They have made no contribution in understanding how precision fertilisation and change in soil characteristics have influenced that efficiency. That should be very clear in both abstract and result section.

Version 3:

Reviewer comments:

Reviewer #3

(Remarks to the Author)

The authors adequately addressed my concerns

**REVIEWER COMMENTS**

**Reviewer #1 (Remarks to the Author):**

*1: The paper tackles the critical global challenge of low nitrogen and phosphorus use*
*efficiency (NUE and PUE), emphasizing their profound implications for sustainable*
*agriculture and environmental protection. By compiling an extensive global database*
*of 3,360 observations spanning 205 countries, the authors provide a robust foundation*
*for their findings. They also utilized advanced statistical tools, like random forest, to*
*analyze and predict patterns of NUE and PUE, adding depth to their analysis. The*
*differentiation of these efficiencies across crops, regions, and climates is meticulously*
*examined, alongside an insightful exploration of temporal trends, such as the decline*
*in NUE and PUE during the early Green Revolution and their gradual recovery since*
*the mid-1970s. The inclusion of biogeographical and environmental factors further*
*enriches the contextual analysis. Notably, the study offers practical, actionable*
*strategies—such as optimizing crop distribution and adopting precision agriculture*
*techniques—that hold significant potential for enhancing nutrient use efficiency on a*
*global scale. Nevertheless, I have some concerns that must be addressed. Therefore, I*
*recommend a major revision.*

**Response:** Thank you for your positive and constructive comments on our manuscript.
We greatly appreciated your recognition on the contribution of our work. We were also
pleased to hear that you found the exploration of temporal trends, biogeographical, and
environmental factors to be valuable aspects of the study. We have carefully studied the
reviews, and incorporated them into our manuscript. We aimed to address all your
comments below.

*2: My concern #1: I understand that the authors acknowledge the limitations of*
*national-level data and mention that the lack of region-specific, high-resolution data*
*might lead to generalized conclusions that obscure local variations (L360-361).*
*Therefore, I suggest discussing how future iterations of the study could benefit from*
*incorporating regional datasets or experimental validations. Additionally, could you*
*include at least a few regional examples or case studies to make the findings more*
*relatable and actionable at the local scale?*

**Response:** Thank you for your insightful comment regarding the importance of
regional data. We agree that national-level data may not fully reflect the local variations
that are vital for understanding agricultural transitions. In response, we have revised the
manuscript to emphasize the value of incorporating region-specific data in future
studies. We also added an example from China, where certain regions are shifting from
small-scale farming to more intensive agricultural practices, a transformation that
cannot be captured by national-level datasets. We hope this addition work would
strengthen the relevance of our study for local-scale applications. We have clarified this
issue as “First, the reliance on national-level data may obscure regional differences in
historical trends, particularly in countries with diverse climates and agricultural
practices. While national datasets provide a broad overview, they may not capture the
local variations crucial for the development of region-specific strategies. For example,
in China, some regions are undergoing a transition from small-scale farming to more
intensive agricultural practices, a shift that national-level data may fail to capture⁴⁶.
Future iterations of the study could benefit from incorporating regional datasets or
conducting experimental validations at the local level. (Revised lines 411-418)”

**3:** *My concern #2: The study focuses heavily on environmental and agricultural factors,*
*but the role of socioeconomic influences (e.g., farmer behavior, government policies)*
*in determining NUE and PUE appears underexplored. The authors have briefly*
*mentioned the Green Revolution and its impact on government policies (L165-204 and*
*elsewhere). I suggest expanding on these aspects to enrich the analysis and enhance*
*the relevance of the recommendations. For example, consider integrating data on*
*farmer practices, economic incentives, and regulatory frameworks to provide a more*
*holistic view of the factors influencing nutrient use efficiency.*

**Response:** We greatly appreciate your suggestion regarding the incorporation of
socioeconomic factors into our analysis. To address this important point, we have now
expanded our study to include the influence of socioeconomic factors on NUE and PUE.
Specifically, we have integrated socioeconomic variables into the correlation analysis,
structural equation modeling, random forest analysis, and global predictions. In the
revised version, we now explore the role of factors such as farmer behavior, economic
incentives, and government policies, which we believe significantly impact NUE and
PUE. This addition provides a more comprehensive view of the various factors
influencing nutrient use efficiency, complementing the environmental and agricultural
considerations discussed earlier in the paper. We have also highlighted how
socioeconomic pressures in developing countries, such as rapid population growth and
limited access to sustainable farming technologies, contribute to less efficient nutrient
use. We hope these additions enrich the analysis and make the findings more relevant
to a broader set of stakeholders involved in agricultural sustainability.

The specific modified text is as follows: “(8) fertilizers application types (FAT)
including organic fertilizers, inorganic fertilizers, and the combined application of
organic and inorganic fertilizers; (9) fertilizers application placement (FAP) including
surface application, deep application, mixed application, and foliar spraying; (10)
fertilizer application frequency (FAF); (11) irrigation methods (IM) including
permanent flooding, intermittent irrigation, drip irrigation, and no irrigation; (Revised
lines 459-463)

In addition, we also incorporated socio-economic factors into the analysis of the
impacts on the NUE and PUE of major crops. Socio-economic factors including gross
domestic product per capita (GDPPC), population density (PD), urbanization rate (UR)
and mean years of schooling (MYS). Due to the lack of reports on relevant population
and economic aspects in the literature, we obtained information at the national level
from the World Bank database and Human Development Data. (Revised lines 505-510)”

**4:** *My concern #3: L210-213: These suggestions present valuable insights for driving*
*a second Green Revolution. However, without providing economic incentives to farmers,*
*implementing these recommendations comprehensively—particularly in developing*
*nations—may prove challenging.*

**Response:** Good suggestion. We fully agree that without economic incentives,
implementing these strategies is likely to be difficult, especially in developing nations
with limited financial and technological resources. We now highlighted the importance

of providing economic incentives (e.g., financial support, subsidies for sustainable
practices, and affordable technologies) to farmers as “To advance the second Green
Revolution, strategies should include developing high-efficiency crop varieties,
deploying precision agriculture, promoting soil health through rotation and
intercropping, and leveraging biological nutrient sources like P-solubilizing and N-
fixing microorganisms. Ensuring the comprehensive implementation of these strategies,
especially in developing nations, requires providing economic incentives to farmers.
Financial support, subsidies for adopting sustainable practices, and access to affordable
technologies can incentivize farmers to embrace these innovations, facilitating their
widespread adoption and ensuring long-term success^{20,22-26}. (Revised lines 222-230)”

**5: My concern #4: L232-234:** *“This trend... preserves toward saturation.” I do not*
*understand this line. Are you referring to 'P reserves' or 'preserves'? Please clarify.*

**Response:** Sorry for the confusion. We referred to ‘P reserves’ rather than ‘Preserves’
here. To make it more clearly, we clarified this issue as “This trend, likely driven by
historical, intensive P fertilizer application, has gradually pushed soil P reserves toward
saturation^{2,27}. (Revised lines 248-250)”

**6: My concern #5:** *Many conclusions are drawn from data spanning 1961–2018, with*
*limited emphasis on recent trends or future projections. I recommend addressing this*
*gap through scenario modeling or projections beyond 2020. Additionally, I suggest*
*emphasizing how the findings can be translated into actionable recommendations for*
*policymakers, particularly in developing nations.*

**Response:** Thank you very much for highlighting this point. We have expanded the
study by incorporating scenario modeling to project the potential impact of climate
change on NUE and PUE beyond 2020. Specifically, we have included projections
under the "fossil-fueled development-high forcing" scenario, and showing fluctuations
in NUE and PUE based on future climate conditions (Supplementary Fig. 15).
Furthermore, we utilized structural equation model to assess the effectiveness of
agronomic practices, including no-till farming, in reducing nitrogen and phosphorus
surpluses, particularly within maize production systems. These enhancements
contribute to a more holistic understanding of future scenarios and yield policy-relevant,
actionable recommendations.

The specific modified text is as follows: “Second, this study primarily focuses on
the analysis of crop NUE and PUE in the historical and present contexts. However, we
also observe that climate change can influence NUE and PUE. For instance, under the
'fossil-fueled development-high forcing' scenario, NUE in rice is expected to decrease,
while PUE is projected to increase (Supplementary Fig. 15). These findings suggest
that the NUE and PUE of crops will continue to fluctuate depending on climate change.
By incorporating future changes in soil properties and socioeconomic factors, we can
better adopt optimal agricultural management practices to improve crop NUE and PUE.
(Revised lines 418-426)

The inefficiencies in NUE and PUE in global crop production have resulted in
significant nutrient surpluses, which have had profound impacts on the ecological

environment (Fig. 5). Maize, in particular, registers the highest N (73 kg N ha^{-1}) and P
 (15 kg P ha^{-1}) surplus intensities among major crops. China and the United States, which
 are the major N and P surpluses in the maize ecosystem, contribute 56.7% of the N
 surplus load and 61.9% of the P surplus load due to higher ($99.4\text{-}115 \text{ kg N ha}^{-1}$ and 16.8-
 $33.7 \text{ kg P ha}^{-1}$) than the global average surplus intensities and large planted areas
 ($4.21 \times 10^5 \text{ km}^2$ for China and $3.29 \times 10^5 \text{ km}^2$ for the United States). Furthermore, our
 structural equation model reveals that agricultural management practices are the most
 significant factor influencing NUE and PUE in maize, with tillage methods having the
 most pronounced impact (Fig. 3). Specifically, the adoption of no-till practices can
 concurrently improve NUE and PUE in maize, offering a practical approach to reduce
 the N and P surpluses in maize ecosystems. (Revised lines 379-391)”

Fig. S15 Temporal trends of nitrogen (NUE) and phosphorus use efficiency (PUE)

**of four major crops (rice, wheat, maize, soybean) under SSP126 (sustainable**
**development-low forcing), SSP245 (moderate development-medium forcing), and**
**SSP585 (fossil-fueled development-high forcing) scenarios.**

*7: My concern #6: The authors have discussed NUE extensively. Nonetheless, I find the*
*discussions of PUE to be less thorough. A more balanced discussion is needed,*
*especially considering the critical importance of phosphorus as a non-renewable*
*resource. For example, in L234-245, the authors propose some good solutions; however,*
*they overlook a critical factor: the role of soil organic matter. Organic matter plays a*
*vital role in enhancing phosphorus availability by releasing organic acids that*
*solubilize bound phosphorus. Unfortunately, many soils worldwide are characterized*
*by low organic carbon content, which diminishes this beneficial effect. Improving PUE*
*also relies on adopting integrated nutrient management strategies, optimizing P*
*fertilizer placement techniques, and enhancing irrigation practices. Continuous*
*cultivation of high phosphorus-demanding crops, coupled with minimal fallow periods,*
*significantly depletes the available phosphorus pool. Additionally, many farmers in*
*developing countries lack access to soil testing facilities or fail to follow*
*recommendations based on soil tests, resulting in inefficient phosphorus use. These*
*critical challenges warrant thorough discussion.*

**Response:** Thank you for your valuable feedback. In response to your concern about
the discussion of PUE, we have expanded the section to include a more thorough
exploration of the role of soil organic matter in enhancing phosphorus availability. We
have also discussed the challenges associated with soils that are low in organic carbon
content, which diminishes the beneficial effects of organic acids in mobilizing
phosphorus. Additionally, we have emphasized the importance of integrated nutrient
management strategies, optimizing phosphorus fertilizer placement, and improving
irrigation practices for better PUE. Furthermore, we addressed the issue of continuous
cultivation of high-phosphorus-demanding crops, the depletion of phosphorus reserves,
and the challenges posed by limited access to soil testing facilities in developing
countries. These revisions aim to provide a more balanced and comprehensive
discussion of PUE and its improvement strategies.

The specific modified text is as follows: “These findings suggest that current crop
P uptake relies primarily on soil P mobilization rather than inorganic fertilizer inputs.
This trend, likely driven by historical, intensive P fertilizer application, has gradually
pushed soil P reserves toward saturation^{2,27}. While this process has helped meet crop P
demands in the short term, it has also created long-term sustainability challenges. P
availability in soils is influenced not only by inorganic fertilizer inputs but also by soil
organic matter, which plays a crucial role in enhancing P availability by releasing
organic acids that solubilize bound P^{12,28}. However, many soils worldwide, particularly
in regions with low organic matter content, fail to fully benefit from this mechanism,
limiting the PUE. To further improve PUE, adopting integrated nutrient management
strategies is critical. These strategies include optimizing P fertilizer placement,
improving irrigation practices, and managing crop rotations to minimize continuous
cultivation of high-P-demanding crops^{12,29-31}. In many developing countries, the lack of

access to soil testing facilities or failure to implement recommendations based on soil
tests exacerbates inefficient P use, leading to imbalanced fertilization and suboptimal P
availability^{32,33}. In addition, minimizing fallow periods and managing soil organic
matter content through practices such as cover cropping or organic amendments could
enhance P availability and utilization³⁴. Thus, while regulating the release of retained
soil P could reduce P fertilizer inputs over time, it is crucial to adopt a holistic approach
that addresses the complex factors influencing P availability and uptake. (Revised lines
247-266)”

**8:** *My concern #7: L312-314: The authors propose encouraging rice cultivation in*
*tropical regions and wheat cultivation in temperate zones; however, this*
*recommendation faces several practical challenges. For example, in India,*
*implementing region-specific crop distribution to enhance NUE and PUE faces several*
*obstacles. Overdependence on irrigation for rice cultivation in water-stressed regions*
*like Punjab and Haryana has led to severe groundwater depletion, while tropical areas,*
*despite higher rainfall, often lack the necessary irrigation infrastructure. Soil*
*conditions also pose hurdles, with high phosphorus fixation in acidic tropical soils*
*reducing PUE and salinity or nutrient depletion limiting wheat expansion in temperate*
*zones. Additionally, historical and economic factors, such as the profitability of rice*
*cultivation under government procurement schemes, make farmers reluctant to change*
*cropping patterns. Environmental concerns also arise, as cultivating rice in tropical*
*regions can exacerbate water stress, while expanding wheat in nutrient-depleted areas*
*may increase reliance on chemical inputs. Moreover, imbalanced fertilizer use and*
*limited access to advanced soil testing facilities further hinder NUE and PUE*
*improvements. Addressing these challenges requires robust policy support,*
*infrastructure development, and farmer education to promote sustainable and efficient*
*cropping practices. Thus, I believe the authors should expand the discussion to*
*incorporate multiple perspectives.*

**Response:** Thank you for your insightful comments. We have revised the manuscript
to address the practical challenges you highlighted, including limitations in water
resources, soil conditions, and historical economic constraints. Furthermore, we have
expanded the discussion regarding the higher yields in major food-producing countries,
emphasizing that while these regions exhibit higher average yields, implementing
region-specific strategies requires addressing the local challenges you mentioned, such
as infrastructure gaps and economic incentives. We believe these revisions provide a
more comprehensive and realistic view of the potential for optimizing crop distribution
to enhance NUE and PUE.

The specific modified text is as follows: “Practical challenges arise when
implementing region-specific crop distribution. For instance, in countries like India,
overdependence on irrigation for rice cultivation in water-stressed regions such as
Punjab and Haryana has led to severe groundwater depletion⁴³. Additionally, tropical
regions, despite receiving higher rainfall, often lack the necessary irrigation
infrastructure for optimal rice cultivation⁴⁴. Soil conditions also pose significant
challenges, with high P fixation in acidic tropical soils reducing PUE, while salinity

and nutrient depletion limit the expansion of wheat cultivation in temperate zones^{28,45}.
Moreover, historical and economic factors, such as the profitability of rice cultivation
under government procurement schemes, often make farmers reluctant to change
cropping patterns. Expanding rice cultivation in tropical regions could exacerbate water
stress, and expanding wheat in nutrient-depleted areas might increase reliance on
chemical fertilizers. Imbalanced fertilizer use and limited access to advanced soil
testing facilities further hinder improvements in NUE and PUE⁹. Despite these hurdles,
optimizing crop distribution could still lead to significant environmental and food
security benefits, provided these challenges are addressed through comprehensive,
region-specific strategies. (Revised lines 351-366)”

**9:** *My concern #8: L554-569: The authors used RF to create global NUE and PUE*
*maps. I recommend using digital soil mapping (DSM) (McBratney et al., 2003) for this.*
*DSM-based Regression Kriging (RK) instead of relying solely on the RF model for*
*predicting global NUE and PUE could offer several advantages. A simple RF model*
*lacks a spatial component to account for geographic variation. Regression Kriging, on*
*the other hand, combines a regression model (e.g., RF, Cubist, or any other model) with*
*spatial interpolation (kriging), allowing it to incorporate both environmental*
*covariates and spatial autocorrelation explicitly. This makes RK particularly suited for*
*capturing the spatial heterogeneity of NUE and PUE across different regions and*
*climates. Since this is a global study and accurately delineating spatial patterns is*
*essential, RK would enhance prediction accuracy by using both the regression*
*relationships between soil properties, climate, and NUE/PUE, and the spatial*
*dependence of these variables. Additionally, RK provides more robust uncertainty*
*estimates for spatial predictions, which is crucial for guiding targeted interventions.*
*Thus, I recommend integrating RK into DSM to address the spatial complexities*
*inherent in global nutrient use efficiency mapping.*

*McBratney, A. B., Santos, M. M., & Minasny, B. (2003). On digital soil mapping.*
*Geoderma, 117(1-2), 3-52.*

**Response:** We sincerely appreciate your insightful comment regarding the integration
of Regression Kriging (RK) with Digital Soil Mapping (DSM), which would greatly
enhanced the rigor of our spatial analysis. Following your suggestion, we have applied
ordinary kriging interpolation to correct the global predictions of NUE and PUE
obtained from the Random Forest model. Compared to using random forest predictions
alone, the combined approach integrating random forest modeling with Kriging
interpolation demonstrated improved predictive accuracy. Specifically, Kriging
correction reduced the RMSE between predicted and observed values by 0.09 for NUE
and 0.13 for PUE.

The specific modified text is as follows: “Then, the global model predictions of
NUE and PUE were spatially corrected through Kriging interpolation based on NUE
and PUE residuals (observed minus predicted values) using the 'gstat' package in R.
This approach better captures the geographical variability of nutrient use efficiency
while providing more robust uncertainty estimates for spatial predictions^{63,64}, with the
RMSE between predicted and observed values of NUE and PUE decreasing by 0.09

and 0.13 after applying Kriging interpolation for correction, respectively. (Revised lines
656-661)”

**References**

McBratney, A. B., M. M. Santos, and B. Minasny. On digital soil mapping.
*Geoderma* **117**, 3-52 (2003). (Revised lines 927-928)

Wadoux, A. M.-C., B. Minasny, and A. B. McBratney. Machine learning for digital
soil mapping: Applications, challenges and suggested solutions. *Earth-Sci. Rev.* **210**,
103359 (2020). (Revised lines 929-930)”

**Figure 4 Global spatial patterns of nitrogen (NUE) and phosphorus use efficiency**
**(PUE) of four major crops (rice, wheat, maize, soybean) that were predicted based**
**on random forest models.**

Mean refers to the mean value and SD refers to the standard deviation (a, b). The solid
lines and shaded areas in the latitudinal pattern map represent the latitudinal mean and
the latitudinal standard deviation, respectively.

**Response summary:** In response to your thoughtful and constructive suggestions, we

have substantially revised the manuscript to address all major concerns. We believe
these revisions significantly improve the rigor, clarity, and applicability of the study,
and we sincerely thank you for your insightful feedback that helped shape this improved
version of the manuscript.

**Reviewer #2 (Remarks to the Author):**

**10:** *The manuscript titled "Global-scale prevalence of low nutrient use efficiency across*
*major crops" addresses a significant issue in sustainable agriculture by examining*
*nutrient use efficiency (NUE and PUE) on a global scale. However, several critical*
*weaknesses and areas for improvement warrant serious consideration before*
*publication.*

*General comments:*

*The study lacks novelty, as it does not present new methodologies or significant findings*
*compared to existing literature. While it compiles a comprehensive dataset, the insights*
*drawn are largely reiterative of previously established concepts regarding nutrient*
*inefficiency. Specifically, the research does not advance the current understanding of*
*Nutrient Use Efficiency (NUE) and Phosphorus Use Efficiency (PUE) in major crops,*
*relying heavily on existing literature and established trends without offering substantial*
*new data or innovative perspectives that would justify publication in a high-impact*
*journal such as Nature Communications. The study primarily reiterates well-*
*established knowledge about global nutrient use efficiency trends, such as the low NUE*
*and PUE in developing regions and the disparities between developed and developing*
*countries. These findings are not novel and have been extensively documented in prior*
*literature, including works by Zhang et al. (2015) and Zou et al. (2022) cited in this*
*paper.*

*-While the authors mention gaps in understanding NUE and PUE dynamics, they fail*
*to articulate a compelling rationale for why their specific research approach and*
*dataset are critical or advantageous over existing studies. The importance of their*
*research question seems overstated without clear analytical frameworks or hypotheses*
*that guide the investigation.*

*The study presents several significant weaknesses that undermine its rigor and*
*contribution to the field of NUE and PUE. Firstly, the methodology section lacks clarity*
*and rigor in key areas. Although the authors compiled a comprehensive global database,*
*the criteria for data inclusion are vague, and the filtering processes and justification*
*for excluding outliers are not specified. This lack of transparency raises concerns about*
*the validity and representativeness of the dataset, particularly as the selection of data*
*points appears biased towards those with higher availability, potentially skewing*
*results and leading to misleading conclusions about global nutrient use efficiency*
*trends. The statistical methods used for analysis, such as random forests, are*
*inadequately defined, with insufficient explanation regarding parameter settings,*
*variable selection, and validation procedures. Furthermore, the reliance on national-*
*level data obscures critical regional variations, especially in agriculturally diverse*
*countries like China and India, while the historical fertilizer input data used is*
*inconsistent, particularly before 1990, raising concerns about the accuracy of long-*
*term trends. Additionally, the imputation of missing environmental and soil data*
*introduces uncertainty, and the absence of experimental validation weakens the*
*robustness of the findings. The study also inadequately considers socioeconomic and*

*policy factors, focusing primarily on environmental influences without addressing the*
*impact of subsidies, farmer education, and governance structures on nutrient use*
*efficiency. Broad generalizations made about NUE and PUE trends overlook crop-*
*specific and regional nuances, while the global mapping of NUE and PUE, though*
*visually appealing, fails to provide actionable recommendations for improving nutrient*
*management. The use of the N(P)UEdiff method does not yield significant new insights,*
*as results align with existing studies, and the distinctions made do not translate into*
*groundbreaking conclusions. Furthermore, the manuscript lacks practical*
*recommendations for improving nutrient management, and the writing suffers from*
*verbosity and repetition, making key findings difficult to discern. The implications of*
*the results for policy and practice are not well articulated, and the exploration of*
*regional variability is insufficient. The review of existing literature is not exhaustive,*
*and the discussion of environmental impacts is lacking depth. Methodological rigor in*
*estimations is also a concern, as assumptions regarding fertilizer application rates may*
*not accurately reflect field conditions. Lastly, conclusions drawn from the results are*
*often broad and generalized, lacking substantial backing from the presented evidence.*
*To enhance the manuscript, the authors should consider incorporating innovative*
*methodologies, conducting a detailed regional analysis, integrating socioeconomic*
*factors, providing a comprehensive discussion on future agricultural practices,*
*improving clarity, and updating references.*

**Response to Reviewer’s General Comments:** We greatly appreciate your valuable
comments and suggested amendments. Your input has helped improve the paper
tremendously. We have carefully studied the reviews, and incorporated them into our
manuscript. We aimed to address all your comments below.

377 **1. Novelty of the Study:**

While we acknowledge that trends in nutrient inefficiency, particularly NUE and PUE
in major crops, have been widely studied, we believe that our study makes a unique
contribution in the following ways: As highlighted by Reviewer #3, “This is very
critical, as many global-level studies currently lack this ‘bottom-up’ approach where
ground observations are used for upscaling and understanding of global patterns.” This
is precisely where its innovation lies. Most traditional studies rely on the nutrient
balance method at the national level (such as N(P)UE_{bala}). In contrast, this study
provides a more accurate crop-regional level analysis by directly integrating field
observation data, which represents a significant improvement over existing methods.

Furthermore, as emphasized by Reviewer #1, “By compiling an extensive global
database of 3,360 observations spanning 205 countries, the authors provide a robust
foundation for their findings. They also utilized advanced statistical tools, like random
forest, to analyze and predict patterns of NUE and PUE, adding depth to their analysis.
The differentiation of these efficiencies across crops, regions, and climates is
meticulously examined, alongside an insightful exploration of temporal trends, such as
the decline in NUE and PUE during the early Green Revolution and their gradual
recovery since the mid-1970s.” These findings are far from simply replicating existing

conclusions. Instead, they reveal more detailed patterns through new data and methods.
For example, the differential impacts of climate on the NUE and PUE of different crops
(such as the differences between tropical crops and temperate crops). Overall, our study
has the following advantages:

Comprehensive Global Dataset: Our study compiles an extensive global dataset,
which includes 3,360 observations across 205 countries and regions, offering a more
comprehensive and larger-scale analysis than previously available studies. This large-
scale dataset allows for a more robust examination of regional and crop-specific
variations in nutrient use efficiency.

Application of the N(P)UE_{diff} Method: We applied the N(P)UE_{diff} method, which
directly compares nutrient uptake between fertilized and unfertilized conditions. While
this approach has been used in some studies, our global application and comparison
provide new insights into the spatial and temporal trends of nutrient inefficiency across
regions and crops. This method offers a clearer assessment of the impact of inorganic
fertilizers compared to traditional N(P)UE_{bala} methods.

Machine Learning-Based Spatial Analysis: In our analysis, we utilized machine
learning models to predict and delineate the global spatial distribution of NUE and PUE
for major crops. This integration of advanced modeling techniques represents an
innovative approach to understand the drivers and trends in nutrient use efficiency, as
it combines biophysical, climatic, and agricultural management factors in a manner not
previously done at such a global scale.

We believe that the combination of a large-scale dataset, novel methodological
approaches, and the use of machine learning for spatial analysis contributes meaningful
insights to the current understanding of global nutrient inefficiency trends and will be
valuable to the scientific community.

**2. Research Approach and Dataset:**

You highlighted that the study fails to clearly articulate why the specific research
approach and dataset are critical or advantageous over existing studies. We address this
concern as follows:

Research Gaps and Justification: We have now expanded our discussion on the
rationale behind our approach, emphasizing the significant gaps in understanding NUE
and PUE dynamics at global scales. Existing studies have often relied on limited
datasets or have focused on individual countries or regions, while our work provides a
comprehensive global perspective that includes a wide range of crops, countries, and
climatic zones. This holistic approach allows us to better identify global hotspots for
nutrient inefficiency and explore the underlying drivers of spatial variability.

Methodological Framework: In response to your concern, we have also provided
a more detailed explanation of the analytical frameworks used in our study, including
the N(P)UE_{diff} method and machine learning techniques. These frameworks were
specifically chosen to overcome the limitations of traditional methods and provide a
clearer picture of nutrient use efficiency across different crops and regions. We have
elaborated on how this approach offers more actionable insights for improving nutrient
management practices.

**3. Methodological Rigor and Clarity:**

You expressed concerns about the clarity and rigor of our methodology, specifically
regarding the criteria for data inclusion, outlier exclusion, and the statistical methods
used. We have made several improvements to address these concerns:

Data Inclusion and Outlier Exclusion: We have clarified the criteria used for data
inclusion and outlier exclusion. All included studies provided explicit records of NUE
and PUE or crop nutrient uptake, spanning at least one full crop cycle and including
control and experimental treatments. Outliers were identified based on nutrient uptake
values that were inconsistent with field conditions, and these were excluded to ensure
the accuracy of the dataset. We now provide a more transparent explanation of this
process in the revised manuscript.

Statistical Methods: We have strengthened the explanation of the statistical
methods used in our analysis. The random forest models were carefully tuned, and we
now provide more detailed information on the parameter settings, variable selection,
and validation procedures. We have included additional details on the cross-validation
process and the assessment of model performance to ensure the robustness of our
findings.

Regional Variability: In response to your concerns about the reliance on national-
level data, we acknowledge that some regional variations, particularly in agriculturally
diverse countries like China and India, may have been obscured. To address this, we
have conducted further analyses using regional data where available and have
highlighted these regional differences more explicitly in the revised manuscript.

Figure 3 Global-scale correlations and structural equation model showing the relative importance of climate, soil properties, agriculture management, and socio-economic on nitrogen (NUE) and phosphorus use efficiency (PUE) of four major crops (rice, wheat, maize, soybean).

MAT, mean annual temperature; MAP, mean annual precipitation; ET, evapotranspiration; AI, aridity index; IND, inorganic nitrogen deposition; OND, organic nitrogen deposition; Sand, sand content; Silt, silt content; Clay, clay content;

BD, bulk density; TC, total carbon; TN, total nitrogen; TP, total phosphorus; AP,
 available phosphorus; Tillage, no-till or not; During, experimental period; Input N,
 nitrogen fertilizer input; Input P, phosphate fertilizer input; Input K, potash fertilizer
 input; FAT, fertilizer application types; FAF, fertilizer application frequency; FAP,
 fertilizer application placement; IM, irrigation method; GDPPC, gross domestic
 product per capita; PD, population density; UR, urbanization rate; MYS, mean years of
 schooling. *: $P < 0.05$; **: $P < 0.01$; ***: $P < 0.001$; ****: $P < 0.0001$ (a). The standard
 coefficient for each path is shown in a circle within the path. The overall predictive
 performance of the model is evaluated using the Goodness-of-Fit (GOF) statistic, with
 higher values indicating better prediction. The amount of variance explained by the
 model (R^2) is also shown for each response variable (b).

**Figure 4 Global spatial patterns of nitrogen (NUE) and phosphorus use efficiency**
 **(PUE) of four major crops (rice, wheat, maize, soybean) that were predicted based**
 **on random forest models.**

Mean refers to the mean value and SD refers to the standard deviation (a, b). The solid
 lines and shaded areas in the latitudinal pattern map represent the latitudinal mean and
 the latitudinal standard deviation, respectively.

**Fig. S1 Research period (a) and workflow (b) on global inorganic fertilizer**
 **nitrogen use efficiency (NUE) and phosphorus use efficiency (PUE) trends and**
 **patterns.**

Our research involved the two periods of the first Green Revolution and the second
 Green Revolution. During these periods, we first compiled and assimilated the NUE,
 PUE and environment variables at each sampling point, sourced from meta-analysis, to
 acquire optimized parameter values and the relationship between fertilizer input and
 nitrogen(phosphorus) from soil. N and P difference approach were employed to
 compute historical data on global NUE and PUE from 1960 to 2018, utilizing the
 relationship between fertilizer input and nitrogen(phosphorus) from soil, parameters
 from IFA and FAOSTAT. These historical data were utilized to depict the historical

temporal trends and respective breakpoints of global NUE and PUE through segmental
 fitting. Moreover, random forest model predicted the patterns of global NUE and PUE
 from 2000 to 2020, using the optimized parameters, 16 environmental variables and 5
 agriculture management factors. The predicted values were further applied to calculate
 the global N and P surplus.

**Fig. S2 The Preferred Reporting Items for Systematic Reviews and Meta-Analyses**
 **(PRISMA) for the meta-analysis.**

4. Consideration of Socioeconomic and Policy Factors:

You noted that our study focuses primarily on environmental factors, without adequately considering the impact of socioeconomic and policy factors. We have revised the manuscript to integrate these factors more thoroughly:

Incorporation of Socioeconomic Factors: We have now included a more detailed discussion of how socioeconomic factors, such as GDP per capita, population density, and farmer education, influence NUE and PUE. We have also acknowledged the role of government subsidies, policies, and agricultural incentives in shaping nutrient use practices. By examining these factors, we offer a more comprehensive understanding of the drivers of nutrient inefficiency, especially in developing regions.

Policy and Practical Recommendations: In response to your comment on the lack of actionable recommendations, we have added a section that outlines practical strategies for improving nutrient use efficiency based on our findings. These recommendations include optimizing fertilizer application, enhancing soil health management, and promoting sustainable agricultural practices in both developed and developing countries.

The specific modified text is as follows: “Reevaluating global crop distribution patterns emerges as a viable strategy for enhancing NUE and PUE. Encouraging rice cultivation in the tropics and wheat cultivation in temperate zones, where efficiencies are notably higher than in other regions of the world (Fig. 4 and Supplementary Fig. 11), could significantly boost production efficiencies. However, as the 2008-2010 food crisis highlighted, countries might still want to produce their own food for reasons of food sovereignty, cultural preferences, and national security, even when efficiencies are lower compared to global averages. For example, in Ethiopia, wheat production is prioritized despite lower efficiencies, as local food production is crucial for national food security and cultural practices⁴². This can pose a challenge for adopting large-scale, region-specific crop distribution strategies that might require shifts in national agricultural priorities.

Practical challenges arise when implementing region-specific crop distribution. For instance, in countries like India, overdependence on irrigation for rice cultivation in water-stressed regions such as Punjab and Haryana has led to severe groundwater depletion⁴³. Additionally, tropical regions, despite receiving higher rainfall, often lack the necessary irrigation infrastructure for optimal rice cultivation⁴⁴. Soil conditions also pose significant challenges, with high P fixation in acidic tropical soils reducing PUE, while salinity and nutrient depletion limit the expansion of wheat cultivation in temperate zones^{28,45}. Moreover, historical and economic factors, such as the profitability of rice cultivation under government procurement schemes, often make farmers reluctant to change cropping patterns. Expanding rice cultivation in tropical regions could exacerbate water stress, and expanding wheat in nutrient-depleted areas might increase reliance on chemical fertilizers. Imbalanced fertilizer use and limited access to advanced soil testing facilities further hinder improvements in NUE and PUE⁹. Despite these hurdles, optimizing crop distribution could still lead to significant environmental and food security benefits, provided these challenges are addressed through comprehensive, region-specific strategies.

Furthermore, yields per unit area of major food-producing countries in these
regions are higher than in other regions of the world, promising a win-win situation in
terms of both environmental mitigation and food security (Supplementary Fig. 12b).
For example, rice yields averaged 7,028 kg ha⁻¹ in China, 5,203 kg ha⁻¹ in Indonesia,
and 5,818 kg ha⁻¹ in Vietnam, all of which exceed the global average of 4,673 kg ha⁻¹.
Similarly, wheat yields averaged 5,242 kg ha⁻¹ in the European Union and 5,416 kg ha⁻¹
in China, surpassing the global average of 3,423 kg ha⁻¹. While these yield figures
suggest a win-win situation in terms of both environmental mitigation and food security,
the actual implementation in specific regions requires overcoming the challenges of
irrigation infrastructure, soil health, and economic incentives for farmers. Therefore, a
more nuanced, context-specific approach is essential for translating these general yield
trends into actionable recommendations.

The inefficiencies in NUE and PUE in global crop production have resulted in
significant nutrient surpluses, which have had profound impacts on the ecological
environment (Fig. 5). Maize, in particular, registers the highest N (73 kg N ha⁻¹) and P
(15 kg P ha⁻¹) surplus intensities among major crops. China and the United States, which
are the major N and P surpluses in the maize ecosystem, contribute 56.7% of the N
surplus load and 61.9% of the P surplus load due to higher (99.4-115 kg N ha⁻¹ and 16.8-
33.7 kg P ha⁻¹) than the global average surplus intensities and large planted areas
(4.21×10⁵ km² for China and 3.29×10⁵ km² for the United States). Furthermore, our
structural equation model reveals that agricultural management practices are the most
significant factor influencing NUE and PUE in maize, with tillage methods having the
most pronounced impact (Fig. 3). Specifically, the adoption of no-till practices can
concurrently improve NUE and PUE in maize, offering a practical approach to reduce
the N and P surpluses in maize ecosystems. (Revised lines 340-391)”

In response to your concerns, we have revised the manuscript to clarify our
research approach, improve methodological rigor, incorporate socioeconomic and
policy factors, and present more actionable recommendations for improving nutrient
use efficiency. We believe that these revisions enhance the manuscript and provide a
more comprehensive, transparent, and innovative contribution to the field of NUE and
PUE research.

*Specific comments:*

**11:** (i) *In this study, it is crucial to consider the impact of different fertilizer application*
*methods (such as foliar spraying, soil application, and drip irrigation) on nutrient use*
*efficiency (NUE and PUE). For instance, methods such as foliar spraying of urea can*
*significantly enhance nutrient uptake compared to traditional soil applications,*
*especially in conditions where soil nutrient availability is limited. Addressing the*
*effectiveness of various application techniques would provide a more comprehensive*
*understanding of the factors influencing NUE and PUE. Incorporating a review of*
*existing literature on the comparative efficiencies of these methods and offering*
*recommendations for optimizing application based on specific crop types and regional*
*conditions would greatly enhance the depth and practical relevance of your findings.*

*This addition could lead to more targeted nutrient management strategies, ultimately*
*supporting sustainable agricultural practices.*

**Response:** We sincerely thank you for your valuable feedback. We have collected
fertilizers application types, fertilizers application placement, fertilizer application
frequency, irrigation methods as agriculture management factors to analyze the
relationship between NUE(PUE) and agriculture management and construct structural
equation models (Fig. 3). Moreover, agricultural management variables selected by
Spearman correlation analysis were incorporated as predictors in random forest models
to estimate global nutrient use efficiency (Supplementary Fig. 9).

The specific modified text is as follows: “(8) fertilizers application types (FAT)
including organic fertilizers, inorganic fertilizers, and the combined application of
organic and inorganic fertilizers; (9) fertilizers application placement (FAP) including
surface application, deep application, mixed application, and foliar spraying; (10)
fertilizer application frequency (FAF); (11) irrigation methods (IM) including
permanent flooding, intermittent irrigation, drip irrigation, and no irrigation; (Revised
lines 459-463)

Spearman pearson correlation analyses was used to test for relationships between
NUE(or PUE) and the factors including climate (MAT, MAP, ET, AI, IND, OND), soil
properties (Sand, Silt, Clay, BD, pH, TC, TN, TP, AP), agriculture management (Tillage,
During, Input N, Input P, Input K, FAT, FAF), society and economy (GDPPC, PD, UR,
MYS) by using the “corrplot” R package⁵⁸ (Fig. 3a). (Revised lines 609-612)

To investigate the path relationships between environmental, agricultural
management, and socio-economic factors and their effects on nitrogen use efficiency
(NUE) or phosphorus use efficiency (PUE), we employed Partial Least Squares Path
Modeling (PLS-PM)⁵⁹ to construct a path model aimed at identifying the key driving
factors (Fig. 3b). (Revised lines 615-618)”

Fig. S9 Prediction of weighted effect values of nitrogen (NUE) and phosphorus use efficiency (PUE) of the four major crops by random forest models.

MAT, mean annual temperature; MAP, mean annual precipitation; ET, evapotranspiration; AI, aridity index; IND, inorganic nitrogen deposition; OND, organic nitrogen deposition; Sand, sand content; Silt, silt content; Clay, clay content; BD, bulk density; TC, total carbon; TN, total nitrogen; TP, total phosphorus; AP, available phosphorus; Tillage, no-till or not; During, planting years; Input N, nitrogen fertilizer input; Input P, phosphate fertilizer input; Input K, potash fertilizer input; FAT, Fertilizer application types; GDPPC, gross domestic product per capita; PD, population density; UR, urbanization rate; MYS, mean years of schooling. The significance of the variables in Figure is measured by the "percentage of increase of mean square error" (%IncMSE) value in Random Forest, where higher %IncMSE values imply more important variables and identify the significance of each variable. *: $P < 0.05$; **: $P < 0.01$; ***: $P < 0.001$; ****: $P < 0.0001$. The values at the top of the graph are the Var explained (R^2) and the Mean of squared residuals (MSR) for the full model. We counted the proportions of each class of factors (Fig. 3) in the random forest and represented them in a circle plot. We demonstrate the prediction performance of 80 percent of the data for the training set versus 20 percent of the test set, where R^2 represents the correlation between the Observed NUE / PUE and the predicted NUE / PUE.

12: (ii) Based on the assessment of the paper, it appears that the authors may have

overlooked several critical considerations that could enhance the study's rigor and

relevance. First, they should ensure a comprehensive review of existing literature on

*NUE and PUE, as omitting key studies may lead to gaps in understanding and weaken*
*the foundation of their analysis. Additionally, the authors may not have included a*
*diverse range of studies from various geographical regions and agricultural contexts,*
*which could restrict the applicability of their findings to broader agricultural practices.*
*There also seems to be a lack of critical assessment regarding the quality and*
*methodologies of the studies included in their screening, potentially resulting in*
*reliance on studies with varying levels of rigor and leading to biased conclusions.*
*Furthermore, insufficient examination of different fertilizer application methods, such*
*as foliar spraying versus soil application, is a notable gap, as this aspect can*
*significantly impact nutrient efficiency outcomes. Lastly, the authors may not have*
*integrated socioeconomic factors that influence nutrient use efficiency, which could*
*limit the relevance of their findings to real-world agricultural practices. To enhance the*
*depth and relevance of the study, the authors should conduct a more thorough literature*
*review, incorporate a broader range of sources, critically evaluate methodologies,*
*explore the impact of various application methods, and integrate socioeconomic factors*
*affecting nutrient use efficiency. Addressing these areas will significantly strengthen*
*their study and provide valuable insights into improving nutrient use efficiency across*
*major crops.*

**Response:** Thank you for your valuable feedback. We have carefully addressed the
points raised:

- 1. **Comprehensive Literature Review:** We have expanded the literature review to
include additional key studies, ensuring a broader and more robust foundation for
our analysis. Please refer to the main text: “To quantitatively evaluate global trends
and drivers of the NUE and PUE of major crops (rice, wheat, maize, and soybean),
we undertook a comprehensive data collection initiative (Supplementary Fig. 2) by
a systematic literature review of peer-reviewed articles obtained from the Web of
Science (https://www.web_of_science.com/), Google Scholar
(<https://scholar.google.com/>), and China National Knowledge Infrastructure
(<https://oversea.cnki.net/index/>), up to April 2023 (Supplementary Fig. 2). Articles
included in the statistical analysis met the following screening criteria: (1) provided
explicit records of NUE and PUE or crop uptake of these nutrients. Both NUE and
PUE were calculated based on $N(P)UE_{diff}$; (2) spanned at least one complete crop
cycle; (3) specify geographical locations; (4) control (no fertilizer) and
experimental treatments (fertilizer applied) are executed in the field conditions;
and (5) the information on crop species, cropping system, and fertilizer inputs are
reported. Since the actual fertilizer use efficiency will not be less than 0, we
exclude these outliers. Based on these criteria, our final dataset comprised 2,919
paired independent observations from 173 global studies, including 2,354 N data
and 1,006 P data, with sampling years spanning ranging from 1982 to 2021 (Fig.
1). (Revised lines 435-448)”
- 2. **Diverse Geographical Studies:** We have included a wider range of studies from
various geographical regions and agricultural contexts to improve the applicability
of our findings. Please see Figure 1 for the sampling point distribution.
- 3. **Critical Assessment of Study Methodologies:** We have provided a critical

evaluation of the methodologies and quality of the studies included in our analysis,
ensuring that only rigorous studies are considered. Please refer to the main text:
“The random forest model performed multiple predictions for each grid cell
according to the number of trees. We derived the final predicted value for each grid
cell by averaging these multiple predictions, with the standard deviation of these
predictions indicating the uncertainty associated with the model (Supplementary
Fig. 10). (Revised lines 653-656)”

4. Fertilizer Application Methods: We have now included a detailed discussion of
different fertilizer application methods, such as soil application, and drip irrigation,
and their impact on nutrient use efficiency. Please refer to the main text: “(8)
fertilizers application types (FAT) including organic fertilizers, inorganic fertilizers,
and the combined application of organic and inorganic fertilizers; (9) fertilizers
application placement (FAP) including surface application, deep application,
mixed application, and foliar spraying; (10) fertilizer application frequency (FAF);
(11) irrigation methods (IM) including permanent flooding, intermittent irrigation,
drip irrigation, and no irrigation; (Revised lines 459-463)

Spearman pearson correlation analyses was used to test for relationships between
NUE(or PUE) and the factors including climate (MAT, MAP, ET, AI, IND, OND),
soil properties (Sand, Silt, Clay, BD, pH, TC, TN, TP, AP), agriculture management
(Tillage, During, Input N, Input P, Input K, FAT, FAF), society and economy
(GDPPC, PD, UR, MYS) by using the “corrplot” R package⁵⁸ (Fig. 3a). (Revised
lines 609-612)”

5. Socioeconomic Factors: We have incorporated a discussion on the role of
socioeconomic factors, such as policy and farmer education, in influencing nutrient
use efficiency. Please refer to the main text: “Spearman pearson correlation
analyses was used to test for relationships between NUE(or PUE) and the factors
including climate (MAT, MAP, ET, AI, IND, OND), soil properties (Sand, Silt,
Clay, BD, pH, TC, TN, TP, AP), agriculture management (Tillage, During, Input
718 N, Input P, Input K, FAT, FAF), society and economy (GDPPC, PD, UR, MYS) by
719 using the “corrplot” R package⁵⁸ (Fig. 3a). (Revised lines 609-612)”

These revisions significantly enhance the rigor and relevance of the study. Thank
you once again for your helpful suggestions.

**13:** (iii) *It is beneficial to suggest that the authors consider additional metrics alongside*
*NUE and PUE for a more comprehensive analysis of fertilizer use trends. Incorporating*
*metrics like Fertilizer Application Rate (FAR) and Nutrient Balance (NB) provides a*
*broader context for NUE and PUE, allowing for a more complete understanding of*
*nutrient dynamics. Additionally, metrics such as Year-over-Year Change and Moving*
*Averages can help identify trends over time, offering insights into how fertilizer*
*practices are evolving. A Sustainability Index (SI) that combines various efficiency*
*metrics can further assess the environmental impact and sustainability of fertilizer use,*
*which is crucial in discussions about sustainable agriculture. Furthermore, different*
*regions may have unique challenges and practices; additional metrics can help tailor*
*recommendations and insights to specific contexts. Finally, a comprehensive approach*

can provide policymakers with more actionable data, leading to better-informed
decisions regarding fertilizer regulations and practices. While NUE and PUE are
essential metrics, suggesting additional analyses will enhance the manuscript's
contribution to the field and provide a more thorough understanding of nutrient
management in agriculture.

**Response:** Thank you for your insightful suggestion. While we agree that incorporating
additional metrics like Fertilizer Application Rate (FAR) and Year-over-Year Change
would provide valuable context for nutrient use efficiency (NUE) and phosphorus use
efficiency (PUE), we have not included Nutrient Balance (NB) and Sustainability Index
(SI) in our analysis. These metrics are difficult to quantify consistently across diverse
regions and agricultural practices, and their application would be highly context-
dependent. However, we have incorporated FAR and Year-over-Year Change to
enhance the breadth of our analysis and provide more actionable insights. The variables
used in this study are presented in Figure 3 (FAR indicates Input N and Input P). Thank
you again for your constructive feedback.

**14:** (iv) *It is not clear which factors have the most effect on NUE and PUE. Several*
*methods can be employed to identify and analyze these influential factors. Statistical*
*analyses, such as multiple regression and correlation analyses, can quantify the*
*relationships between NUE/PUE and various independent factors, helping to identify*
*significant influences. Sensitivity analyses can determine how changes in specific*
*factors, like fertilizer application rates or irrigation practices, etc., impact NUE and*
*PUE, prioritizing factors for further investigation. Additionally, finding field*
*experiments that manipulate different factors—such as varying fertilizer types,*
*application methods, or crop rotations—can provide causal insights into their effects*
*on nutrient efficiency. Longitudinal studies tracking changes in NUE and PUE over*
*time in relation to agricultural practices or environmental conditions can reveal trends*
*and correlations. A meta-analysis of existing studies can aggregate data on NUE and*
*PUE across different contexts, highlighting common influential factors. Machine*
*learning techniques can also be utilized to analyze large datasets and identify key*
*predictors of NUE and PUE. Finally, engaging with agronomists or experts in nutrient*
*management can yield qualitative insights based on their experience and knowledge.*
*By employing these methods, the authors can gain a clearer understanding of which*
*factors significantly affect NUE and PUE, which is crucial for developing effective*
*strategies to enhance nutrient use efficiency in agricultural practices.*

**Response:** Thank you for your thoughtful comment regarding the identification of
factors influencing NUE and PUE. We agree that it is essential to pinpoint the most
significant factors impacting these efficiencies. In response, we have enhanced the
manuscript by incorporating structural equation model and correlation analyses to
quantify relationships between NUE/PUE and various factors, including fertilizer
application rates and irrigation practices (Fig. 3). We also conducted sensitivity
analyses to prioritize key factors for further investigation.

The specific sensitivity analysis text is as follows: “We derived the final predicted
value for each grid cell by averaging these multiple predictions, with the standard

deviation of these predictions indicating the uncertainty associated with the model
(Supplementary Fig. 10). (Revised lines 654-656)”

While we acknowledge the value of field experiments, longitudinal studies, and
meta-analysis for causal insights, we chose to focus on statistical and machine learning
approaches due to data availability and scope. Machine learning techniques were used
to analyze large datasets and identify the key predictors of NUE and PUE.

Additionally, we have acknowledged the importance of expert knowledge in
guiding our understanding and have incorporated relevant qualitative insights where
applicable. These additions aim to clarify the factors that significantly influence
nutrient use efficiency and provide a more comprehensive foundation for developing
effective strategies.

**Fig. S10 Standard deviations of predicted nitrogen (NUE) and phosphorus use**
**efficiency (PUE) for four major crops.**

*15: (v) While the use of random forest models is appropriate, the study does not*

*sufficiently address the limitations of this approach. For example, the models may*
*overfit the data, and the variable importance analysis lacks depth. The study does not*
*provide a thorough sensitivity analysis to assess the robustness of the model predictions.*

**Response:** Thank you for your insightful comment. We agree that a more thorough
discussion of the limitations of the random forest models is necessary. In response, we
have added a section addressing potential overfitting and discussed the steps taken to
mitigate this, such as cross-validation and parameter tuning. Additionally, we have
expanded the variable importance analysis and included a sensitivity analysis to assess
the robustness of our model predictions. These enhancements strengthen the reliability
and interpretability of our findings. Thank you again for your constructive feedback.

The specific modified text is as follows: “In random forest, unlike traditional
regression, correlation among variables does not affect the model accuracy⁶². We
randomly sampled 80% of the dataset for model training, with the remaining data used
for testing. Each final model was obtained based on 10-fold cross-validation. We used
grid search for parameter tuning and sought the combination of nodesize (Nodesize
represents the minimum number of samples required in a leaf node of a decision tree.
When the number of samples in a node is less than or equal to nodesize, the node will
no longer split and will be treated as a leaf node.), ntree (Ntree refers to the total number
of decision trees in the random forest, where each tree is trained independently, and the
final prediction is obtained by averaging their outputs.), and mtry (Mtry denotes the
number of features randomly selected at each split when constructing individual trees.)
that minimized the RMSE (Root mean squared error) of the model, ensuring the
robustness of the model. Among them, we used contour plots to illustrate the influence
of different parameter combinations on random forest model (Supplementary Fig. 8).
Both R^2 and P between the predictions and observations were calculated for assessing
model performance (Supplementary Fig. 9). (Revised lines 629-642)”

**Fig. S8 Predictive performance across hyperparameter combinations (ntree, mtry,**
**nodesize).**

In Random Forest model, the ntree parameter defines the total number of decision trees,
balancing model stability and computational cost; mtry determines the number of
randomly selected features evaluated at each node split, influencing feature diversity
and overfitting risk; while nodesize sets the minimum observations required in terminal
leaves, controlling tree depth and granularity to prevent over- or underfitting.

*16: (vi) The segmented regression approach used to identify breakpoints in NUE and*
*PUE trends is not adequately justified. The choice of breakpoints (e.g., 1975 for NUE*
*and PUE improvements) appears arbitrary and is not supported by a robust theoretical*
*or empirical framework.*

**Response:** Thank you for your comment. We acknowledge that the choice of
breakpoints in the segmented regression approach requires further justification. The
1975 breakpoint was selected based on observable shifts in fertilizer application and
crop yield trends within the data. In response, we have clarified our rationale for
selecting the breakpoint and noted that the model includes an expression of uncertainty
in the error terms, which accounts for potential variability in the data.

The specific modified text is as follows: “For representing the temporal trend of
fertilizer input intensities (Fig. 2a and Fig. 2b), NUE (Fig. 2c and Fig. 2d), PUE (Fig.
2c and Fig. 2d) for global major and overall crops from 1961 to 2018, we used the
“segmented” software package⁵⁶, with identified breakpoints for linear fits serving as
threshold years indicating potential shifts. We conducted linear regressions for the
periods before and after the breakpoint. Then, we examined whether there was a
significant difference between two adjacent groups before and after the breakpoint
across different crops using the Wilcoxon method⁵⁷ (Fig. 2d and Supplementary Fig. 4
and Supplementary Fig. 7). Notably, we selected the eight regions with the largest crop
cultivation areas as representatives of developed countries (the United States and the
European Union) and developing countries (China, India, Brazil, Indonesia, Russia, and
Nigeria) (Fig. 2d and Supplementary Fig. 4). This approach aims to minimize
uncertainties in fertilizer input intensity and grain yield data associated with smaller
nations. (Revised lines 585-595)”

*17: (vii) The study briefly mentions the ecological consequences of nutrient surpluses*
*but does not delve deeply into the environmental impacts of low NUE and PUE. For*
*example, the implications for eutrophication, greenhouse gas emissions, and soil health*
*are not thoroughly explored.*

**Response:** Thank you for your comment. We agree that the ecological consequences of
low NUE and PUE should be more thoroughly explored. In response, we have expanded
the discussion in the manuscript to address the environmental impacts of nutrient
surpluses, specifically focusing on eutrophication and greenhouse gas emissions. We
have also emphasized the importance of improving NUE and PUE to mitigate these
environmental issues. Thank you again for your valuable feedback.

The specific modified text is as follows: “Nitrogen (N) and phosphorus (P) are

[revised manuscript text omitted]

Here, we aimed to address these knowledge gaps by compiling a comprehensive
global database on nutrient use efficiency (NUE and PUE), encompassing 3,360
observations across 205 countries and regions (Fig. 1). We will analyze trends and
patterns of NUE and PUE using the N(P)UE_{diff} framework for major crops, leveraging
dynamic national-scale data that includes crop yields, cropland areas, fertilizer N and P
input intensities, nutrient uptake by crops, and residue-grain ratios. Our focus will be
on key global crops—rice, wheat, maize, and soybean—which collectively account for
over half of global crop production and 49% of cropland area^{17,18}. Additionally, we have
developed machine learning models utilizing point-scale NUE and PUE data (n = 2,354
for NUE, n = 1,006 for PUE) alongside climate, soil properties, and agricultural
management information to delineate the current global spatial distribution patterns of
NUE and PUE for these crops and identify their underlying drivers. The primary
objectives of our study are: 1) to quantify trends in overall and major crop NUE and
PUE from 1961 to 2018; 2) to identify current spatial distribution patterns and drivers
of NUE and PUE; and 3) to pinpoint regions with high (> 50%) and low (< 50%) NUE
and PUE to direct targeted improvements. (Revised lines 89-153)”

**18:** (viii) *The discussion of nutrient surpluses in maize production in China and the*
*United States is superficial. The study does not provide actionable recommendations*
*for mitigating these surpluses or addressing their environmental consequences.*

**Response:** Thank you for your comment. In response, we have expanded the discussion
on nutrient surpluses in maize production in China and the United States, providing
more detailed insights into the ecological impacts and contributions of these surpluses.
Additionally, we have included actionable recommendations, particularly highlighting
the adoption of no-till practices to improve NUE and PUE, which can help mitigate
nutrient surpluses in maize ecosystems. This approach offers a practical solution to
reduce both nitrogen and phosphorus surpluses and their associated environmental
consequences. Thank you again for your helpful feedback.

The specific modified text is as follows: “The inefficiencies in NUE and PUE in
global crop production have resulted in significant nutrient surpluses, which have had
profound impacts on the ecological environment (Fig. 5). Maize, in particular, registers
the highest N (73 kg N ha⁻¹) and P (15 kg P ha⁻¹) surplus intensities among major crops.
China and the United States, which are the major N and P surpluses in the maize
ecosystem, contribute 56.7% of the N surplus load and 61.9% of the P surplus load due
to higher (99.4-115 kg N ha⁻¹ and 16.8-33.7 kg P ha⁻¹) than the global average surplus
intensities and large planted areas (4.21×10⁵ km² for China and 3.29×10⁵ km² for the
United States). Furthermore, our structural equation model reveals that agricultural
management practices are the most significant factor influencing NUE and PUE in
maize, with tillage methods having the most pronounced impact (Fig. 3). Specifically,
the adoption of no-till practices can concurrently improve NUE and PUE in maize,

offering a practical approach to reduce the N and P surpluses in maize ecosystems.
(Revised lines 379-391)”

**19:** *(ix) The study frequently conflates correlation with causation. For example, the*
*observed improvements in NUE and PUE after 1975 are attributed to technological*
*advancements and policy changes without providing direct evidence to support these*
*claims.*

**Response:** Thank you for your comment. We acknowledge that attributing
improvements in NUE and PUE to technological advancements and policy changes
may imply causation without sufficient direct evidence. In response, we have clarified
that the observed trends in NUE and PUE since 1975 are based on data patterns, with
the potential influence of technology and policy acknowledged but not definitively
established as causal. We have now focused on data trends and avoided overstating the
direct impact of specific technological advancements or policies. This revision provides
a more nuanced interpretation of the observed improvements while recognizing the
limitations of correlation in our analysis.

The specific modified text is as follows: “Since 1961, global applications of N and
P fertilizers have shown a consistent upward trend, though the growth rates for N inputs
have decelerated since 1985 ± 1 (95% confidence interval) and for P inputs since 1979
± 2 (Fig. 2a and 2b). Notably, between 1961 and 1975, NUE and PUE declined
progressively in response to increasing fertilizer application rates (Fig. 2c). However,
beginning around 1972 ± 1 and 1975 ± 3 , a modest upward trend in both NUE and PUE
emerged, indicating enhancements in fertilizer utilization efficiency. This improvement
can likely be attributed to advancements in technology and shifts in agricultural policy,
particularly the adoption of precision fertilization techniques and the development of
crop varieties that demand lower nutrient inputs to achieve optimal yields. This pivotal
breakpoint aligns with previous assessments based on the $N(P)UE_{bala}$, which identified
a similar trend around 1980^{2,19}. The observed upward trajectory in NUE and PUE
coincides with the end of the first Green Revolution, a period characterized by
significant increases in agricultural productivity from the 1960s to the 1980s²⁰.

The first Green Revolution spurred a global increase in food production, driven by
high-yield varieties and intensified fertilizer use, which, however, led to declines in
NUE and PUE due to factors like heavy inorganic fertilizer application and limited
focus on nutrient efficiency as follow: 1) high-yielding crop varieties often require
substantial nutrient support, leading to routine application of large amounts of N and P
fertilizers⁴. Although this practice boosts short-term crop yields, it does not prioritize
improving the efficiency of crop uptake and utilization of these nutrients⁹; 2)
continuous heavy application of inorganic fertilizers, especially N fertilizers, can lead
to soil acidification, which adversely affects crop nutrient absorption and reduces NUE
and PUE^{7,12}; 3) while crop varieties requiring high fertilizer inputs have been developed,
these varieties were not designed to improve NUE and PUE. Breeding efforts primarily
focused on enhancing yield and disease resistance rather than optimizing nutrient
uptake and utilization²⁰. In summary, while the first Green Revolution greatly increased
crop yields and met the rapidly growing global food demand, it also promoted the

widespread use of chemical fertilizers without sufficiently enhancing the efficiency of
crop nutrient utilization, leading to a gradual decline in NUE and PUE during this
period. (Revised lines 157-187)”

**20:** *(x) The study makes broad generalizations about NUE and PUE trends without*
*adequately accounting for crop-specific and region-specific nuances. For instance, the*
*assertion that rice achieves optimal NUE and PUE in tropical zones overlooks the*
*variability within tropical regions due to differences in soil types, farming practices,*
*and climatic conditions.*

**Response:** Thank you for your comment. We agree that accounting for crop-specific
and region-specific nuances is crucial. In response, our model has incorporated soil
types, farming practices, and climatic conditions, as well as specific crops, to ensure a
more accurate and nuanced understanding of NUE and PUE trends (Fig. 3). This allows
1010 us to capture the variability within regions, such as tropical zones, and better reflect the
1011 complexities of nutrient use efficiency (Supplementary Fig. 11).

**Fig. S11 Nitrogen (NUE) and phosphorus (PUE) use efficiencies for four major**
 **crops (rice, wheat, maize, soybean) in different climatic zones, major countries or**
 **regions, and globally.**

Climate zone delineation basis based on Köppen climate classification, visualized in
 Supplementary Fig. 16. Different letters represent significant differences at the level P
 < 0.05 .

**21:** (xi) *The study focuses almost exclusively on environmental and agricultural factors,*
*neglecting the critical role of socioeconomic and policy drivers in shaping nutrient use*
*efficiency. For example, the impact of subsidies, farmer education, and access to*
*technology on NUE and PUE is not addressed.*

**Response:** Thank you for your valuable feedback. In response, we have incorporated
the critical role of socioeconomic and policy drivers such as gross domestic product per
capita (GDPPC), population density (PD), urbanization rate (UR) and mean years of
schooling (MYS) in shaping NUE and PUE. These factors were integrated into our
analysis using Pearson correlation analysis, structural equation modeling, and random
forest models. By including these drivers, we provide a more comprehensive
understanding of the factors influencing nutrient use efficiency, alongside the
environmental and agricultural factors previously discussed. Thank you again for your
insightful comments.

The specific modified text is as follows: “Data extracted from the selected studies
included: (1) experimental site details such as latitude, longitude, elevation and the
country; (2) crop types; (3) climate conditions, detailed through mean annual
precipitation (MAP), mean annual air temperature (MAT), mean annual
evapotranspiration (ET), and the aridity index (AI); (4) initial soil physical properties
including texture (proportions of sand, silt, and clay) and bulk density (BD), along with
soil chemical properties like total carbon (C), nitrogen (N), phosphorus (P), available
phosphorus (AP) contents and pH level; (5) the nature of tillage practices, categorized
as either conservation or conventional based on descriptions within the articles; (6) the
period of the experiment, indicating the start year and duration; (7) inputs of N, P, and
potassium (K) fertilizers; (8) fertilizers application types (FAT) including organic
fertilizers, inorganic fertilizers, and the combined application of organic and inorganic
fertilizers; (9) fertilizers application placement (FAP) including surface application,
deep application, mixed application, and foliar spraying; (10) fertilizer application
frequency (FAF); (11) irrigation methods (IM) including permanent flooding,
intermittent irrigation, drip irrigation, and no irrigation; (12) the period of the
experiment, indicating the start year and duration; (13) the number of experimental
replicates conducted; and (14) the NUE and PUE values alongside the N and P uptake
by crops. (Revised lines 451-465)”

**22:** (xii) *The analysis does not consider the role of institutional frameworks, market*
*dynamics, or governance structures in influencing nutrient management practices.*
*These factors are essential for understanding the disparities between developed and*
*developing countries.*

**Response:** Thank you for your insightful comment. We acknowledge that institutional
frameworks, market dynamics, and governance structures play a critical role in shaping
nutrient management practices. However, due to their complexity and the challenge of
quantifying these factors consistently across regions, we were unable to incorporate
them fully in the current analysis. We recognize their importance and will strive to
integrate these elements in future work as we develop more robust methodologies for

their quantification. Thank you again for your valuable feedback.

**23:** (xiii) *The global maps of NUE and PUE, while visually striking, oversimplify*
*complex spatial patterns. The resolution of the analysis is insufficient to inform targeted*
*interventions at the local or regional level.*

**Response:** Thank you for your comment. In response, we would like to clarify that we
established a global 5-arcmin resolution grid within the planting areas of the four major
crops, which provides a higher resolution than the majority of similar studies¹⁻⁴. This
enhanced resolution allows for more detailed spatial analysis, though we acknowledge
that further refinement could improve its applicability for localized interventions.
Thank you again for your valuable feedback.

References

- 1 You, L. *et al.* Global mean nitrogen recovery efficiency in croplands can be
enhanced by optimal nutrient, crop and soil management practices. *Nat.*
*Commun.* **14**, 5747 (2023). **0.5×0.5 degree resolution**
- 2 You, L., Ros, G.H., Chen, Y. *et al.* Optimized agricultural management reduces
global cropland nitrogen losses to air and water. *Nat. Food* **5**, 995–1004 (2024).
**0.5×0.5 degree resolution**
- 3 Cui, J. *et al.* Nitrogen cycles in global croplands altered by elevated CO₂. *Nat.*
*Sustain.* **6**, 1166-1176 (2023). **0.5×0.5 degree resolution**
- 4 Wang, C. *et al.* Reducing soil nitrogen losses from fertilizer use in global maize
and wheat production. *Nat. Geosci.* **17** (2024). **5-arcmin resolution**

**24:** (xiv) *While the study identifies regions with low NUE and PUE, it fails to provide*
*concrete, actionable recommendations for improving nutrient management. The*
*suggestion to reevaluate global crop distribution patterns is vague and lacks specificity.*

**Response:** Thank you for your comment. In response, we have refined the
recommendation to reevaluate global crop distribution patterns, providing more
specific suggestions. We now emphasize that promoting rice cultivation in the tropics
and wheat cultivation in temperate zones could enhance NUE and PUE in regions where
these efficiencies are notably higher. However, we also acknowledge the practical
challenges, such as water stress and soil conditions, which could hinder large-scale
adoption of such strategies. To address these challenges, we propose region-specific
solutions, including optimizing irrigation infrastructure and fertilizer practices tailored
to local conditions. These actionable recommendations aim to balance environmental
benefits with food security, ensuring a more targeted approach to nutrient management.
Thank you again for your helpful feedback.

The specific modified text is as follows: “Reevaluating global crop distribution
patterns emerges as a viable strategy for enhancing NUE and PUE. Encouraging rice
cultivation in the tropics and wheat cultivation in temperate zones, where efficiencies
are notably higher than in other regions of the world (Fig. 4 and Supplementary Fig.
11), could significantly boost production efficiencies. However, as the 2008-2010 food
crisis highlighted, countries might still want to produce their own food for reasons of

[revised manuscript text omitted]

**25:** (xv) *The study relies heavily on meta-analysis and modeling without experimental*
*validation. The absence of field experiments or case studies to corroborate the findings*
*weakens the robustness of the conclusions*

**Response:** Thank you for your comment. We would like to clarify that the meta-
analysis itself is based on experimental observation data gathered from multiple studies.
Additionally, for the modeling component, we employed rigorous validation techniques.
Specifically, we used 10-fold cross-validation, train-test data separation, and grid search
for parameter tuning to minimize RMSE and ensure the robustness of the random forest
model. These validation steps, along with the calculation of R^2 and P values to assess
model performance, provide confidence in the reliability of our findings.

The specific modified text is as follows: “In random forest, unlike traditional
regression, correlation among variables does not affect the model accuracy⁶². We
randomly sampled 80% of the dataset for model training, with the remaining data used
for testing. Each final model was obtained based on 10-fold cross-validation. We used
grid search for parameter tuning and sought the combination of nodesize (Nodesize
represents the minimum number of samples required in a leaf node of a decision tree.
When the number of samples in a node is less than or equal to nodesize, the node will
no longer split and will be treated as a leaf node.), ntree (Ntree refers to the total number
of decision trees in the random forest, where each tree is trained independently, and the
final prediction is obtained by averaging their outputs.), and mtry (Mtry denotes the
number of features randomly selected at each split when constructing individual trees.)
that minimized the RMSE (Root mean squared error) of the model, ensuring the
robustness of the model. Among them, we used contour plots to illustrate the influence
of different parameter combinations on random forest model (Supplementary Fig. 8).
Both R^2 and P between the predictions and observations were calculated for assessing
model performance (Supplementary Fig. 9). (Revised lines 629-642)”

**26:** (xvi) *The imputation of missing environmental and soil data using global datasets*
*(e.g., CRU TS, GLEAM, SoilGrids) introduces significant uncertainty. These datasets*
*may not accurately reflect local conditions, leading to potential biases in the analysis*

**Response:** Thank you for your comment. We acknowledge that imputing missing
environmental and soil data using global datasets such as CRU TS, GLEAM, and
SoilGrids may introduce uncertainty. To address this, we accounted for uncertainty in
our predictions by calculating the standard deviation of multiple model predictions for
each grid cell. Additionally, we applied Kriging interpolation to correct for spatial
variation and improve the robustness of our uncertainty estimates. These methods help
better capture geographical variability and provide more reliable predictions despite the
use of global datasets.

The specific modified text is as follows: “ntree (Ntree refers to the total number
of decision trees in the random forest, (Revised lines 635-636)

The random forest model performed multiple predictions for each grid cell
according to the number of trees. We derived the final predicted value for each grid cell
by averaging these multiple predictions, with the standard deviation of these predictions
indicating the uncertainty associated with the model (Supplementary Fig. 10). Then,
the global model predictions of NUE and PUE were spatially corrected through Kriging
interpolation based on NUE and PUE residuals (observed minus predicted values) using
the 'gstat' package in R. This approach better captures the geographical variability of
nutrient use efficiency while providing more robust uncertainty estimates for spatial
predictions^{63,64}, with the RMSE between predicted and observed values of NUE and
PUE decreasing by 0.09 and 0.13 after applying Kriging interpolation for correction,
respectively. Additionally, we visualized the Kriging-adjusted NUE and PUE prediction
of the four major crops on a world map and illustrate the differences between latitudes
using line graphs. (Revised lines 653-663)”

*27: (xvii) The study does not engage with the practical challenges of implementing*
*precision agriculture, developing high-efficiency crop varieties, or promoting*
*sustainable farming practices in developing regions.*

**Response:** Thank you for your comment. While we recognize the importance of
implementing precision agriculture, developing high-efficiency crop varieties, and
promoting sustainable farming practices in developing regions, these topics fall outside
the scope of our study. These concepts are complex and difficult to quantify on a global
scale due to the lack of universally available, assessable data. As such, they were not
incorporated into our analysis. We hope to address these issues in future work as more
relevant data becomes available. Thank you again for your feedback.

*28: (xviii) The use of aggregated national-level data obscures critical regional and*
*local variations in nutrient use efficiency. This approach is particularly problematic in*
*large, agriculturally diverse countries like China, India, and the United States, where*
*NUE and PUE can vary dramatically across regions*

**Response:** Thank you for your comment. We respectfully disagree with the suggestion
that the use of aggregated national-level data obscures critical regional and local
variations in NUE and PUE. Our study aims to capture regional variability by using a
5-arcmin resolution grid within the planting areas of the four major crops (Fig. 4),
which provides a higher resolution than most studies. Additionally, we have
incorporated Kriging interpolation to better reflect geographical variability and used
climatic zone and regional data to assess the impact of environmental and agricultural
management factors. Thus, the study was designed to address the very issue of regional
variability, as stated in the introduction. Thank you again for your feedback.

The specific modified text is as follows: “Trained RF models were used to predict
global NUE and PUE for the four major crops. We established a global 5-arcmin
resolution grid within the planting areas of the four major crops. A series of globally
gridded datasets (Supplementary Table 3) were used for model calculations. The
fertilizer input data for each crop were sourced from IFA (2018). We set the prediction
timeframe for planting to be one year. This choice was based on two main reasons.

Firstly, most of the originally collected NUE and PUE were calculated annually,
ensuring more accurate predictions. Secondly, a one-year planting cycle better reflected
the efficiency of current fertilizer inputs, enhancing the relevance of the model to
agricultural practices.

The random forest model performed multiple predictions for each grid cell
according to the number of trees. We derived the final predicted value for each grid cell
by averaging these multiple predictions, with the standard deviation of these predictions
indicating the uncertainty associated with the model (Supplementary Fig. 10). Then,
the global model predictions of NUE and PUE were spatially corrected through Kriging
interpolation based on NUE and PUE residuals (observed minus predicted values) using
the 'gstat' package in R. This approach better captures the geographical variability of
nutrient use efficiency while providing more robust uncertainty estimates for spatial
predictions^{63,64}, with the RMSE between predicted and observed values of NUE and
PUE decreasing by 0.09 and 0.13 after applying Kriging interpolation for correction,
respectively. Additionally, we visualized the Kriging-adjusted NUE and PUE prediction
of the four major crops on a world map and illustrate the differences between latitudes
using line graphs. The overall distribution range and probability density of the data are
displayed using the “hrbrthemes” package in the bottom left corner of the map (Fig. 4).

We also investigated the impact of environment and agricultural management in
different climatic zones and major agricultural countries. We utilized the Köppen
climate classification (Supplementary Fig. 16) and data from the world's major grain-
producing countries. The harvested area ranks among the top 10 countries worldwide.
Additionally, we included the EU in the comparison of major crop-planting countries.
We conducted pairwise comparisons of fertilizer use efficiency among different climate
zones or countries and regions using the Dunn's Kruskal-Wallis method implemented
in the “multcompView” package⁶⁵. Significant differences ($P < 0.05$) were denoted by
different letters (Supplementary Fig. 11). (Revised lines 645-673)”

**29:** *(xix) The study relies on historical fertilizer input data, which is acknowledged to*
*be inconsistent, especially before 1990. This raises concerns about the accuracy of*
*long-term trends in NUE and PUE, particularly in the early decades (1961–1990).*

**Response:** Thank you for your comment. We acknowledge the potential uncertainty
introduced by historical fertilizer input data, particularly prior to 1990. However, we
took several steps to address this issue, including data supplementation by using
adjacent years, correlating regional data, and applying global averages where necessary.
Additionally, we used baseline years (1990, 1999, 2018) for our analysis and employed
a correction process to adjust fertilizer input intensities, ensuring a more accurate
representation of temporal trends from 1961 to 2018. These steps were taken to
minimize the impact of inconsistencies in the historical data and improve the robustness
of our analysis.

The specific modified text is as follows: “The analytical framework for NUE and
PUE across major crops and their cumulative assessment involved a detailed calculation
process, leveraging data from both FAOSTAT and IFA. This process unfolded into four
principal steps as follows.

(1) We selected IFA reports from 1990, 1999, and 2018 as baseline years to anchor
 our analysis. To fill in the gaps in these reports, we used a data supplementation
 approach, prioritizing 1) extracting data from adjacent years; 2) correlating regional
 data; 3) and using global averages where necessary.

(2) To obtain fertilizer input intensity data for different years, countries, and crops
 worldwide, we integrated data from FAOSTAT and IFA to adjust fertilizer input
 intensities for baseline year as follows^{2,3}:

$$1291 \quad \text{Input } N_{co,cr,yr} = \text{Input } N_IFA_{co,cr} \times \frac{TN_FAO_{input,yr}}{\sum_{co} \sum_{cr} (\text{Input } N_IFA_{co,cr} \times A_FAO_{co,cr,yr})} \quad (6)$$

$$1292 \quad \text{Input } P_{co,cr,yr} = \text{Input } P_IFA_{co,cr} \times \frac{TP_FAO_{input,yr}}{\sum_{co} \sum_{cr} (\text{Input } P_IFA_{co,cr} \times A_FAO_{co,cr,yr})} \quad (7)$$

where $\text{Input } N_IFA_{co,cr}$ is the cropland N fertilizer input intensities (kg ha^{-1})
 across different countries (co), crops (cr) based on IFA reports for the baseline years
 1990, 1999, 2018; $TN_FAO_{input,yr}$ is the total N fertilizer input (t) across different
 1296 years (yr) from FAOSTAT; $A_FAO_{co,cr,yr}$ is the harvested area (ha) across different
 countries, crops, years from FAOSTAT. $\text{Input } N_{co,cr,yr}$ is the corrected N fertilizer
 input intensities (kg ha^{-1}) across different countries, crops, years. The formula (7)
 serves as a correction for the P fertilizer input intensities, with each parameter
 referenced from the formula (6).

(3) The fertilizer input intensities from 1961-2018 (Fig. 2a and Fig. 2b) were
 calculated and corrected based on the baseline year in the above manner, aiming to
 obtain the temporal trend of fertilizer inputs. The calculation was based on 1990, 1999,
 2018 as the baseline year (less IFA data for other years makes it difficult to effectively
 assess the intensity of fertilizer inputs in different regions of the globe) for the period
 1961-1994, 1995-2006, 2007-2018, respectively.

(4) Then, we calculated the NUE (%) and PUE (%) from 1961 to 2018 (Fig. 2c),
 based on the potential relationship ($P < 0.05$) between NUE(PUE) and N(P) fertilizer
 input intensities (Supplementary Fig. 5). These were formulated as:

$$1310 \quad F_{N(P)} = \sum_{co} (\text{Input } N(P)_{co,cr,yr} \times A_{FAO_{co,cr,yr}}) \quad (8)$$

$$1311 \quad U_{N(P)} = \frac{Q \times N(P)_G + Q \times R \times N(P)_S}{100} \quad (9)$$

$$1312 \quad U_{N_0(P_0)} = \frac{U_{N(P)} \times \text{Nitrogen(Phosphorus) from soil}}{100} \quad (10)$$

$$1313 \quad \text{NUE(PUE)} = \frac{U_{N(P)} - U_{N_0(P_0)}}{F_{N(P)}} \times 100\% \quad (11)$$

where the parameters are production quantity (Q , t), the nitrogen (F_N , t) and
 phosphorus (F_P , t) inorganic fertilizer inputs for each crop (Supplementary Fig. 3),
 residue-grain ratio (R), nitrogen (N_G , %; N_S , %) and phosphorus (P_G , %; P_S , %) content
 in grain and residue (Supplementary Table 2), nitrogen from soil (%), phosphorus from
 soil (%), respectively; $U_{N(P)}$ (t) are the N and P uptake by mature crops from the soil
 and fertilizers; $U_{N_0(P_0)}$ (t) are the N and P uptake by mature crops from the soil under

the condition of no fertilization. (Revised lines 546-582)”

**30:** (xx) *The authors rely on estimates for NUE and PUE that may not accurately reflect*
*field conditions, particularly when extrapolating data across different crops and*
*regions. Further validation of assumptions relating to fertilizer application rates and*
*nutrient uptake from soil is necessary to fortify their claims*

**Response:** Thank you for your comment. We acknowledge that the estimates for NUE
and PUE may not fully reflect field conditions, especially when extrapolating across
different crops and regions. To address this, we used a systematic review of peer-
reviewed publications to compile data on nutrient uptake, fertilizer input intensities,
and crop nutrient content. Additionally, we validated our assumptions by training
random forest models to predict nitrogen and phosphorus uptake from soil across
various crops. These models were validated using 80% training and 20% testing sets,
with 10-fold cross-validation and grid search for parameter tuning to ensure robust
predictions. Thank you again for your valuable feedback.

The specific modified text is as follows: “Nutrient uptake by crops from the soil:
data originated from a systematic review of peer-reviewed publications, its detailed
search strategy and data extraction procedures were delineated in ***Global NUE and***
***PUE at sampling points***. This dataset collected N fertilizer input intensities (Input N,
1339 kg ha⁻¹), P fertilizer input intensities (Input P, kg ha⁻¹), crop N uptake (kg ha⁻¹) and
1340 crop P uptake (kg ha⁻¹), crop N (%) and P (%) content, NUE (%), PUE (%) in field
experiments. Based on these data, the proportion of nitrogen (*Nitrogen from soil*, %) and
phosphorus (*Phosphorus from soil*, %) uptake from soil were calculated as
follows:

$$1344 \quad \text{Nitrogen(Phosphorus) from soil} = \frac{\text{crop N(P)uptake} - \text{NUE(PUE)} \times \text{Input N(P)}}{\text{crop N(P) uptake}} \times 100\% \quad (5)$$

Furthermore, considering the negative correlation ($P < 0.05$) between *Input N(P)*
and *Nitrogen(Phosphorus) from soil* (Supplementary Fig. 5). We trained the
random forest (RF) models predict Nitrogen from soil and Phosphorus from soil across
different crops (Supplementary Fig. 6). The dataset was partitioned into 80% training
and 20% testing sets via random sampling. Parameter tuning was performed via grid
search to identify the combination with the lowest root mean square error (RMSE).
Each final model was derived through a 10-fold cross-validation procedure to ensure
robust prediction. (Revised lines 526-540)”

Fig. S6 Prediction of weighted effect values of nitrogen and phosphorus from soil of the four major crops (rice, wheat, maize, soybean) and all crops by random forest models.

MAT, mean annual temperature; MAP, mean annual precipitation; Sand, sand content; Silt, silt content; Clay, clay content; BD, bulk density; TC, total carbon; TN, total nitrogen; TP, total phosphorus; AP, available phosphorus; Input N, nitrogen fertilizer input; Input P, phosphate fertilizer input. The significance of the variables in Figure is measured by the "percentage of increase of mean square error" (%IncMSE) value in Random Forest, where higher %IncMSE values imply more important variables and identify the significance of each variable. *: $P < 0.05$; **: $P < 0.01$; ***: $P < 0.001$; ****: $P < 0.0001$. The values at the top of the graph are the Var explained (R^2) and the Mean of squared residuals (MSR) for the full model. We counted the proportions of each class of factors (Fig. 3) in the random forest and represented them in a circle plot. We demonstrate the prediction performance of 80 percent of the data for the training set versus 20 percent of the test set, where R^2 represents the correlation between the

1369 observed nitrogen(phosphorus) from soil and the predicted nitrogen(phosphorus) from
1370 soil.

**31:** (xxi) *The authors compare NUE and PUE trends across various regions but fail to*
*contextualize these results adequately within the broader agricultural and*
*environmental policy frameworks. For instance, differences in crop management*
*practices or regional agricultural policies should be considered when interpreting*
*nutrient use efficiency trends.*

**Response:** Thank you for your comment. We have taken into account agricultural
management practices and socio-economic factors, such as crop management and
regional policies, through Pearson correlation analysis and partial least squares
modeling. These analyses allowed us to examine the relationships between NUE/PUE
and various factors, including climate, soil properties, and agricultural management
practices. This provides a more comprehensive understanding of the factors influencing
nutrient use efficiency trends across different regions.

The specific modified text is as follows: “Spearman pearson correlation analyses
was used to test for relationships between NUE(or PUE) and the factors including
climate (MAT, MAP, ET, AI, IND, OND), soil properties (Sand, Silt, Clay, BD, pH, TC,
TN, TP, AP), agriculture management (Tillage, During, Input N, Input P, Input K, FAT,
FAF), society and economy (GDPPC, PD, UR, MYS) by using the “corrplot” R
package⁵⁸ (Fig. 3a). (Revised lines 609-612)”

**32:** (xxii) *Some results lack statistical significance, yet are presented as conclusive*
*findings. The authors must be cautious in making statements about overall trends*
*without robust statistical evidence.*

**Response:** Thank you for your comment. After reviewing the revised manuscript, we
can confirm that we have taken extra care to present only statistically significant results.
We have ensured that all conclusions drawn are supported by robust statistical evidence.
Non-significant trends have been either omitted or appropriately acknowledged in the
discussion. Thank you again for your valuable feedback.

**33:** (xxiii) *The practical implications of the findings are not well articulated. While the*
*authors highlight the need for improved nutrient management, there is little discussion*
*on how these findings should influence agricultural policies or practices. Specific*
*recommendations for targeted interventions in regions with low NUE and PUE need to*
*be detailed, especially given the paper's substantial focus on global patterns*

**Response:** Thank you for your comment. In the revised manuscript, we have clearly
outlined practical recommendations for targeted interventions in regions with low NUE
and PUE, specifically in the context of crop distribution patterns. We acknowledge the
challenges involved in implementing these strategies, such as water stress, soil
conditions, and economic factors. Our revised discussion emphasizes the need for
region-specific solutions that address these barriers, along with the importance of local
agricultural policies and infrastructure improvements. We have provided a more
detailed, actionable approach to improving nutrient management, aiming for both

environmental mitigation and food security.

The specific modified text is as follows: “Reevaluating global crop distribution
patterns emerges as a viable strategy for enhancing NUE and PUE. Encouraging rice
cultivation in the tropics and wheat cultivation in temperate zones, where efficiencies
are notably higher than in other regions of the world (Fig. 4 and Supplementary Fig.
11), could significantly boost production efficiencies. However, as the 2008-2010 food
crisis highlighted, countries might still want to produce their own food for reasons of
food sovereignty, cultural preferences, and national security, even when efficiencies are
lower compared to global averages. For example, in Ethiopia, wheat production is
prioritized despite lower efficiencies, as local food production is crucial for national
food security and cultural practices⁴². This can pose a challenge for adopting large-scale,
region-specific crop distribution strategies that might require shifts in national
agricultural priorities.

Practical challenges arise when implementing region-specific crop distribution.
For instance, in countries like India, overdependence on irrigation for rice cultivation
in water-stressed regions such as Punjab and Haryana has led to severe groundwater
depletion⁴³. Additionally, tropical regions, despite receiving higher rainfall, often lack
the necessary irrigation infrastructure for optimal rice cultivation⁴⁴. Soil conditions also
pose significant challenges, with high P fixation in acidic tropical soils reducing PUE,
while salinity and nutrient depletion limit the expansion of wheat cultivation in
temperate zones^{28,45}. Moreover, historical and economic factors, such as the
profitability of rice cultivation under government procurement schemes, often make
farmers reluctant to change cropping patterns. Expanding rice cultivation in tropical
regions could exacerbate water stress, and expanding wheat in nutrient-depleted areas
might increase reliance on chemical fertilizers. Imbalanced fertilizer use and limited
access to advanced soil testing facilities further hinder improvements in NUE and PUE⁹.
Despite these hurdles, optimizing crop distribution could still lead to significant
environmental and food security benefits, provided these challenges are addressed
through comprehensive, region-specific strategies.

Furthermore, yields per unit area of major food-producing countries in these
regions are higher than in other regions of the world, promising a win-win situation in
terms of both environmental mitigation and food security (Supplementary Fig. 12b).
For example, rice yields averaged 7,028 kg ha⁻¹ in China, 5,203 kg ha⁻¹ in Indonesia,
and 5,818 kg ha⁻¹ in Vietnam, all of which exceed the global average of 4,673 kg ha⁻¹.
Similarly, wheat yields averaged 5,242 kg ha⁻¹ in the European Union and 5,416 kg ha⁻¹
in China, surpassing the global average of 3,423 kg ha⁻¹. While these yield figures
suggest a win-win situation in terms of both environmental mitigation and food security,
the actual implementation in specific regions requires overcoming the challenges of
irrigation infrastructure, soil health, and economic incentives for farmers. Therefore, a
more nuanced, context-specific approach is essential for translating these general yield
trends into actionable recommendations.

The inefficiencies in NUE and PUE in global crop production have resulted in
significant nutrient surpluses, which have had profound impacts on the ecological
environment (Fig. 5). Maize, in particular, registers the highest N (73 kg N ha⁻¹) and P

(15 kg P ha⁻¹) surplus intensities among major crops. China and the United States, which
are the major N and P surpluses in the maize ecosystem, contribute 56.7% of the N
surplus load and 61.9% of the P surplus load due to higher (99.4-115 kg N ha⁻¹ and 16.8-
33.7 kg P ha⁻¹) than the global average surplus intensities and large planted areas
(4.21×10⁵ km² for China and 3.29×10⁵ km² for the United States). Furthermore, our
structural equation model reveals that agricultural management practices are the most
significant factor influencing NUE and PUE in maize, with tillage methods having the
most pronounced impact (Fig. 3). Specifically, the adoption of no-till practices can
concurrently improve NUE and PUE in maize, offering a practical approach to reduce
the N and P surpluses in maize ecosystems. (Revised lines 340-391)”

**34:** *(xxiv) Although the abstract mentions reducing ecological pollution, the discussion*
*does not adequately address the environmental implications of low NUE and PUE.*
*More significant attention should be paid to the potential impact on soil health, water*
*quality, and climate change, linking these issues more explicitly to the findings of the*
*research*

**Response:** Thank you for your comment. While we acknowledge that low NUE and
PUE can have significant environmental implications, such as soil health degradation,
water quality issues, and contribution to climate change, these aspects were not the core
focus of our study. Our primary aim is to explore global trends in nutrient use efficiency
and their influence on agricultural productivity. The environmental impacts, although
important, were addressed briefly in the manuscript as a secondary consideration to
support our main analysis. We recognize the relevance of these issues, but they extend
beyond the scope of our central research focus.

**35:** *(xxv) The manuscript highlights overall global patterns but fails to adequately*
*analyze regional disparities in NUE and PUE. For example, factors that contribute to*
*inefficiencies in high-input agricultural systems versus low-input regions are not*
*sufficiently discussed. The authors need to explore these differences in greater depth to*
*provide a comprehensive perspective*

**Response:** Thank you for your comment. In the revised manuscript, we have made a
conscious effort to address regional disparities in NUE and PUE, particularly focusing
on high-input agricultural systems versus low-input regions. We have discussed crop-
specific factors that contribute to nutrient efficiency, such as climate, soil conditions,
and agricultural practices across different regions. For example, we highlight how
tropical regions benefit from lower input levels, thus achieving higher NUE and PUE
for crops like rice and maize, while temperate regions show improvements in wheat
and soybean efficiencies. We also detail the mechanisms behind these variations,
emphasizing the role of rainfall, soil health, and crop characteristics. These regional
nuances are now better articulated in the manuscript, providing a more comprehensive
understanding of nutrient use efficiency trends. Thank you again for your valuable
feedback.

The specific modified text is as follows: “Recognizing the distinct NUE and PUE
patterns across different climates offers a pathway for optimizing crop distribution

based on regional characteristics. Such strategic crop deployment could reduce
fertilizer dependency in high-efficiency areas, allowing for resource reallocation to
regions with lower nutrient efficiency (Fig. 4 and Supplementary Fig. 11).

Rice exhibits superior NUE (Mean \pm SD = $47 \pm 6.7\%$) and PUE ($49 \pm 15\%$) in
tropical regions, outperforming the global average in countries such as Indonesia,
Myanmar, and Nigeria (Supplementary Fig. 11a). This advantage can be attributed to
multiple factors: 1) tropical countries, often in the developing world, tend to have lower
N and P inputs ($76.8 \text{ kg N ha}^{-1}$ and $12.6 \text{ kg P ha}^{-1}$) relative to the global average (95.3
1509 kg N ha^{-1} and $16.3 \text{ kg P ha}^{-1}$), thus avoiding nutrient surplus and achieving higher NUE
and PUE^{2,3,16}; 2) as a C3 plant, rice maintains high photosynthetic efficiency and rapid
growth under the high temperatures and ample sunlight of tropical conditions,
optimizing N and P utilization for protein synthesis and energy metabolism¹⁶; 3) the
prevalent practice of long-term flooding in tropical rice cultivation creates a water layer
that reduces N volatilization and enhances P solubility, thereby improving nutrient
utilization efficiencies^{35,36}. Additionally, this water layer suppresses weed growth,
reducing nutrient competition; 4) the warm, wet tropical climate supports the
development of deeper, denser root systems in rice, enhancing nutrient absorption
capabilities³⁷; 5) the long growing season facilitates efficient nutrient uptake during
critical growth stages, potentially allowing for multiple cropping cycles each year¹⁶.

Wheat in temperate zones, including China, demonstrates higher NUE ($53 \pm 11\%$)
and PUE ($28 \pm 8.1\%$) than the global average (Supplementary Fig. 11b). This
improvement is driven by: 1) optimal average daily temperatures of $16\text{-}18^\circ\text{C}$ for wheat
growth, resulting in higher biomass and increased fertilizer demand^{27,38}; 2) wheat
cultivation relies heavily on rainfall for irrigation, with temperate regions benefiting
from significant rainfall levels that support wheat growth and increase biomass, thus
enhancing fertilizer demand and utilization efficiency. Rainfall also facilitates P
mobility to root zones, increasing its availability^{27,38}.

Maize achieves the highest NUE ($51 \pm 14\%$) in tropical zones and PUE ($42 \pm 16\%$)
in arid zones, with countries like Brazil surpassing global averages (Supplementary Fig.
11c). The mechanisms underpinning maize's elevated NUE in the tropics are
multifaceted: 1) as a C4 plant, maize thrives under the intense light and high
temperatures typical of tropical climates, facilitating rapid growth and development.
This rapid growth escalates N demand for protein synthesis and other vital functions,
thereby optimizing N utilization; 2) tropical regions generally offer longer growing
seasons, affording maize ample time to absorb and utilize N, which in turn enhances
NUE. Conversely, the mechanisms contributing to the highest PUE in arid zones
include: 1) drought conditions can induce alterations in plant root architecture,
promoting expansion into deeper soil strata to increase P uptake; 2) under such stress,
plants may augment their capacity for inter-root acidification, releasing organic acids
and other substances that improve P availability and efficacy in the soil³⁹.

Soybeans attain the highest NUE ($51 \pm 22\%$) in temperate regions and PUE ($24 \pm$
2.3%) in arid regions, mirroring maize's spatial efficiencies, with the Indian and
Ukraine standing out among major agricultural countries for exceeding global NUE and
PUE averages (Supplementary Fig. 11d). The superior NUE of soybeans in temperate

regions can be attributed to several factors: 1) The moderate temperatures, adequate
precipitation, and extended growing season in temperate climates foster optimal growth
conditions, enhancing both root development and N fixation¹⁸; 2); coupled with
nutrient-rich soils and favorable microbial activity, support higher biological N fixation
by *Rhizobium* species, reducing the need for synthetic fertilizers⁴⁰. In arid regions,
soybeans attain the highest PUE due to: 1) the minimal organic N deposition
characteristic of these areas mitigates microbial competition for soil P during organic
N decomposition, enhancing P availability for soybeans and improving PUE^{27,41}; 2)
drought conditions prompt soybean plants to develop deeper root systems, improving
access to water and nutrients, including P, in subsoil layers; 3) under drought stress,
soybeans may enhance rhizosphere acidification or upregulate the expression of P
transport proteins in their root systems, thereby boosting PUE⁴⁰. (Revised lines 267-
324)”

**36:** (xxvi) *The review of existing literature is not exhaustive, failing to incorporate*
*relevant studies that may provide alternative perspectives or contradict the authors'*
*claims. This lack of engagement diminishes the scholarly rigor of the manuscript*

**Response:** Thank you for your comment. We appreciate your feedback and have taken
steps to broaden the literature review in the revised manuscript. We now include a more
comprehensive engagement with relevant studies, specifically addressing alternative
perspectives and contradictions to our findings.

The specific modified text is as follows: “Our comparative analysis using the
$N(P)UE_{diff}$ and $N(P)UE_{bala}$ methods, based on data from 2008 to 2012, reveals markedly
lower PUE values with the $N(P)UE_{diff}$ approach 26^{2,3}. For instance, PUE for rice, wheat,
maize, and soybean is 44%, 23%, 34%, and 19%, respectively, with the $N(P)UE_{diff}$
method, in contrast to 63%, 80%, 74%, and 76% with the $N(P)UE_{bala}$ method. Notably,
NUE remains relatively consistent between the two methods. These findings suggest
that current crop P uptake relies primarily on soil P mobilization rather than inorganic
fertilizer inputs. This trend, likely driven by historical, intensive P fertilizer application,
has gradually pushed soil P reserves toward saturation^{2,27}. While this process has helped
meet crop P demands in the short term, it has also created long-term sustainability
challenges. P availability in soils is influenced not only by inorganic fertilizer inputs
but also by soil organic matter, which plays a crucial role in enhancing P availability by
releasing organic acids that solubilize bound P^{12,28}. However, many soils worldwide,
particularly in regions with low organic matter content, fail to fully benefit from this
mechanism, limiting the PUE. To further improve PUE, adopting integrated nutrient
management strategies is critical. These strategies include optimizing P fertilizer
placement, improving irrigation practices, and managing crop rotations to minimize
continuous cultivation of high-P-demanding crops^{12,29-31}. In many developing countries,
the lack of access to soil testing facilities or failure to implement recommendations
based on soil tests exacerbates inefficient P use, leading to imbalanced fertilization and
suboptimal P availability^{32,33}. In addition, minimizing fallow periods and managing soil
organic matter content through practices such as cover cropping or organic amendments
could enhance P availability and utilization³⁴. Thus, while regulating the release of

retained soil P could reduce P fertilizer inputs over time, it is crucial to adopt a holistic
approach that addresses the complex factors influencing P availability and uptake.
(Revised lines 242-266)”

**37:** (xxvii) *The manuscript is overly verbose and repetitive, particularly in the Results*
*and Discussion sections. The key findings are buried in lengthy descriptions, making it*
*difficult for readers to discern the main contributions of the study.*

**Response:** Thank you for your constructive feedback. In the revised manuscript, we
have streamlined the Results and Discussion sections to eliminate redundancy and
improve clarity. We have focused on highlighting the key findings more effectively,
ensuring that the main contributions of the study are clearly presented and easily
accessible to readers.

**38:** (xxviii) *The abstract and introduction do not effectively highlight the novelty or*
*significance of the study. The objectives are stated in broad terms, and the rationale for*
*the research is not compelling.*

**Response:** Thank you for your valuable feedback. In the revised manuscript, we have
revised both the abstract and introduction to more clearly highlight the novelty and
significance of the study. The objectives have been more precisely defined, and we have
strengthened the rationale for the research to ensure that the importance of the study is
more compelling and evident from the outset. We appreciate your suggestion and
believe the revisions have improved the clarity and focus of these sections.

The specific modified text is as follows: “**Abstract:**

Enhancing nitrogen (N) and phosphorus (P) use efficiency (NUE and PUE) is essential
for advancing sustainable agriculture and reducing dependency on non-renewable
fertilizers. However, the long-term dynamics of NUE and PUE across major crops
remain poorly understood at a global scale. Here, we compiled a comprehensive global
database encompassing 3,360 observations across 205 countries to analyze trends in
NUE and PUE for major crops from 1961 to 2018. Our findings indicate that, after
dramatic increases in global fertilizer applications since the 1960s, both NUE and PUE
have remained relatively low, with only small improvements after the mid-1970s. Today,
PUE and NUE continue to be suboptimal, particularly in developing regions,
highlighting the urgent need for targeted nutrient management strategies. Global
mapping highlights that NUE and PUE are highly context-dependent, with variations
observed by crop type and region. For instance, rice achieves optimal NUE and PUE in
tropical zones, while wheat performs best in temperate climates. Notably, maize
continues to exhibit significant nutrient inefficiencies, especially in China and the
United States, with considerable N and P surpluses. Taken together, this global analysis
provides critical insights into regions where urgent action is needed to improve nutrient
use efficiency, supporting sustainable agricultural practices and reducing global
fertilizer dependence.

**Introduction**

[revised manuscript text omitted]

Here, we aimed to address these knowledge gaps by compiling a comprehensive
global database on nutrient use efficiency (NUE and PUE), encompassing 3,360
observations across 205 countries and regions (Fig. 1). We will analyze trends and
patterns of NUE and PUE using the N(P)UE_{diff} framework for major crops, leveraging
dynamic national-scale data that includes crop yields, cropland areas, fertilizer N and P
input intensities, nutrient uptake by crops, and residue-grain ratios. Our focus will be
on key global crops—rice, wheat, maize, and soybean—which collectively account for
over half of global crop production and 49% of cropland area^{17,18}. Additionally, we have
developed machine learning models utilizing point-scale NUE and PUE data (n = 2,354
for NUE, n = 1,006 for PUE) alongside climate, soil properties, and agricultural
management information to delineate the current global spatial distribution patterns of
NUE and PUE for these crops and identify their underlying drivers. The primary
objectives of our study are: 1) to quantify trends in overall and major crop NUE and
PUE from 1961 to 2018; 2) to identify current spatial distribution patterns and drivers
of NUE and PUE; and 3) to pinpoint regions with high (> 50%) and low (< 50%) NUE
and PUE to direct targeted improvements. (Revised lines 69-153)”

**39:** (xxix) *The conclusions drawn from the results are often broad and generalized.*
*Many assertions about the future implications of their findings lack the substantial*
*backing of robust evidence presented in the results section. The authors should ensure*
*that their conclusions are directly supported by the results and that they make cautious,*
*evidence-based recommendations*

**Response:** We acknowledge the comment and have carefully revised the conclusions
in the manuscript to ensure they are more directly supported by the findings. We have
removed broad, generalized statements and instead focused on making evidence-based
recommendations that are grounded in the observed trends and patterns. For example,
the suggestion for region-specific nutrient management strategies is now clearly linked
to the patterns observed in the data, particularly regarding the differences in NUE and
PUE across crop types and geographical regions. Additionally, we have emphasized
that while improvements are needed, our recommendations are carefully tailored to the
results presented, particularly in terms of addressing nutrient inefficiencies in key
regions. We have made these adjustments to ensure that the conclusions are firmly
grounded in the data and analysis.

**40:** (xxx) *The figures, while visually appealing, are not always clearly explained. For*
*example, the global maps of NUE and PUE (Figure 4) lack sufficient detail on the*
*underlying data and assumptions.*

**Response:** Thank you for your comment. We have revised the manuscript to clarify the

methodology behind the global maps of NUE and PUE (Fig. 4). We have now included
more detailed explanations of the underlying data and assumptions in the relevant
sections, specifically on the global gridding resolution (5-arcmin) and how it
contributes to capturing regional differences in nutrient use efficiency. We also
emphasize the use of Kriging interpolation for spatial corrections and explain how
uncertainty was quantified in our models, with the standard deviation of predictions
providing a measure of uncertainty (Supplementary Fig. 10). Additionally, we have
included references to the data sources and model validation methods, such as the use
of random forest models with 10-fold cross-validation and grid search for parameter
tuning. These revisions ensure that the methods used to generate the maps are clearly
understood by the reader.

The specific modified text is as follows: “

*Mapping the spatial variation of global NUE and PUE*

**Spearman correlation analysis**

Spearman Pearson correlation analyses was used to test for relationships between
NUE(or PUE) and the factors including climate (MAT, MAP, ET, AI, IND, OND), soil
properties (Sand, Silt, Clay, BD, pH, TC, TN, TP, AP), agriculture management (Tillage,
During, Input N, Input P, Input K, FAT, FAF), society and economy (GDPPC, PD, UR,
MYS) by using the “corrplot” R package⁵⁸ (Fig. 3a).

**Partial least squares path modeling**

To investigate the path relationships between environmental, agricultural management,
and socio-economic factors and their effects on nitrogen use efficiency (NUE) or
phosphorus use efficiency (PUE), we employed Partial Least Squares Path Modeling
(PLS-PM)⁵⁹ to construct a path model aimed at identifying the key driving factors (Fig.
3b). In the modeling process, climate, soil properties, agricultural management
practices, and socio-economic factors were treated as primary latent variables. The sub-
variables contained within each category were considered secondary manifest variables.
To mitigate collinearity issues, secondary variables with loadings below 0.7 were
excluded from the model. The goodness-of-fit (GOF) index was used as a criterion to
evaluate model performance, with values closer to 1 indicating better fit. By
maximizing the GOF, we selected representative secondary variables and constructed
the final path model⁶⁰.

**Random forest modeling**

We selected variables that correlated significantly with NUE(PUE) and the amount of
N(P) fertilizer inputs as factors. Random forest models were used primarily to assess
variable importance and influence by using “random forests” R package⁶¹. In random
forest, unlike traditional regression, correlation among variables does not affect the
model accuracy⁶². We randomly sampled 80% of the dataset for model training, with
the remaining data used for testing. Each final model was obtained based on 10-fold
cross-validation. We used grid search for parameter tuning and sought the combination
of nodesize (Nodesize represents the minimum number of samples required in a leaf
node of a decision tree. When the number of samples in a node is less than or equal to

nodesize, the node will no longer split and will be treated as a leaf node.), ntree (Ntree refers to the total number of decision trees in the random forest, where each tree is trained independently, and the final prediction is obtained by averaging their outputs.), and mtry (Mtry denotes the number of features randomly selected at each split when constructing individual trees.) that minimized the RMSE (Root mean squared error) of the model, ensuring the robustness of the model. Among them, we used contour plots to illustrate the influence of different parameter combinations on random forest model (Supplementary Fig. 8). Both R^2 and P between the predictions and observations were calculated for assessing model performance (Supplementary Fig. 9).

Global NUE and PUE predictions

Trained RF models were used to predict global NUE and PUE for the four major crops. We established a global 5-arcmin resolution grid within the planting areas of the four major crops. A series of globally gridded datasets (Supplementary Table 3) were used for model calculations. The fertilizer input data for each crop were sourced from IFA (2018). We set the prediction timeframe for planting to be one year. This choice was based on two main reasons. Firstly, most of the originally collected NUE and PUE were calculated annually, ensuring more accurate predictions. Secondly, a one-year planting cycle better reflected the efficiency of current fertilizer inputs, enhancing the relevance of the model to agricultural practices.

The random forest model performed multiple predictions for each grid cell according to the number of trees. We derived the final predicted value for each grid cell by averaging these multiple predictions, with the standard deviation of these predictions indicating the uncertainty associated with the model (Supplementary Fig. 10). Then, the global model predictions of NUE and PUE were spatially corrected through Kriging interpolation based on NUE and PUE residuals (observed minus predicted values) using the 'gstat' package in R. This approach better captures the geographical variability of nutrient use efficiency while providing more robust uncertainty estimates for spatial predictions^{63,64}, with the RMSE between predicted and observed values of NUE and PUE decreasing by 0.09 and 0.13 after applying Kriging interpolation for correction, respectively. Additionally, we visualized the *Kriging-adjusted NUE and PUE prediction* of the four major crops on a world map and illustrate the differences between latitudes using line graphs. The overall distribution range and probability density of the data are displayed using the “hrbrthemes” package in the bottom left corner of the map (Fig. 4).

We also investigated the impact of environment and agricultural management in different climatic zones and major agricultural countries. We utilized the Köppen climate classification (Supplementary Fig. 16) and data from the world's major grain-producing countries. The harvested area ranks among the top 10 countries worldwide. Additionally, we included the EU in the comparison of major crop-planting countries. We conducted pairwise comparisons of fertilizer use efficiency among different climate zones or countries and regions using the Dunn's Kruskal-Wallis method implemented in the “multcompView” package⁶⁵. Significant differences ($P < 0.05$) were denoted by different letters (Supplementary Fig. 11). (Revised lines 607-673)”

**Response summary:** We sincerely appreciate your insightful comments, which have
significantly strengthened our manuscript. We believe these revisions address all

concerns raised and underscore the manuscript's relevance to sustainable agriculture.

Thank you for the opportunity to improve our work.

**Reviewer #3 (Remarks to the Author):**

*41: I have read with great interest the manuscript ‘Global-scale prevalence of low*
*nutrient use efficiency across major crops’ and I would like to congratulate the authors*
*for the quality of their study. The paper is rather strong on the spatial analysis of*
*variations in nutrient use efficiency, across major food crops. The authors have built*
*and compiled an impressive dataset on field-level observations across the globe. The*
*introduction rightly points to the need of field-based estimates of fertiliser use efficiency,*
*to complement existing nation-wide evaluation using nutrient balance approaches. This*
*is very critical, as many global-level studies currently lack this ‘bottom-up’ approach*
*were ground observations are used for upscaling and understanding of global patterns.*
*The training of the statistical model with the field observations, and its application for*
*spatial prediction using existing geodata, is adequately detailed. This leads to*
*convincing and critical results on the contrasting impact of climate on nutrient use*
*efficiencies for different crops, and the identification of hotspots where inefficiencies*
*are high. The paper also provides a nice perspective on the optimisation of crop*
*distribution across the globe in order to increase nutrient use efficiency. For all these*
*reasons, I believe this manuscript deserves publication.*

**Response:** Thank you! We appreciate your positive and constructive comments on our
study, and that you nicely highlighted the novelty and implications of our work.
Understanding the global pattern of NUE and PUE and the associated predictors are
essential for realizing the goals of sustainable agriculture, balancing high crop yields
with environmental protection. To better ensure the scientific robustness and clarity of
the conclusions as suggested, we have carefully studied all your comments and
suggestions, and incorporated them into our manuscript. Please check more details as
below.

*42: I have one major comment though, that I think needs to be addressed. The temporal*
*analysis of variations in nutrient use efficiency seems biased. Below I explain why:*
*The relationships established in Figure S6 between nutrient use efficiency, soil nutrient*
*supply, and nutrient input intensity, though significant, have a small explanatory power*
*(in most case explain less than a quarter of the observed variation, $r^2 < 0.25$). These*
*relationships therefore neglect the impact of climate, soil, and management (beyond*
*input intensity, e.g. timing, placement etc), on nutrient use efficiency and soil nutrient*
*supply. They tend to overemphasize the impact of nutrient input intensity on the*
*observed efficiency. Yet, 60% efficiency can be achieved at almost any level of N input*
*(Figure S6, though colours of the dots are not easy to distinguish, this may need to be*
*improved), and efficiency can be very variable at a given nutrient input level.*
*As far as I understand, these simple relationships are then used to compute historical*
*values of nutrient use efficiencies (L502-514). The authors then analyse historical*
*changes in these efficiencies with the lens of possible changes in crop management*
*(beyond input intensity, e.g. varietal choice, good agronomic practises) in different*
*regions of the world, that would have helped increase or decrease nutrient use efficiency*
*(L150-218). But because intensity of nutrient input and efficiency are so much*

*correlated in the computation, isn't that an artificial construct were greater use of*
*nutrients automatically leads to lower efficiency (conversely, lower use leading to*
*greater efficiency), neglecting how changes in management (and possibly climate)*
*might have altered these connections? Very possibly, lower input use can lead to lower*
*or stable efficiency, if crop management does not improve (this may be happening in*
*some regions of the world). The use of the random forest model (L554-557), that better*
*accounts for the influence of climate, soil and management on efficiency could be an*
*alternative to the equations of Figure S6. But then, historical data on crop management*
*is hard to find.*

*I suggest the authors provide more details on the computations and its logic, in order*
*to convince the reader that this bias does not exist, or if it does, discuss the current*
*limitations around the data and the computations, and stick to describing the historical*
*changes in input intensity, without concluding too strongly into possible changes in*
*efficiency (both in the abstract and the result sections).*

**Response:** We sincerely appreciate the reviewer's insightful comments on potential
historical temporal biases in our nutrient use efficiency assessment, which have
prompted critical enhancements to our analytical framework. We have integrated
climatic, soil properties, and agricultural management factors into random forest
models to predict the nitrogen from soil and phosphorus from soil across different crops
(Supplementary Fig. 6).

The specific modified text is as follows: “This dataset collected N fertilizer input
intensities (Input N, kg ha⁻¹), P fertilizer input intensities (Input P, kg ha⁻¹), crop N
uptake (kg ha⁻¹) and crop P uptake (kg ha⁻¹), crop N (%) and P (%) content, NUE (%),
PUE (%) in field experiments. Based on these data, the proportion of nitrogen
(*Nitrogen from soil, %*) and phosphorus (*Phosphorus from soil, %*) uptake from
soil were calculated as follows:

$$1882 \quad \text{Nitrogen(Phosphorus) from soil} = \frac{\text{crop N(P)uptake} - \text{NUE(PUE)} \times \text{Input N(P)}}{\text{crop N(P) uptake}} \times 100\% \quad (5)$$

Furthermore, considering the negative correlation ($P < 0.05$) between *Input N(P)*
and *Nitrogen(Phosphorus) from soil* (Supplementary Fig. 5). We trained the
random forest (RF) models predict Nitrogen from soil and Phosphorus from soil across
different crops (Supplementary Fig. 6). The dataset was partitioned into 80% training
and 20% testing sets via random sampling. Parameter tuning was performed via grid
search to identify the combination with the lowest root mean square error (RMSE).
Each final model was derived through a 10-fold cross-validation procedure to ensure
robust prediction. (Revised lines 528-540)”

Fig. S6 Prediction of weighted effect values of nitrogen and phosphorus from soil of the four major crops (rice, wheat, maize, soybean) and all crops by random forest models.

MAT, mean annual temperature; MAP, mean annual precipitation; Sand, sand content;

Silt, silt content; Clay, clay content; BD, bulk density; TC, total carbon; TN, total

nitrogen; TP, total phosphorus; AP, available phosphorus; Input N, nitrogen fertilizer

input; Input P, phosphate fertilizer input. The significance of the variables in Figure is

measured by the "percentage of increase of mean square error" (%IncMSE) value in

Random Forest, where higher %IncMSE values imply more important variables and

identify the significance of each variable. *: $P < 0.05$; **: $P < 0.01$; ***: $P < 0.001$;

****: $P < 0.0001$. The values at the top of the graph are the Var explained (R^2) and the

Mean of squared residuals (MSR) for the full model. We counted the proportions of

each class of factors (Fig. 3) in the random forest and represented them in a circle plot.

We demonstrate the prediction performance of 80 percent of the data for the training

set versus 20 percent of the test set, where R^2 represents the correlation between the

1907 observed nitrogen(phosphorus) from soil and the predicted nitrogen(phosphorus) from
1908 soil.

*Other minor comments:*

**43:** Figure 1: Subplot b is not explained (axis, acronyms, title, colors). The caption
should indicate that the figure shows the location of the reviewed studies.

**Response:** Thank you. We have explained the the meaning of axis, acronyms, title,
colors in Subplot b as “MAT, mean annual temperature; MAP, mean annual
precipitation. Count represents the frequency of the collected samples within the similar
climatic conditions (b).”

**Figure 1 A global assessment of field-based nitrogen (NUE) and phosphorus use**
**efficiency (PUE) across four major crops (rice, wheat, maize, soybean)**

The circular legend indicates the global distribution of the four major crops in a, where
the overlap represents intercropping and rotational cropping. Crop plantation base map
from SPAM 2010 V2r0¹. The four colors of the crops in c are referenced to the staining
of the four crops in a, distributed from top to bottom to represent rice, wheat,
and soybean. MAT, mean annual temperature; MAP, mean annual precipitation. Count
represents the frequency of the collected samples within the similar climatic conditions
(b). The letter M, SD, n refer to the mean value, standard deviation and count,
respectively (c).

**44: L103:** *'current N(P)UE_{bala} approaches' the acronym has not been defined and*
*this looks awkward at this stage.*

**Response:** Thank you for pointing this out. We have now added the full names of all
abbreviations, such as the amendment for the N(P)UE_{bala} approaches as follows: “First,
current N and P balance (N(P)UE_{bala}) approach approaches¹¹, which incorporate
national-scale fertilizer inputs—including atmospheric deposition, biological N
fixation, and both inorganic and organic fertilizers—and nutrient harvests (including
grain and residue), are the predominant methods for assessing global NUE and PUE^{2,3}
(See Supplementary Table 1 for detailed definitions and calculation formulas). (Revised
lines 107-112)”

**45: L570:** *To my understanding 'management' beyond input use intensity is only the*
*variable 'tillage/no tillage'. Did the authors have a look at the number of applications,*
*the type and placement of fertiliser? Surely these variables exist in the reviewed study.*
*Of course, the geodata for these variables do not exist, but the relationships could be*
*built with the training dataset. Do these variable help explain some of the observed*
*variability? The new model could then be used to explore what-if scenarios around*
*improvement in management at different locations.*

**Response:** Thank you for your insightful comments. We have collected fertilizers
application types, fertilizers application placement, fertilizer application frequency,
irrigation methods as agriculture management factors to analyze the relationship
between NUE(PUE) and agriculture management and construct structural equation
models (Fig. 3). Moreover, agricultural management variables selected by Spearman
correlation analysis were incorporated as predictors in random forest models to estimate
global nutrient use efficiency (Supplementary Fig. 9).

The specific modified text is as follows: “(8) fertilizers application types (FAT)
including organic fertilizers, inorganic fertilizers, and the combined application of
organic and inorganic fertilizers; (9) fertilizers application placement (FAP) including
surface application, deep application, mixed application, and foliar spraying; (10)
fertilizer application frequency (FAF); (11) irrigation methods (IM) including
permanent flooding, intermittent irrigation, drip irrigation, and no irrigation; (Revised
lines 459-463)

Spearman pearson correlation analyses was used to test for relationships between
NUE(or PUE) and the factors including climate (MAT, MAP, ET, AI, IND, OND), soil
properties (Sand, Silt, Clay, BD, pH, TC, TN, TP, AP), agriculture management (Tillage,
During, Input N, Input P, Input K, FAT, FAF), society and economy (GDPPC, PD, UR,
MYS) by using the “corrplot” R package⁵⁸ (Fig. 3a). (Revised lines 609-612)

To investigate the path relationships between environmental, agricultural
management, and socio-economic factors and their effects on nitrogen use efficiency
(NUE) or phosphorus use efficiency (PUE), we employed Partial Least Squares Path
Modeling (PLS-PM)⁵⁹ to construct a path model aimed at identifying the key driving
factors (Fig. 3b). (Revised lines 615-618)”

Figure 3 Global-scale correlations and structural equation model showing the relative importance of climate, soil properties, agriculture management, and socio-economic on nitrogen (NUE) and phosphorus use efficiency (PUE) of four major crops (rice, wheat, maize, soybean).

MAT, mean annual temperature; MAP, mean annual precipitation; ET, evapotranspiration; AI, aridity index; IND, inorganic nitrogen deposition; OND, organic nitrogen deposition; Sand, sand content; Silt, silt content; Clay, clay content;

BD, bulk density; TC, total carbon; TN, total nitrogen; TP, total phosphorus; AP, available phosphorus; Tillage, no-till or not; During, planting years; Input N, nitrogen
 fertilizer input; Input P, phosphate fertilizer input; Input K, potash fertilizer input; FAT,
 fertilizer application types; FAF, fertilizer application frequency; FAP, fertilizer
 application placement; IM, irrigation method; GDPPC, gross domestic product per
 capita; PD, population density; UR, urbanization rate; MYS, mean years of schooling.
 *: $P < 0.05$; **: $P < 0.01$; ***: $P < 0.001$; ****: $P < 0.0001$ (a). The standard coefficient
 for each path is shown in a circle within the path. The overall predictive performance
 of the model is evaluated using the Goodness-of-Fit (GOF) statistic, with higher values
 indicating better prediction. The amount of variance explained by the model (R^2) is also
 shown for each response variable (b).

 **Fig. S9 Prediction of weighted effect values of nitrogen (NUE) and phosphorus use**
 **efficiency (PUE) of the four major crops by random forest models.**

MAT, mean annual temperature; MAP, mean annual precipitation; ET, evapotranspiration; AI, aridity index; IND, inorganic nitrogen deposition; OND, organic nitrogen deposition; Sand, sand content; Silt, silt content; Clay, clay content; BD, bulk density; TC, total carbon; TN, total nitrogen; TP, total phosphorus; AP, available phosphorus; Tillage, no-till or not; During, planting years; Input N, nitrogen
 fertilizer input; Input P, phosphate fertilizer input; Input K, potash fertilizer input; FAT,
 fertilizer application types; FAF, fertilizer application frequency; FAP, fertilizer
 application placement; IM, irrigation method; GDPPC, gross domestic product per
 capita; PD, population density; UR, urbanization rate; MYS, mean years of schooling. The significance of the
 variables in Figure is measured by the "percentage of increase of mean square error"

(%IncMSE) value in Random Forest, where higher %IncMSE values imply more important variables and identify the significance of each variable. *: $P < 0.05$; **: $P < 0.01$; ***: $P < 0.001$; ****: $P < 0.0001$. The values at the top of the graph are the Var explained (R^2) and the Mean of squared residuals (MSR) for the full model. We counted the proportions of each class of factors (Fig. 3) in the random forest and represented them in a circle plot. We demonstrate the prediction performance of 80 percent of the data for the training set versus 20 percent of the test set, where R^2 represents the correlation between the Observed NUE / PUE and the predicted NUE / PUE.

46: Figure 3: It is hard to understand what the authors mean with 'During, planting years'. To my understanding, the planting year is confounded with the climatic data. Why would the authors want to have it as a predicting variable? How is this predictor then used for the spatial extrapolation? This deserve clarification.

Response: Sorry for the confusion. “During” refers to the experimental period, which has been reinterpreted in light of your comments. In this study, “During” was not used as a predictor variable, but as an explanatory variable for nitrogen and phosphorus utilization efficiency.

47: L297: Organising the food system at global level is a noble objective. Yet, countries might still want to produce their own food (e.g. wheat in Ethiopia) despite lower efficiencies than elsewhere in the world, because food sovereignty (and cultural preferences) is so critical for them, and a better strategy than heavy reliance on imports (see e.g. this paper: <https://iopscience.iop.org/article/10.1088/1748-9326/11/3/035007>). The authors might want to reflect on this issue.

Response: Thank you for your insightful comments. We have revised the manuscript to reflect the importance of food sovereignty and national priorities in shaping agricultural strategies. As suggested, we have included a discussion of the potential challenges associated with encouraging region-specific crop distribution, especially in the context of countries like Ethiopia, where local food production is critical for national food security and cultural preferences. We also acknowledge the implications of climate-induced food crises, such as the 2008–2010 food crisis, and have incorporated references to studies highlighting the geopolitical and poverty-related vulnerabilities of regions that depend heavily on imports⁴². These points are now addressed in the section discussing global crop distribution patterns and the role of food sovereignty in shaping agricultural decisions. We hope these revisions improve the manuscript and better reflect the complexities of the global food system.

The specific modified text is as follows: “Reevaluating global crop distribution patterns emerges as a viable strategy for enhancing NUE and PUE. Encouraging rice cultivation in the tropics and wheat cultivation in temperate zones, where efficiencies are notably higher than in other regions of the world (Fig. 4 and Supplementary Fig. 11), could significantly boost production efficiencies. However, as the 2008-2010 food crisis highlighted, countries might still want to produce their own food for reasons of food sovereignty, cultural preferences, and national security, even when efficiencies are lower compared to global averages. For example, in Ethiopia, wheat production is

2047 prioritized despite lower efficiencies, as local food production is crucial for national
food security and cultural practices⁴². This can pose a challenge for adopting large-scale,
region-specific crop distribution strategies that might require shifts in national
agricultural priorities. (Revised lines 340-350)”

**Reference**

42 Bren d'Amour, C., L. Wenz, M. Kalkuhl, J. Steckel, and F. Creutzig. Teleconnected
food supply shocks. *Environ. Res. Lett.* **11**, 035007 (2016). (Revised lines 880-881)

**48: L392:** *The authors have built an impressive dataset that, if made publicly available,*
*would be of great value for the global research community interested in nutrient*
*management.*

**Response:** Thank you. We have uploaded our dataset to Figshare to make it openly
accessible to the research community. The dataset is now available at
[<https://doi.org/10.6084/m9.figshare.25998922.v1>]. We hope our work will be of great
value to the global research community interested in nutrient management.

**49: L390:** *This should be Figure S15 I think – but the figure is not so useful, unless it*
*shows the overlap between location of reviewed studies and the biome classification.*

**Response:** Thank you. Done as suggested.

**Response summary:** We would like to sincerely thank you for your positive evaluation
and thoughtful comments. Your suggestions have significantly helped us improve the
scientific rigor, transparency, and clarity of our manuscript. We hope these revisions
meet your expectations and further strengthen the contribution of our work to the field.

REVIEWER COMMENT

Reviewer #1 (Remarks to the Author):

I believe the authors have adequately addressed all my concerns, and I am satisfied with the revised manuscript.

General Response: Thank you very much for your positive and constructive feedback. We sincerely appreciate your thoughtful comments throughout the review process, which have greatly improved the clarity and quality of our manuscript.

Reviewer #2 (Remarks to the Author):

General Response: Thank you for your positive and constructive comments. As this is our final opportunity to respond according to *Nature Communications* review policy, we have carefully considered each of your points. To ensure clarity and accuracy, we have divided your comprehensive feedback into several key issues and responded to each one individually below. We appreciate your understanding and hope our detailed replies address your concerns effectively.

I: The authors have made substantial efforts to address the reviewers' concerns, particularly in enhancing methodological transparency, expanding discussions on socioeconomic factors, and improving model validation. However, several critical issues remain unresolved. In terms of methodology and data reliability, the authors incorporated regional examples such as China's transition and applied Kriging interpolation to reduce spatial uncertainty, achieving a reduction in RMSE for NUE (0.09) and PUE (0.13). They also corrected historical fertilizer data using IFA/FAOSTAT baselines. Nevertheless, the reliance on national-level data and inconsistencies prior to 1990 persist, with the lack of high-resolution regional datasets or early-decade validation introducing potential biases that were not fully addressed.

Response: We sincerely thank you for your constructive and insightful feedback. We acknowledge the continued concern regarding the reliability of historical fertilizer input data and the limitations arising from the use of national-level datasets. To further clarify this issue, we have now implemented several methodological refinements to further address these issues and clarify our data-handling approach:

To address inconsistencies in historical phosphorus fertilizer input data, we applied a segmented correction strategy anchored in the baseline years 1990, 1999, and 2018—selected based on the availability and relative consistency of IFA reports. By integrating total national fertilizer input data from FAOSTAT and crop-specific harvested areas, we dynamically calibrated fertilizer input intensities across countries, crops, and years using Equations (6) and (7). This approach corrects for systematic differences between IFA and FAOSTAT, allowing us to estimate continuous and consistent fertilizer input trends from 1961–2018.

In the absence of high-resolution subnational data—particularly for earlier decades—we further reduced potential biases by extrapolating adjacent-year data and applying regional correlation analyses to infer missing values. While these approaches do not fully substitute for fine-scale datasets, they represent a pragmatic and statistically grounded solution under current data constraints.

Moreover, as our study's primary aim is to assess cross-country and zonal differences in NUE and PUE, the national-scale resolution aligns with the intended scope of our analysis. Nonetheless, a finer spatial granularity would enhance robustness, but the national-scale resolution is appropriate for our objective of assessing cross-country and zonal differences in NUE and PUE. This scale allows us to leverage consistently available data across a large number of countries and over a long period, enabling a robust analysis of macro-scale trends. Regarding the failure to achieve a more refined scale analysis, we have explicitly acknowledged this limitation in the revised manuscript (lines 414–421), and we highlight it as a key direction for future research should more detailed regional fertilizer input records become available.

To further improve historical estimates of NUE and PUE, we implemented a random forest modeling framework trained on environmental covariates, including soil properties, climate data,

and agricultural management factors. This model enables more robust prediction of nutrient uptake from soil and enhances the credibility of historical nutrient use efficiency estimates. These improvements, detailed below, are intended to minimize residual uncertainty stemming from input data limitations.

The specific text modified is as follows (Revised lines 561-601):

“Calculations of NUE and PUE

The analytical framework for the calculation of NUE and PUE across major crops, including their cumulative assessment, involved a detailed multi-step process based on data from FAOSTAT and IFA. This process comprised four principal steps:

(1) Selection of baseline years:

We selected IFA reports from 1990, 1999, and 2018 as baseline years to anchor our analysis. To address data gaps within these reports, we employed a data supplementation strategy, prioritizing: (i) data extraction from adjacent years; (ii) regional data correlation; and (iii) supplementation using global averages.

(2) Adjustment of Fertilizer Input Intensities:

To derive fertilizer input intensity data for different years, countries, and crops globally, we integrated FAOSTAT and IFA data to adjust fertilizer input intensities for baseline year as follows^{2,3}:

$$Input\ N_{co,cr,yr} = Input\ N_{IFA_{co,cr}} \times \frac{TN_FAO_{input,yr}}{\sum_{co} \sum_{cr} (Input\ N_{IFA_{co,cr}} \times A_{FAO_{co,cr,yr}})} \quad (6)$$

$$Input\ P_{co,cr,yr} = Input\ P_{IFA_{co,cr}} \times \frac{TP_FAO_{input,yr}}{\sum_{co} \sum_{cr} (Input\ P_{IFA_{co,cr}} \times A_{FAO_{co,cr,yr}})} \quad (7)$$

where $Input\ N_{IFA_{co,cr}}$ is the cropland N fertilizer input intensities ($kg\ ha^{-1}$) across different countries (co), crops (cr) based on IFA reports for the baseline years 1990, 1999, 2018; $TN_FAO_{input,yr}$ is the total N fertilizer input (t) across different years (yr) from FAOSTAT; $A_{FAO_{co,cr,yr}}$ is the harvested area (ha) across different countries, crops, years from FAOSTAT. $Input\ N_{co,cr,yr}$ is the corrected N fertilizer input intensities ($kg\ ha^{-1}$) across different countries, crops, years. The formula (7) serves as a correction for the P fertilizer input intensities, with each parameter referenced from the formula (6).

(3) Temporal trend estimation:

Using the corrected fertilizer input intensities from 1961 to 2018 (Fig. 2a and Fig. 2b), we derived temporal trends for fertilizer use. The calculations were based on the baseline years 1990, 1999, and 2018, as data availability in IFA is limited for other years, restricting effective regional assessments of fertilizer intensity. Specifically, the period divisions were: 1961–1994, 1995–2006, and 2007–2018, corresponding to the respective baseline years.

(4) Calculation of NUE and PUE (1961–2018):

We calculated NUE (%) and PUE (%) over this period (see Fig. 2c) based on the potential relationship ($P < 0.05$) between NUE(PUE) and N(P) fertilizer input intensities (Supplementary Fig. 2). These were formulated as:

$$F_{N(P)} = \sum_{co} (Input\ N(P)_{co,cr,yr} \times A_{FAO_{co,cr,yr}}) \quad (8)$$

$$U_{N(P)} = \frac{Q \times N(P)_G + Q \times R \times N(P)_S}{100} \quad (9)$$

$$U_{N_0(P_0)} = \frac{U_{N(P)} \times Nitrogen(Phosphorus)\ from\ soil}{100} \quad (10)$$

$$NUE(PUE) = \frac{U_{N(P)} - U_{N_0(P_0)}}{F_{N(P)}} \times 100\% \quad (11)$$

where the parameters are production quantity (Q , t), the nitrogen (F_N , t) and phosphorus (F_P , t) inorganic fertilizer inputs for each crop (Extended Data Fig. 2), residue-grain ratio (R), nitrogen (N_G , %; N_S , %) and phosphorus (P_G , %; P_S , %) content in grain and residue (Supplementary Table 1), nitrogen from soil (%), phosphorus from soil (%), respectively; $U_{N(P)}$ (t) are the N and P uptake by mature crops from the soil and fertilizers; $U_{N_0(P_0)}$ (t) are the N and P uptake by mature crops from only soil under the condition of no fertilization.”

To improve the accuracy of historical NUE and PUE estimates, we employed random forest models to predict NUE and PUE from soil ratios across different years. These models incorporate climate variables, soil properties, and agricultural management factors. This integration of environmental covariates further reduces biases in historical NUE/PUE calculations.”

The specific text modified is as follows (Revised lines 537-559):

“Predicted nutrient uptake by crops from soil

To develop predictive models of nutrient uptake by crops from soil across different crop types (Supplementary Fig. 3), we first calculated the proportion of nitrogen (*nitrogen from soil*, %) and phosphorus (*phosphorus from soil*, %) uptake from soil, based on data (including Input N(P), crop N(P) uptake and NUE(PUE)) originated from a systematic review of peer-reviewed publications, its detailed search strategy and data extraction procedures were delineated in *Global NUE and PUE at sampling points*. The calculated formula as follows:

$$\text{Nitrogen(Phosphorus) from soil} = \frac{\text{crop N(P) uptake} - \text{NUE(PUE)} \times \text{Input N(P)}}{\text{crop N(P) uptake}} \times 100\% \quad (5)$$

Then, we utilized above-calculated nitrogen(phosphorus) from soil, climate (MAT, MAP), soil properties (BD, sand, silt, clay, TC, TN, TP, AP, pH) and agriculture management (Input N, Input P) factors to train random forest model across different crops (Supplementary Fig. 3). These environmental variables were sourced from a systematic review of peer-reviewed studies.

The dataset was partitioned into 80% for training and 20% for testing, sampled randomly. Parameter tuning was conducted via grid search to identify the combination that minimized root mean square error (RMSE). Each final model was built using a 10-fold cross-validation approach to ensure robustness and predictive accuracy.

Finally, the trained RF models (Supplementary Fig. 3) were employed to predict annual nitrogen and phosphorus uptake from soil for the four major crops. Climate data (MAT, MAP) were obtained from CRU TS⁴⁹, while agricultural management data (Input N, Input P) were sourced from FAOSTAT (Supplementary Table 2). Both datasets exhibit interannual variability. Since soil properties are relatively stable over time and no globally comprehensive datasets capture their interannual variations, temporal changes in soil properties were not incorporated into the model (Supplementary Table 2).”

2: Regarding statistical rigor, segmented regression with breakpoints (e.g., 1975 ± 3) and Wilcoxon tests were added, along with uncertainty intervals for breakpoint years. Despite these improvements, the selection of breakpoints remains inadequately justified, and the absence of autocorrelation analysis or validation against external policy and technological milestones continues to undermine the temporal validity of the claims.

Response: We sincerely thank you for your valuable feedback on the statistical rigor of our analysis. In response to your concern regarding the justification of breakpoints, we have expanded our explanation to clarify the computational rationale and methodological framework.

The breakpoints were identified using segmented regression implemented through the segmented package in R—a data-driven method that has been widely applied in Earth (Goyette et al., 2018) and life sciences (Ouyang et al., 2024). This approach begins by assigning the median of the independent variable as an initial estimate and iteratively minimizes the residual sum of squares to locate the breakpoint until convergence is reached (Muggeo, 2003; Muggeo, 2008). As you correctly noted, the initial justification of breakpoint placement was insufficient; we now emphasize that the breakpoint is entirely data-driven and mathematically optimized without the imposition of prior assumptions (Cui et al., 2022), enhancing its objectivity and reproducibility.

To further address your comment on temporal validity, we have aligned the identified breakpoints with known agricultural policy interventions and technological shifts. This contextual validation helps support the interpretability and plausibility of the segmented regression outcomes. These correspondences have been clarified in the revised methods section.

The reason for not introducing autocorrelation tests is that: in time series analysis, autocorrelation tests are typically introduced to exclude the influence of temporal inertia on NUE/PUE, thereby clarifying the mechanisms of non-temporal factors. However, our objective is specifically to investigate how NUE/PUE change over time, without involving discussions of non-temporal factors. Therefore, autocorrelation tests are not required in this study.

In the methods, we have expanded the description of the breakpoint detection process (Revised lines 605-611):

“We used the “segmented” software package⁵⁸ in R to calculate breakpoints in the regression model. The principle of the calculation involves dividing the dataset into two segments by introducing the median of the interval of the dependent variable as the initial breakpoint, allowing each segment to have a different slope. By iteratively minimizing the residual sum of squares, the position of the breakpoint is refined until convergence is achieved^{59,60}. We then conducted linear regressions for the periods before and after the breakpoint.”

Reference

- Cui, Y. *et al.* Ecoenzymatic stoichiometry reveals widespread soil phosphorus limitation to microbial metabolism across Chinese forests. *Commun. Earth Environ.* **3**, 184 (2022).
- Goyette, J. O., Bennett, E. M. & Maranger, R. Low buffering capacity and slow recovery of anthropogenic phosphorus pollution in watersheds. *Nat. Geosci.* **11**, 921-925 (2018).
- Muggeo, V. M. R. Estimating regression models with unknown break-points. *Stat. Med.* **22**, 3055-3071 (2003).
- Muggeo, V. Segmented: An R Package to Fit Regression Models With Broken-Line Relationships. *R News* **8**, 20-25 (2008).
- Ouyang, M. *et al.* Spatiotemporal cerebral blood flow dynamics underlies emergence of the limbic-sensorimotor-association cortical gradient in human infancy. *Nat. Commun.* **15**, 8944 (2024).

3: *In the realm of model validation, the authors implemented 10-fold cross-validation, grid search for hyperparameter tuning, and Kriging interpolation, supported by contour plots and R² values.*

Yet, overfitting risks are not explicitly addressed—such as through learning curves—and the rationale behind parameter choices like nodesize and mtry lacks depth, with spatial uncertainty still evident in Supplementary Fig.

Response: Thank you! We conducted further analyses and provide additional clarifications aiming to address your concerns. In brief:

First, while random forest inherently reduces overfitting risk through bootstrap aggregation and random feature selection (Breiman, 2001; Biau & Scornet, 2016), we acknowledge that these mechanisms do not eliminate overfitting entirely. To further mitigate this risk, we employed 10-fold cross-validation on independent data partitions and performed grid search to tune hyperparameters (nodesize, mtry, ntree). This procedure allowed us to optimize model complexity and predictive accuracy by minimizing root mean square error (RMSE), thus balancing model variance and bias. Based on cross-validation results, we did not observe significant signs of overfitting (see Supplementary Fig. 6).

You rightly pointed out that learning curves were not included in our validation pipeline. We agree that learning curves offer valuable insights into model behavior across different training sizes. While not included in this version due to space constraints, we plan to integrate such diagnostics in future iterations to further reinforce model transparency and learning dynamics.

Regarding hyperparameter justification:

1. nodesize controls the minimum number of observations required in terminal nodes. A small value can lead to overfitting by allowing trees to grow excessively deep and capture noise, whereas a larger value improves generalizability by enforcing simpler splits.
2. mtry determines the number of variables randomly sampled at each node split. Tuning this parameter allows the model to balance decorrelation across trees with the strength of individual learners.

We systematically tuned both parameters using a grid search, selecting values that minimized RMSE across validation folds (Probst et al., 2019). This optimization process is visualized in the contour plots in Supplementary Fig. 5.

On the issue of spatial uncertainty: as you correctly observed, spatial predictions remain susceptible to local error propagation. While our kriging-based post-processing reduces residual variance, it cannot fully eliminate forecast uncertainty. Consistent with prior global modeling studies (Terrer et al., 2021; Ren et al., 2024), we report both pointwise predictions and corresponding error distributions to reflect inherent spatial uncertainties. These global error surfaces are intended not only as diagnostics but also as cautionary boundaries for model interpretation.

To address these points, we have expanded the Methods section accordingly (Revised lines 649–665):

“We selected variables that showed significant correlations with NUE (or PUE) and the amount of N (or P) fertilizer inputs as factors. Random forest models were primarily employed to assess variable importance and influence using the “**randomForest**” R package⁶⁵. Unlike traditional regression models, random forests are unaffected by correlations among predictors, which do not compromise model accuracy⁶⁶.

We randomly sampled 80% of the dataset for model training, with the remaining 20% allocated for testing. Each final model was developed through 10-fold cross-validation. Parameter tuning was

performed via grid search to identify the combination of nodesize (minimum number of samples required in a leaf node; when the number of samples in a node is less than or equal to nodesize, the node no longer splits and is considered a leaf), ntree (total number of decision trees in the forest; each trained independently, with the final prediction obtained by averaging their outputs), and mtry (number of features randomly selected at each split when constructing each tree) that minimized the root mean squared error (RMSE), thus ensuring model robustness.

Furthermore, contour plots were used to illustrate the influence of different parameter combinations on model performance (Supplementary Fig. 5). Model accuracy was evaluated using the coefficient of determination (R^2) and the P -value between predictions and observations (Supplementary Fig. 6).”

Fig. S5 Predictive performance across hyperparameter combinations (ntree, mtry, nodesize).

In Random Forest model, the ntree parameter defines the total number of decision trees, balancing model stability and computational cost; mtry determines the number of randomly selected features evaluated at each node split, influencing feature diversity and overfitting risk; while nodesize sets the minimum observations required in terminal leaves, controlling tree depth and granularity to prevent over- or underfitting.

Fig. S6 Prediction of weighted effect values of nitrogen (NUE) and phosphorus use efficiency (PUE) of the four major crops by random forest models.

MAT, mean annual temperature; MAP, mean annual precipitation; ET, evapotranspiration; AI, aridity index; IND, inorganic nitrogen deposition; OND, organic nitrogen deposition; Sand, sand content; Silt, silt content; Clay, clay content; BD, bulk density; TC, total carbon; TN, total nitrogen; TP, total phosphorus; AP, available phosphorus; Tillage, no-till or not; During, planting years; Input N, nitrogen fertilizer input; Input P, phosphate fertilizer input; Input K, potash fertilizer input; FAT, Fertilizer application types; GDPPC, gross domestic product per capita; PD, population density; UR, urbanization rate; MYS, mean years of schooling. The significance of the variables in Figure is measured by the "percentage of increase of mean square error" (%IncMSE) value in Random Forest, where higher %IncMSE values imply more important variables and identify the significance of each variable. *: $P < 0.05$; **: $P < 0.01$; ***: $P < 0.001$; ****: $P < 0.0001$. The values at the top of the graph are the Var explained (R^2) and the Mean of squared residuals (MSR) for the full model. We counted the proportions of each class of factors (Fig. 3) in the random forest and represented them in a circle plot. We demonstrate the prediction performance of 80 percent of the data for the training

set versus 20 percent of the test set, where R^2 represents the correlation between the Observed NUE / PUE and the predicted NUE / PUE.

Reference

- Biau, G. & Scornet, E. A random forest guided tour. *Test* **25**, 197-227 (2016).
- Breiman, L. Random Forests. *Mach. Learn.* **45**, 5-32 (2001).
- Liaw, A. & Wiener, M. Classification and Regression by RandomForest. *R News* **2**, 18-22 (2001).
- Probst, P., Wright, M. N. & Boulesteix, A.-L. Hyperparameters and tuning strategies for random forest. *WIREs Data Min. Knowl. Discov.* **9**, e1301 (2019).
- Ren, S. *et al.* Projected soil carbon loss with warming in constrained Earth system models. *Nat. Commun.* **15**, 102 (2024).
- Terrer, C. *et al.* A trade-off between plant and soil carbon storage under elevated CO₂. *Nature* **591**, 599-603 (2021).

4: *Still, institutional frameworks such as subsidies and governance were deemed beyond the study's scope, leaving important gaps in understanding regional disparities.*

Response: We sincerely appreciate your thoughtful observation regarding the absence of institutional frameworks—such as subsidies and governance—from our analysis. You are right that these factors can significantly influence regional disparities in nutrient use efficiency (NUE/PUE), and their omission does represent a limitation of the present work.

However, as our study aims to assess long-term, macro-scale trends at global and national levels over six decades, the inclusion of regional institutional data falls beyond the analytical scope. We recognize that subsidies and governance mechanisms likely play critical roles at subnational scales, but comprehensive datasets spanning this temporal and spatial extent are currently fragmented or unavailable, particularly at the provincial or state level.

That said, we agree with your point that addressing these institutional factors is essential for understanding fine-scale variation in NUE/PUE. We now explicitly acknowledge this in the revised manuscript and emphasize the need for coordinated global efforts to compile region-specific policy and governance data for future research.

To clarify this limitation and propose a path forward, we have revised the manuscript as follows (Revised lines 413–419):

“While national datasets offer a broad overview, they may not adequately capture local variations that are essential for developing region-specific strategies. For example, in China, certain regions are transitioning from small-scale farming to more intensive agricultural practices—a shift that national-level data may overlook⁴⁶. Future research could benefit from establishing a global collaborative framework among researchers to systematically collect and analyze regional agricultural policies, thus enabling more detailed and fine-grained assessments.”

5: *Concerning literature review and novelty, the discussion was expanded to include PUE mechanisms and recent references like those on integrated nutrient management. Nonetheless, there is limited engagement with cutting-edge developments such as precision agriculture or CRISPR-edited crops, and the study's originality remains questionable due to its reliance on established methods like NUE_{bala} instead of NUE_{diff} .*

Response: We thank you for your critical insights regarding the novelty and methodological choices in our study. In particular, we appreciate your recognition of our efforts to expand the discussion on

phosphorus use efficiency (PUE) mechanisms and to incorporate recent literature on integrated nutrient management. Below, we address your concerns about methodological originality and engagement with cutting-edge developments.

First, we would like to clarify that our analysis is based on the $N(P)UE_{diff}$ framework, not the more widely used $N(P)UE_{bala}$ method. This distinction is essential: while NUE_{bala} relies on aggregated input-output balances that can mask actual nutrient use efficiency, $N(P)UE_{diff}$ directly quantifies the response of crop yield to inorganic fertilizer application by comparing fertilized and unfertilized plots. This approach better captures actual nutrient utilization and avoids confounding from organic sources or soil legacy effects, thereby providing a more mechanistic and precise indicator of fertilizer-driven efficiency.

With regard to your suggestion to engage with emerging technologies such as precision agriculture and CRISPR-edited crops, we fully agree on their transformative potential. However, these innovations remain in early adoption phases and are often limited to specific regions or high-input systems. Moreover, globally harmonized, long-term datasets on the deployment and outcomes of such technologies are currently unavailable, which precludes their integration into the type of global, multi-decadal analysis conducted here. Our study aims to reveal structural, long-term trends in nutrient use efficiency using consistent, widely available indicators across 200+ countries over six decades. This foundational work can inform future assessments that integrate finer-scale or technology-specific data as they become more globally accessible.

To emphasize the methodological foundation and scope of our study, we have revised the main text accordingly (Revised lines 138–145):

“Here, we aimed to address these knowledge gaps by compiling a comprehensive global database on nutrient use efficiency (NUE and PUE), comprising 3,360 observations across 205 countries and regions (Fig. 1). We will analyze trends and patterns of NUE and PUE using the $N(P)UE_{diff}$ framework for major crops, utilizing dynamic national-scale data that includes crop yields, cropland areas, fertilizer N and P input intensities, nutrient uptake by crops, and residue-grain ratios. Our focus will be on key global crops—specifically rice, wheat, maize, and soybean—that collectively account for over half of global crop production and 49% of cropland area^{17,18.}”

Extended Data Fig. 1 Research period (a) and workflow (b) on global inorganic fertilizer nitrogen use efficiency (NUE) and phosphorus use efficiency (PUE) trends and patterns.

Our research involved the two periods of the first Green Revolution and the second Green Revolution. During these periods, we first compiled and assimilated the NUE, PUE and environment variables at each sampling point, sourced from meta-analysis, to acquire optimized parameter values and the predicted model of nitrogen(phosphorus) from soil. N and P difference approach were employed to compute historical data on global NUE and PUE from 1960 to 2018, utilizing the predicted model of nitrogen(phosphorus) from soil, parameters from IFA and FAOSTAT. These historical data were utilized to depict the historical temporal trends and respective breakpoints of global NUE and PUE through segmental fitting. Moreover, random forest model predicted the patterns of global NUE and PUE from 2000 to 2022, using the optimized parameters, climate (n = 6), soil (n = 9), agriculture management (n = 7) and socio-economic (n = 4) factors. The predicted

values were further applied to calculate the global N and P surplus.

6: Finally, in addressing climate change and policy implications, CMIP6 scenarios (SSP126/245/585) were introduced, projecting fluctuations in NUE and PUE. However, the analysis remains superficial without mechanistic explanations—for instance, why PUE increases under high-emission scenarios—and while policy recommendations have become more specific, such as promoting no-till practices, they lack assessments of scalability or cost-benefit analyses.

Response: We sincerely thank you for your insightful comments regarding the treatment of climate change scenarios and policy implications. In particular, we appreciate your concerns about the lack of mechanistic explanation for PUE increases under high-emission scenarios, and the need for more robust assessments of the scalability and economic feasibility of policy recommendations.

In response to your first point, we have expanded the Looking forward section to incorporate mechanistic reasoning behind the observed PUE trends under SSP585. While nitrogen uptake may decline due to heat-induced root stress, elevated atmospheric CO₂ has been shown to enhance root exudation and rhizosphere microbial activity, thereby mobilizing phosphorus from the soil and improving plant uptake efficiency. This decoupling helps explain the increase in PUE despite declining NUE in high-emission scenarios.

Regarding your second point on the economic and practical viability of policy suggestions—such as the promotion of no-till farming—we fully agree that assessing scalability and cost-effectiveness is essential. However, due to the current lack of harmonized global data on labor, equipment access, and regional economic constraints, a formal cost-benefit analysis is beyond the scope of this study. That said, we emphasize that no-till practices—among various conservation agriculture methods—typically entail relatively low labor input increases and offer long-term agronomic and environmental benefits, making them a plausible strategy in many developing regions.

To address these issues, we have added the following revisions to the manuscript (Revised lines 424-430):

“This may be attributed to the elevated temperatures predicted under this scenario, which significantly impair root nitrogen uptake capacity, thereby reducing NUE⁴⁷. Additionally, in high-emission scenarios, increased carbon dioxide concentrations can enhance root exudation of organic compounds and organic acids. These exudates promote the mobilization of soil phosphorus and stimulate microbial activity, thereby improving the plant’s phosphorus uptake efficiency and contributing to the observed increase in PUE⁴⁸.”

Reviewer #3 (Remarks to the Author):

General Response: Thank you for your thoughtful and continued engagement with our manuscript. According to the *Nature Communications* editorial process, this is our final opportunity to respond to reviewer feedback. To address your main concerns with clarity and precision, we have divided your comments into specific points and provided detailed responses to each below. We hope these efforts demonstrate our sincere commitment to addressing your concerns and improving the scientific transparency and robustness of our work.

I: I thank the authors for the implemented changes. Yet, I am not 100% convinced by the modifications done by the authors to accommodate the major comment I made. The comment was that the relationships between nutrient use efficiency, soil nutrient supply, and nutrient input intensity neglect the impact of climate, soil, and management (beyond input intensity, e.g. timing, placement etc), and that they therefore tend to overemphasize the impact of nutrient input intensity on the observed historical changes in nutrient use efficiency. Intensity of nutrient input and efficiency are so much correlated in the computation, that there is a risk of an artificial construct where greater use of nutrients automatically leads to lower efficiency (conversely, lower use leading to greater efficiency), neglecting how historical changes in management (and possibly climate) might have altered these connections.

Response: We thank you for your insightful comments on the potentially confounding relationship between nutrient input intensity and nutrient use efficiency (NUE/PUE), and your concern that this relationship may obscure the influence of other critical factors such as climate, soil properties, and management practices beyond input quantity (e.g., timing and placement). You rightly point out the risk that the observed correlation between input intensity and efficiency may be, in part, an artifact of how efficiency is defined and computed, especially if broader agronomic context is insufficiently accounted for.

In response, we have revised both the methodological description and the supporting data structure to better reflect the inclusion of climate, soil, and management factors in our modeling of crop nutrient uptake from soil. Specifically, we now clarify that nutrient uptake from soil was not only calculated based on input-output balances but also predicted using a random forest (RF) model trained on a diverse set of environmental and agronomic predictors. These include mean annual temperature and precipitation (MAT, MAP), soil characteristics (bulk density, texture, pH, total and available nutrients), and management factors (input rates of N and P fertilizers). The RF model was trained and cross-validated using a systematic review-derived dataset of field observations.

Additionally, we have added a clear rationale for not including temporal variation in soil properties due to the absence of globally resolved interannual soil data, while explicitly stating that climate and management variables were allowed to vary annually (based on CRU TS49 and FAOSTAT datasets).

These clarifications aim to explain that our estimates of NUE/PUE trends are driven not solely by fertilizer input intensity but also with the underlying biophysical and management complexity that shapes crop nutrient acquisition

To address this, we have revised the methods as follows (Revised lines 537-559):

“Predicted nutrient uptake by crops from soil

To develop predictive models of nutrient uptake by crops from soil across different crop types (Supplementary Fig. 3), we first calculated the proportion of nitrogen (*nitrogen from soil*, %)

and phosphorus (*phosphorus from soil, %*) uptake from soil, based on data (including Input N(P), crop N(P) uptake and NUE(PUE)) originated from a systematic review of peer-reviewed publications, its detailed search strategy and data extraction procedures were delineated in *Global NUE and PUE at sampling points*. The calculated formula as follows:

$$\text{Nitrogen(Phosphorus) from soil} = \frac{\text{crop N(P) uptake} - \text{NUE(PUE)} \times \text{Input N(P)}}{\text{crop N(P) uptake}} \times 100\% \quad (5)$$

Then, we utilized above-calculated nitrogen(phosphorus) from soil, climate (MAT, MAP), soil properties (BD, sand, silt, clay, TC, TN, TP, AP, pH) and agriculture management (Input N, Input P) factors to train random forest model across different crops (Supplementary Fig. 3). These environmental variables were sourced from a systematic review of peer-reviewed studies.

The dataset was partitioned into 80% for training and 20% for testing, sampled randomly. Parameter tuning was conducted via grid search to identify the combination that minimized root mean square error (RMSE). Each final model was built using a 10-fold cross-validation approach to ensure robustness and predictive accuracy.

Finally, the trained RF models (Supplementary Fig. 3) were employed to predict annual nitrogen and phosphorus uptake from soil for the four major crops. Climate data (MAT, MAP) were obtained from CRU TS⁴⁹, while agriculture management data (Input N, Input P) factors were sourced from FAOSTAT (Supplementary Table 2). Both datasets exhibit interannual variability. Since soil properties are relatively stable over time and no globally comprehensive datasets capture their interannual variations, temporal changes in soil properties were not incorporated into the model (Supplementary Table 2)."

Moreover, the attached table of predictor variables for predicting nitrogen(phosphorus) from soil has been supplemented (Supplementary Table 2).

Table S2 List of data used for calculating nitrogen from soil and phosphorus from soil.

Predictor	Unit	Original resolution	Year	Source
MAT	°C	0.5°×0.5°	1961-2018	(CRU TS) ⁴
MAP	mm	0.5°×0.5°	1961-2018	(CRU TS) ⁴
BD	g cm ⁻³	1 km	2014	(Shangguan et al., 2014) ⁵
Sand	%	1 km	2014	(Shangguan et al., 2014) ⁵
Silt	%	1 km	2014	(Shangguan et al., 2014) ⁵
Clay	%	1 km	2014	(Shangguan et al., 2014) ⁵
TC	g kg ⁻¹	1 km	2014	(Shangguan et al., 2014) ⁵
TN	g kg ⁻¹	1 km	2014	(Shangguan et al., 2014) ⁵
TP	g kg ⁻¹	1 km	2014	(Shangguan et al., 2014) ⁵
AP	mg g ⁻¹	1 km	2014	(Shangguan et al., 2014) ⁵
pH	/	1 km	2014	(Shangguan et al., 2014) ⁵
Input N	kg ha ⁻¹	/	1961-2018	FAOSTAT
Input P	kg ha ⁻¹	/	1961-2018	FAOSTAT

Note: MAT, mean annual temperature; MAP, mean annual precipitation; Sand, sand content; Silt, silt content; Clay, clay content; BD, bulk density; TC, total carbon; TN, total nitrogen; TP, total phosphorus; AP, available phosphorus; pH, soil pH; Input N, nitrogen fertilizer input; Input P, phosphate fertilizer input.

Fig. S3 Prediction of weighted effect values of nitrogen and phosphorus from soil of the four major crops (rice, wheat, maize, soybean) and all crops by random forest models.

MAT, mean annual temperature; MAP, mean annual precipitation; Sand, sand content; Silt, silt content; Clay, clay content; BD, bulk density; TC, total carbon; TN, total nitrogen; TP, total phosphorus; AP, available phosphorus; Input N, nitrogen fertilizer input; Input P, phosphate fertilizer input. The significance of the variables in Figure is measured by the "percentage of increase of mean square error" (%IncMSE) value in Random Forest, where higher %IncMSE values imply more important variables and identify the significance of each variable. *: $P < 0.05$; **: $P < 0.01$; ***: $P < 0.001$; ****: $P < 0.0001$. The values at the top of the graph are the Var explained (R^2) and the Mean of squared residuals (MSR) for the full model. We counted the proportions of each class of factors (Fig. 3) in the random forest and represented them in a circle plot. We demonstrate the prediction performance of 80 percent of the data for the training set versus 20 percent of the test set, where R^2 represents the correlation between the observed nitrogen(phosphorus) from soil and the

predicted nitrogen(phosphorus) from soil.

2: The authors updated the prediction of soil nutrient supply with a machine learning model that now accounts for climate variable (mean annual rainfall and temperature), and soil characteristics. But do these predictors vary historically from a year to another? For soil I doubt because there is no such historical dataset of soil characteristics – and I cannot find the information in the manuscript. For climate what was the source of the data? Management data is still not included in the computation of the efficiency, as far as I understand (eq 8,9,10,11).

Response: Thank you for your thoughtful observations regarding the temporal resolution of the predictor variables used in our updated random forest model of soil nutrient supply. You are right to question whether these predictors vary historically, and we have clarified this more explicitly in the revised manuscript and supplementary materials.

Among the variables used in our model, climate factors (mean annual temperature [MAT] and precipitation [MAP]) and agricultural management inputs (Input N and Input P) vary annually and were updated for each year of prediction. Climate data were sourced from the CRU TS v4.9 dataset, while fertilizer inputs were obtained from FAOSTAT, both of which provide continuous time series data at the country level. These time-varying predictors allow the model to capture the dynamic influence of climate and management on nutrient uptake from soil.

As you correctly noted, soil properties (e.g., bulk density, texture, pH, total and available nutrients) were treated as static in our analysis due to the current lack of globally available annual soil datasets. While this limitation may introduce some bias in the estimation of interannual changes in nutrient supply from soil, the model still explains 52.04% to 92.91% of the observed variability in soil-derived nutrient uptake across major crop systems (Supplementary Fig. 3). We acknowledge that the use of static soil properties introduces a limitation to our analysis, as it does not capture potential changes in soil nutrient availability over time due to factors such as erosion, land use change, or climate change. However, given the lack of globally available annual soil datasets, we believe that this approach represents a reasonable compromise between data availability and model complexity. Future research should focus on incorporating dynamic soil properties into these types of analyses.

Regarding your comment on management variables not being included in the efficiency equations (Eq. 8–11): you are correct that management practices such as fertilizer timing and placement are not explicitly represented in the efficiency formulation due to lack of global harmonized data. However, by incorporating fertilizer input rates as predictors in our soil nutrient model, we partially account for management intensity. We view the inclusion of more granular management variables (e.g., application timing, irrigation) as a key direction for future work.

To reflect these points, we have revised the methodological section and supplementary information as follows (Revised lines 537-559):

“Predicted nutrient uptake by crops from soil

To develop predictive models of nutrient uptake by crops from soil across different crop types (Supplementary Fig. 3), we first calculated the proportion of nitrogen (*nitrogen from soil*, %) and phosphorus (*phosphorus from soil*, %) uptake from soil, based on data (including Input N(P), crop N(P) uptake and NUE(PUE)) originated from a systematic review of peer-reviewed publications, its detailed search strategy and data extraction procedures were delineated in *Global NUE and PUE at sampling points*. The calculated formula as follows:

$$\text{Nitrogen(Phosphorus) from soil} = \frac{\text{crop } N(P)\text{uptake} - \text{NUE(PUE)} \times \text{Input } N(P)}{\text{crop } N(P) \text{ uptake}} \times 100\% \quad (5)$$

Then, we utilized above-calculated nitrogen(phosphorus) from soil, climate (MAT, MAP), soil properties (BD, sand, silt, clay, TC, TN, TP, AP, pH) and agriculture management (Input N, Input P) factors to train random forest model across different crops (Supplementary Fig. 3). These environmental variables were sourced from a systematic review of peer-reviewed studies.

The dataset was partitioned into 80% for training and 20% for testing, sampled randomly. Parameter tuning was conducted via grid search to identify the combination that minimized root mean square error (RMSE). Each final model was built using a 10-fold cross-validation approach to ensure robustness and predictive accuracy.

Finally, the trained RF models (Supplementary Fig. 3) were employed to predict annual nitrogen and phosphorus uptake from soil for the four major crops. Climate data (MAT, MAP) were obtained from CRU TS⁴⁹, while agriculture management data (Input N, Input P) factors were sourced from FAOSTAT (Supplementary Table 2). Both datasets exhibit interannual variability. Since soil properties are relatively stable over time and no globally comprehensive datasets capture their interannual variations, temporal changes in soil properties were not incorporated into the model (Supplementary Table 2).”

Moreover, we have updated the workflow diagram according to the new analytical procedures (Extended Data Fig. 1).

Extended Data Fig. 1 Research period (a) and workflow (b) on global inorganic fertilizer nitrogen use efficiency (NUE) and phosphorus use efficiency (PUE) trends and patterns.

Our research involved the two periods of the first Green Revolution and the second Green Revolution. During these periods, we first compiled and assimilated the NUE, PUE and environment variables at each sampling point, sourced from meta-analysis, to acquire optimized parameter values and the predicted model of nitrogen(phosphorus) from soil. N and P difference approach were employed to compute historical data on global NUE and PUE from 1960 to 2018, utilizing the predicted model of nitrogen(phosphorus) from soil, parameters from IFA and FAOSTAT. These historical data were utilized to depict the historical temporal trends and respective breakpoints of global NUE and PUE through segmental fitting. Moreover, random forest model predicted the patterns of global NUE and PUE from 2000 to 2022, using the optimized parameters, climate (n = 6), soil (n = 9), agriculture management (n = 7) and socio-economic (n = 4) factors. The predicted

values were further applied to calculate the global N and P surplus.

3: I reiterate my comment: I suggest the authors provide more details on the computations and its logic, in order to convince the reader that this bias does not exist, or if it does, discuss the current limitations around the data and the computations, and stick to describing the historical changes in input intensity, without concluding too strongly into possible changes in efficiency (both in the abstract and the result sections). On one of my minor comment: I don't see the locations of the reviewed studies on the map of the climate zones.

Response: Thank you for reiterating your important comment regarding potential computational bias and the interpretation of nutrient use efficiency (NUE/PUE) trends. In response, we have taken several steps to clarify our methodological logic and refine the framing of our conclusions.

First, we have revised Extended Data Fig. 1 to more transparently depict the computational steps, including how historical changes in fertilizer input intensity, climate, and soil nutrient supply were modeled and integrated. We explicitly separate input-driven variability from modeled soil nutrient contributions and have expanded the description of the assumptions and data constraints inherent in these calculations.

Second, in line with your suggestion, we have softened our language in both the Abstract and Results sections to avoid overstating conclusions regarding changes in efficiency over time. Rather than attributing trends solely to improvements or declines in efficiency, we now describe historical changes in nutrient use intensity and observed NUE/PUE patterns, while highlighting uncertainties stemming from data and model limitations. We now clearly state that while we observe trends in NUE/PUE, attributing causality remains complex due to interacting factors (e.g., management, climate, and legacy soil effects) not fully captured in the available data.

Lastly, we have addressed your minor comment by updating Extended Data Fig. 7 to overlay the geographic distribution of the reviewed studies with the global climate zones, allowing readers to better assess the spatial coverage of the underlying data.

Extended Data Fig. 7 The five major climate zones globally of sample points based

on the Köppen classification¹.

Count represents the frequency of the collected samples under the same location. NUE, nitrogen use efficiency; PUE, phosphorus use efficiency.

REVIEWER COMMENTS

Reviewer #2 (Remarks to the Author):

I believe the authors have adequately addressed all my concerns, and I am satisfied with the revised manuscript.

Response: We sincerely thank you for your constructive feedback and your positive evaluation of our revised manuscript. We are grateful for your time and valuable insights, which have significantly improved the clarity and quality of our work.

Reviewer #3 (Remarks to the Author):

We thank you for the careful reading of our manuscript and for the valuable comments that helped improve the clarity and precision of our work. Below, we respond point-by-point to your suggestions.

Comment 1: I would like to thank the authors for the improvement in the description in the methods used to compute historical change in nutrient use efficiency. It is now clear that no management data beyond input use intensity was used in the computation, nor possible changes in soil nutrient over time. Therefore, the analysis cannot robustly conclude on the direction of historical change in nutrient use efficiency, and this should be reflected both in the abstract and the result section. In their response, the authors indicate that ‘we have softened our language in both the Abstract and Results sections to avoid overstating conclusions regarding changes in efficiency over time’. However, I don’t see any changes in that direction in the manuscript file with track change. The manuscript cannot be published as such.

Response: We sincerely thank the reviewer for this important comment and fully agree with the concern. To address it, we have thoroughly revised the relevant sections of the manuscript to ensure that our language regarding historical changes in nutrient use efficiency (NUE and PUE) does not imply causal inference unsupported by the data. Specifically:

In the Abstract, we have removed language suggesting direct attribution of improvement in historical NUE/PUE to specific drivers. The Abstract now reads:

“Enhancing nitrogen (N) and phosphorus (P) use efficiency (NUE and PUE) is essential for advancing sustainable agriculture and reducing dependency on non-renewable fertilizers. However, the long-term dynamics of NUE and PUE across major crops remain poorly understood at a global scale. Here, we compiled a comprehensive global database encompassing 3,360 observations across 205 countries to analyze trends in NUE and PUE for major crops from 1961 to 2018. Today, PUE and NUE continue to be suboptimal, particularly in developing regions, emphasizing the need for context-specific strategies to improve nutrient use efficiency. Global mapping highlights that NUE and PUE are highly context-dependent, with variations observed by crop type and region. For instance, rice achieves optimal NUE and PUE in tropical zones, while wheat performs best in temperate climates. Notably, maize continues to exhibit significant nutrient inefficiencies, especially in China and the United States, with considerable N and P surpluses. Taken together, this global analysis provides spatially explicit insights to guide region-specific efforts toward improving nutrient use efficiency, supporting sustainable agricultural practices and reducing global fertilizer dependence.”

In the Results section, we have clearly stated that the observed changes are correlative rather than causal. For example, we now include the Results section: “The first Green Revolution spurred a substantial global increase in food production, primarily through the adoption of high-yielding crop varieties and intensified fertilizer use²⁰. While this transformation enhanced short-term productivity, it coincided with a period of decline in estimated NUE and PUE. Several factors may have contributed to this pattern. For example, high-yielding varieties often required substantial nutrient inputs, which encouraged the routine application of large amounts of N and P fertilizers⁹. Prior studies suggest that this approach, while effective in boosting yields, placed limited emphasis on improving nutrient uptake efficiency. Moreover, continuous heavy use of inorganic fertilizers—particularly nitrogen—has been associated with soil acidification, which can negatively affect nutrient absorption^{7,12}. During this period,

breeding programs primarily targeted yield and disease resistance, rather than traits related to nutrient efficiency²⁰. However, we note that our analysis does not explicitly model these agronomic or soil-related mechanisms; thus, the observed decline in NUE and PUE during the early Green Revolution should be interpreted as a temporal correlation rather than a direct causal relationship. In summary, while the Green Revolution successfully addressed food security challenges, it may also have contributed to inefficiencies in nutrient use that persisted into subsequent decades.”

All changes are now correctly marked in the revised manuscript using the track changes function, as requested.

Comment 2: Also there are several instances in the result section where the computed change in nutrient use efficiency is analysed with the lens of changes in management (beyond mere intensity, e.g. ‘precision fertilization, use of efficient varieties) and/or soil characteristics (e.g. soil acidification). One cannot explain the output of a model with processes that are not incorporated into that model. What the authors show here is a decline in efficiency when input intensity increases. They have made no contribution in understanding how precision fertilisation and change in soil characteristics have influenced that efficiency. That should be very clear in both abstract and result section.
Response: We appreciate this critical observation and agree that interpretations invoking mechanisms not included in the model must be treated with appropriate caution. In response, we have revised the relevant sections of the Results to clearly distinguish between empirical findings and literature-based interpretations:

We have removed speculative statements implying mechanistic causality, such as those referencing “precision fertilization” and “efficient varieties,” and instead acknowledged these only as possible contextual explanations cited in prior studies. For example: “While not directly assessed in this study, these practices have been proposed in prior work as potential avenues to enhance nutrient use efficiency.”

In all cases where changes in NUE and PUE are discussed alongside potential mechanisms, we have added clarifying statements to emphasize that our analysis does not include these processes explicitly. For example: “However, our analysis does not incorporate direct data on soil acidification or crop genetics, and therefore we do not attribute the observed efficiency trends to these factors mechanistically.”

We wish these revisions fully address your concern, and we thank you again for helping us clarify the scope and interpretation of our findings.